# Low-Rank and Sparsity Are All You Need: Exploring Robust Hierarchical Latent Subspaces for Transferable Adversarial Attack

Shuangshuang Pu [1]   Wen Yang [1]   Min Li [1]   Guodong Liu [2]   Chris Ding [3]   Di Ming [1 4]

## Abstract

Adversarial examples pose serious threats to deep neural networks, exposing fundamental vulnerabilities in model robustness. However, most existing adversarial attacks directly manipulate dense and redundant feature representations, often leading to overfitting on surrogate models and poor black-box transferability. Recent SVD-based attack attempts to exploit low-rank feature subspaces, yet its reliance on single-layer optimization and single-gradient pathway neglects structural redundancy in feature representations and hierarchical heterogeneity across layers. To address these limitations, we propose LRS-Attack, a low-rank and sparse decomposition attack that explicitly models robust hierarchical subspaces in latent feature spaces. Specifically, the low-rank component captures dominant semantic directions, while the sparse component captures localized and discriminative patterns. To efficiently extract low-rank structure while preserving subspace fidelity, we develop a warm-started alternating low-rank approximation algorithm. Moreover, we introduce a hierarchical mixture of robust experts that leverages depth-dependent feature characteristics and guides gradient optimization toward more transferable adversarial directions. Extensive experiments on ImageNet show that LRS-Attack consistently improves black-box transferability over state-of-the-art methods across diverse CNN/ViT architectures and defense settings. Code is available at https://github.com/AdvML-Group/LRS-Attack.

[1]School of Computer Science and Engineering, Chongqing University of Technology, Chongqing, China [2]Eli Lilly & Company, Indiana, USA [3]School of Data Science, The Chinese University of Hong Kong, Shenzhen, China [4]Chongqing Key Laboratory of Trusted Human-Machine Interaction and Security/Safety for Intelligent Connected Vehicles, Chongqing, China. Correspondence to: Di Ming <diming@cqut.edu.cn>.

*Proceedings of the $43^{rd}$ International Conference on Machine Learning*, Seoul, South Korea. PMLR 306, 2026. Copyright 2026 by the author(s).

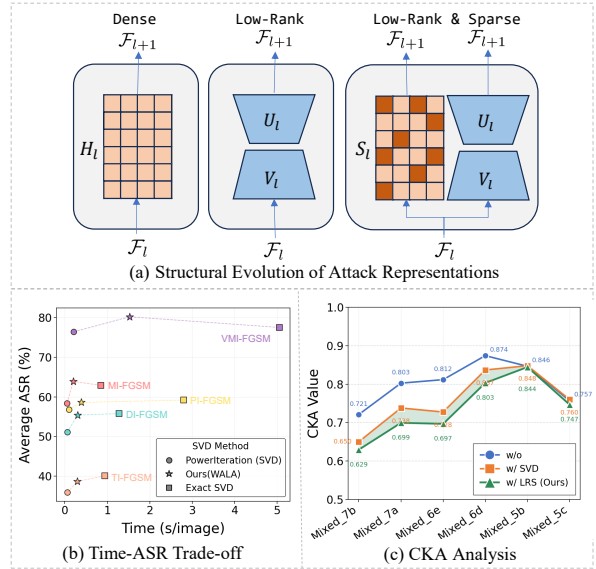

(a) Structural Evolution of Attack Representations

(b) Time-ASR Trade-off

(c) CKA Analysis

*Figure 1.* **(a) Structural evolution of the attack model:** transitioning from *"dense"* (*e.g.*, MI-FGSM) to *"low-rank"* (*e.g.*, SVD-based attack), and further to *"low-rank plus sparse"* (Ours), highlighting the progression from unstructured to structured representations. **(b) Low-rank optimization algorithm comparison across multiple attack frameworks:** built on MI/DI/TI/PI/VMI-FGSM variants, our *"Warm-started Alternating Low-rank Approximation (WALA)"* algorithm achieves the best trade-off between computational time and average attack success rate (ASR), outperforming *"Exact SVD"* and *"Power Iteration"*. **(c) Centered kernel alignment (CKA) analysis across network layers:** using Inception-v3 as the surrogate model, our LRS-Attack yields lower CKA similarity to the original features than dense attacks (*w/o*) and low-rank attacks (*w/ SVD*), indicating more effective layerwise feature disruptions.

## 1. Introduction

Convolutional neural networks (CNNs) and vision transformers (ViTs) have achieved remarkable success in vision tasks, including image classification and object detection (Simonyan & Zisserman, 2015; He et al., 2016a; Huang et al., 2017; Ren et al., 2015; Liu et al., 2021b; Carion et al., 2020) advanced the development of deep learning (Long et al., 2015; Wang et al., 2018). Despite these achievements, modern deep neural networks (DNNs) still exhibit significant robustness vulnerabilities. Adversarial attacks (Goodfel-

low et al., 2015; Szegedy et al., 2014), which introduce imperceptible yet carefully crafted perturbations, have thus become a standard tool for evaluating model robustness. In white-box settings, where model architectures and parameters are fully accessible, existing methods (Carlini & Wagner, 2017; Madry et al., 2018) can achieve high attack success rates (ASR). In contrast, black-box attacks (Guo et al., 2019; Uesato et al., 2018; Li et al., 2020; Wang et al., 2022; Ge et al., 2023b; Xie et al., 2019) cannot directly access the target model and must instead rely on adversarial transferability across models, making effective attacks substantially more challenging. Therefore, improving transferability and generalization of adversarial examples (AEs) remains a central challenge in adversarial attack research.

Existing approaches enhance transferability from various perspectives, including gradient-based optimization (Goodfellow et al., 2015; Dong et al., 2018; Gao et al., 2020; Ge et al., 2023a; Niu et al., 2025), loss-function design (Carlini & Wagner, 2017; Zhao et al., 2021; Zhang et al., 2024), and input transformations (Xie et al., 2019; Byun et al., 2022; Wang et al., 2023; 2024b). While effective, these methods predominantly operate at the input or output level, leaving the structured information embedded in intermediate representations largely underexplored.

To exploit internal representations, several feature-level attacks have been proposed. FDA (Ganeshan et al., 2019) suppresses intermediate activations toward their feature mean, while FIA (Wang et al., 2021) and NAA (Zhang et al., 2022a) perturb important neurons via masking or attribution. Although these methods effectively disrupt salient features, they still rely on densely activated and highly redundant representations as illustrated on the left of Fig. 1 (a), and lack cross-layer structural modeling. More recently, an SVD-based attack (Weng et al., 2024) introduces low-rank constraints to improve transferability by perturbing dominant singular directions at a specific intermediate layer, as shown in the middle of Fig. 1 (a). However, this strategy remains restricted to a single-layer and tends to over-smooths feature representations, thereby discarding fine-grained and layer-dependent discriminative information.

To address these limitations, we propose a *Low-Rank and Sparse Decomposition Attack (LRS-Attack)*, which explicitly decomposes intermediate features into low-rank and sparse components to capture global semantic structures and localized discriminative patterns, respectively, as illustrated on the right of Fig. 1 (a). To efficiently extract low-rank components under a limited attack iteration budget, we further develop a *Warm-started Alternating Low-rank Approximation (WALA) solver*, which achieves a practical trade-off between attack effectiveness and computational efficiency across different attack frameworks, as shown in Fig. 1 (b). Building on this structured decomposition framework, we design a

*Hierarchical Mixture Of Robust Experts* that impose depth-dependent constraints, *i.e.*, sparse-dominant decomposition for shallow layers, balanced low-rank and sparse decomposition for middle layers, and low-rank-dominant decomposition for deep layers. Finally, we analyze inter-layer feature relationships using centered kernel alignment (CKA) (Cianfarani et al., 2022), which quantifies the similarity between feature representations across layers, as shown in Fig. 1 (c). The results show that LRS-Attack reduces similarity to the original features, indicating that its perturbations more effectively disrupt layer-specific representations and thereby enhance adversarial transferability across models.

Overall, our main contributions are summarized as follows:

- We propose LRS-Attack, a low-rank and sparse decomposition attack that explicitly separates intermediate representations into global semantic and localized discriminative patterns, enabling structured feature-space modeling beyond conventional dense attacks.

- We introduce a warm-started alternating low-rank approximation (WALA) solver that reuses historical decomposition states to efficiently extract low-rank components, achieving a favorable balance between attack effectiveness and computational cost.

- We design a hierarchical mixture of robust experts that applying sparse-dominant, balanced low-rank/sparse, and low-rank-dominant constraints to shallow, middle, and deep layers, respectively, thereby improving cross-layer consistency and adversarial transferability.

- Extensive experiments across CNN and ViT architectures demonstrate that our proposed LRS-Attack consistently outperforms state-of-the-art methods in black-box attack settings, while remaining compatible with a wide range of gradient-based, input transformation-based, and feature-level attack frameworks.

## 2. Related Work

**Gradient-based Attacks.** A common paradigm for adversarial example generation is to update input perturbations along the loss gradient to induce misclassification. Representative methods include FGSM (Goodfellow et al., 2015) and its iterative or momentum-based variants, *e.g.*, I-FGSM (Kurakin et al., 2018b), MI-FGSM (Dong et al., 2018), and PGD (Szegedy et al., 2014). Subsequent works further improve transferability by stabilizing or diversifying gradient estimation, including NI-FGSM (Lin et al., 2020) and VMI-FGSM (Wang & He, 2021). Beyond gradient update rules, other methods refine the backpropagation process by modifying gradient paths, *e.g.*, LinBP (Guo et al., 2020) and TAIG (Huang & Kong, 2022), or encourage optimiza-

tion toward flatter loss regions to obtain more transferable perturbations, *e.g.*, PGN (Ge et al., 2023a).

**Transformation-based Attacks.** To enhance black-box transferability, transformation-based attacks apply stochastic or structured input transformations during adversarial optimization. Widely used strategies include random resizing and padding (DI/DIM (Xie et al., 2019)), translation-invariant smoothing (TI/TIM (Dong et al., 2019)), multi-scale perturbations (SI (Lin et al., 2020)), and local noise amplification (PI/PIM (Gao et al., 2020)). Recent extensions further explore frequency-domain manipulations, *e.g.*, SSA (Long et al., 2022), and block-level randomization, *e.g.*, SIA (Wang et al., 2023) and BSR (Wang et al., 2024b), to improve structural generalization.

**Feature-based Attacks.** Another line of work enhances transferability by manipulating internal representations of the surrogate model. Some methods disrupt salient features by aggregating intermediate gradients, *e.g.*, FIA (Wang et al., 2021), RPA (Zhang et al., 2022b), BFA (Wang et al., 2024c), SMP (Yang et al., 2025), and P2FA (Liu et al., 2025), while others weight important neurons according to their attribution scores, *e.g.*, NAA (Zhang et al., 2022a), DANAA (Jin et al., 2023). Recently, an SVD-based attack (Weng et al., 2024) retains dominant singular directions of intermediate activations, and can be incorporated into various attack methods for further enhancing adversarial performance.

However, existing attacks still face challenges in black-box transfer settings, largely due to their reliance on *densely activated, redundant feature representations* and *single-layer modeling*. To address this limitation, we aim to improve transferability by leveraging complementary global and localized feature information through *low-rank and sparse decomposition* across *hierarchical network layers*.

## 3. Methodology

### 3.1. Preliminary

**Notations and Definitions.** Let $\boldsymbol{x} \in \mathbb{R}^{C \times H \times W}$ denote a clean input image, where $C$, $H$, $W$ correspond to the channel, height, and width dimensions, respectively. The associated ground-truth label is represented as $y \in \{1, 2, \cdots, K\}$, with $K$ being the total number of categories in the classification task. A deep neural network (DNN) for image classification is expressed as $\mathcal{F} = \mathcal{F}^{(L)} \circ \cdots \circ \mathcal{F}^{(1)}$, where $\mathcal{F}^{(i)}$ denotes the $i$-th DNN layer and $L$ is the total number of layers. $\mathcal{F}_{i:j} = \mathcal{F}^{(j)} \circ \cdots \circ \mathcal{F}^{(i)}$ denotes the feed-forward mapping through consecutive layers from $i$ to $j$. The discrepancy between $\mathcal{F}(\boldsymbol{x})$ and the ground-truth label $y$ is measured by a loss function $\mathcal{L}(\cdot, \cdot)$, typically instantiated as $\mathcal{L}(\mathcal{F}(\boldsymbol{x}), y)$. For a correctly classified clean image, the model predicts $\mathcal{F}(\boldsymbol{x}) = y$. In contrast, when an adversarial perturbation $\boldsymbol{\delta}$ is introduced, an adversarial example is

formed as $\boldsymbol{x}_{adv} = \boldsymbol{x} + \boldsymbol{\delta}$, which is crafted to cause misclassification, *i.e.*, $\mathcal{F}(\boldsymbol{x}_{adv}) \neq y$.

**Adversarial Attack.** In the adversarial setting (Szegedy et al., 2014), the goal is to construct a perturbed sample that remains visually indistinguishable from the clean input while causing the DNN to make an incorrect prediction. Formally, an adversarial example $\boldsymbol{x}_{adv}$ is obtained by solving the following constrained optimization problem:

$$\underset{\boldsymbol{x}_{adv}}{\arg\max} \ \mathcal{L}(\mathcal{F}(\boldsymbol{x}_{adv}), y), \quad \text{s.t.} \ \|\boldsymbol{x}_{adv} - \boldsymbol{x}\|_\infty \leq \epsilon, \quad (1)$$

where $\mathcal{L}$ is commonly instantiated as the cross-entropy loss, and $\epsilon$ enforces an $\ell_\infty$-bounded constraint on the perturbation $\boldsymbol{\delta}$. In practice, however, existing attacks typically solve Eq. 1 via direct gradient updates over densely activated feature maps, without explicitly imposing structural constraints on the resulting perturbation directions.

**Low-Rank Adversarial Attack.** To impose structural constraints beyond dense feature optimization, recent work explores low-rank modeling to guide perturbations toward dominant semantic subspaces. Specifically, Weng *et al.* (Weng et al., 2024) construct a low-rank feature pathway by applying singular value decomposition (SVD) to the full-rank hidden representation at the $l$-th intermediate layer. The feature map at layer $l$, originally of size $C_l \times H_l \times W_l$, is first reshaped into a matrix $\boldsymbol{H}_l = \mathcal{F}_{1:l}(\boldsymbol{x}) \in \mathbb{R}^{C_l \times H_l W_l}$, so that SVD can operate on its channel–spatial flattened form. The decomposition yields $\boldsymbol{H}_l = \boldsymbol{U} \boldsymbol{\Sigma} \boldsymbol{V}^T$ where $\boldsymbol{U} = [\boldsymbol{u}_1, \cdots, \boldsymbol{u}_r] \in \mathbb{R}^{C_l \times r}$, $\boldsymbol{V} = [\boldsymbol{v}_1, \cdots, \boldsymbol{v}_r] \in \mathbb{R}^{H_l W_l \times r}$ and $\boldsymbol{\Sigma} = \text{diag}(\sigma_1, \cdots, \sigma_r) \in \mathbb{R}^{r \times r}$. A rank-1 approximation is then constructed by retaining the dominant component, $\widetilde{\boldsymbol{H}}_l = \mathcal{G}(\boldsymbol{H}_l) = \sigma_1 \boldsymbol{u}_1 \boldsymbol{v}_1^\top$, which is forwarded through the remaining layers in parallel with the original dense representation $\boldsymbol{H}_l$. This process produces two parallel logits, $\boldsymbol{z} = \mathcal{F}_{l+1:L}(\boldsymbol{H}_l) = \mathcal{F}(\boldsymbol{x})$ and $\widetilde{\boldsymbol{z}} = \mathcal{F}_{l+1:L}(\widetilde{\boldsymbol{H}}_l) = \widetilde{\mathcal{F}}(\boldsymbol{x}; l) = \mathcal{F}_{l+1:L} \circ \mathcal{G} \circ \mathcal{F}_{1:l}(\boldsymbol{x})$, which are then fused as $\beta \cdot \boldsymbol{z} + (1 - \beta) \cdot \widetilde{\boldsymbol{z}}$ for adversarial optimization:

$$\begin{aligned} \underset{\boldsymbol{x}_{adv}}{\max} \quad & \mathcal{L}\big(\beta \cdot \mathcal{F}(\boldsymbol{x}_{adv}) + (1 - \beta) \cdot \widetilde{\mathcal{F}}(\boldsymbol{x}_{adv}), y\big), \\ \text{s.t.} \quad & \|\boldsymbol{x}_{adv} - \boldsymbol{x}\|_\infty \leq \epsilon. \end{aligned} \quad (2)$$

where $\beta \in [0, 1]$ is a hyperparameter that controls the relative contribution of the dense and low-rank components.

### 3.2. Learning Robust Latent Subspace with Low-Rank and Sparse Decomposition

As discussed above, low-rank attacks extract dominant semantic subspaces from intermediate representations, but their emphasis on smooth and globally correlated structures may overlook localized discriminative feature patterns. Thus, relying solely on low-rank representation is insufficient to capture diverse adversarially effective directions.

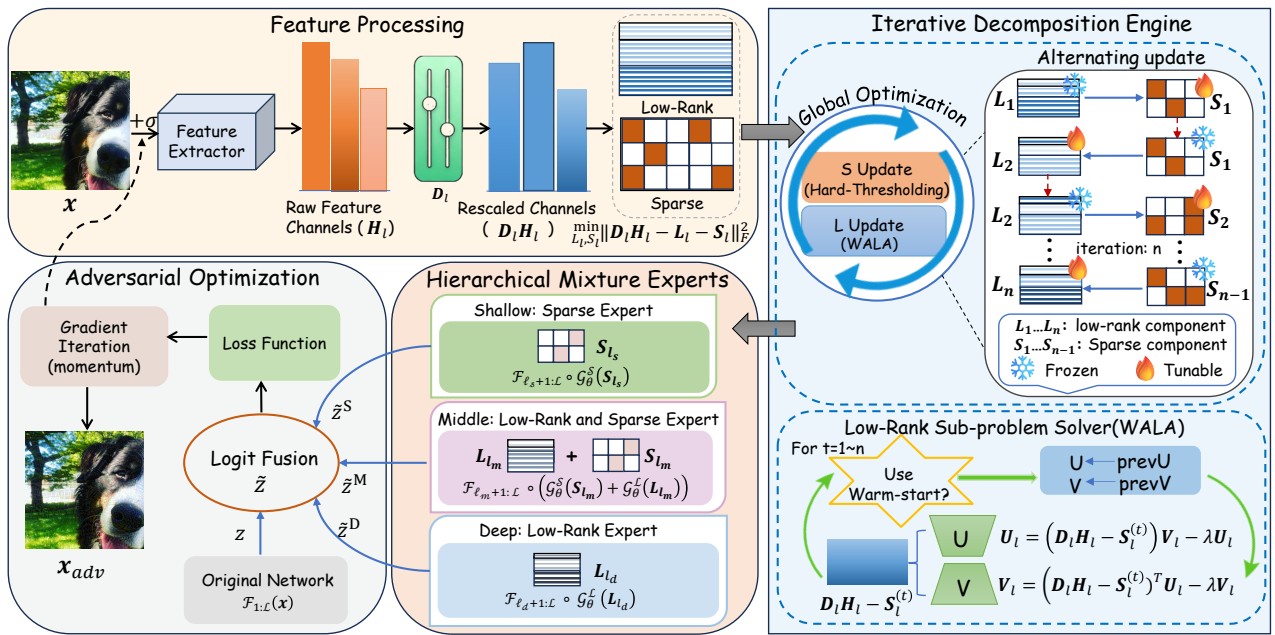

Figure 2. Illustration of the proposed LRS-Attack framework. Given an input image $x$, intermediate features are extracted, channel-wise rescaled, and decomposed into low-rank ($L_l$) and sparse ($S_l$) components through alternating optimization with hard-thresholding and the WALA solver. To leverage depth-dependent features, these components are integrated into a hierarchical mixture of experts across shallow, middle, and deep layers. The resulting fused logits guide gradient-based adversarial updates, improving black-box transferability.

**Rescaled Low-Rank and Sparse Decomposition.** Robust PCA (Wright et al., 2009; Zhang et al., 2018) is a representative low-rank and sparse decomposition framework that separates global structures from sparse deviations. Recently, related ideas such as low-rank adaptation and sparse training have also been widely explored in large language model (LLM) (Zhang & Papyan, 2025; Yu et al., 2017; Li et al., 2023), where they are used for efficient tuning, pruning, and structured compression.

However, adversarial attacks differ fundamentally from compression and pruning, as their goal is not to simplify model *weights* but to identify transferable and decision-sensitive perturbation directions in the *feature* space. This motivates us to revisit low-rank and sparse decomposition from an adversarial perspective. As shown in our preliminary analysis in Fig. 3, the two components exhibit complementary properties for adversarial transferability: the low-rank component preserves global semantic structures, while the sparse component captures local discriminative directions.

Motivated by these observations, we propose *LRS-Attack*, as illustrated in Fig. 2. Given the intermediate feature matrix $H_l$ at layer $l$, LRS-Attack explicitly decomposes it into a low-rank semantic component $L_l$ and a sparse discriminative component $S_l$, *i.e.*, $H_l \approx L_l + S_l$. By jointly modeling these complementary components, our formulation constructs a structured and expressive latent subspace that balances global semantic consistency with localized adversarial cues, thereby improving attack transferability.

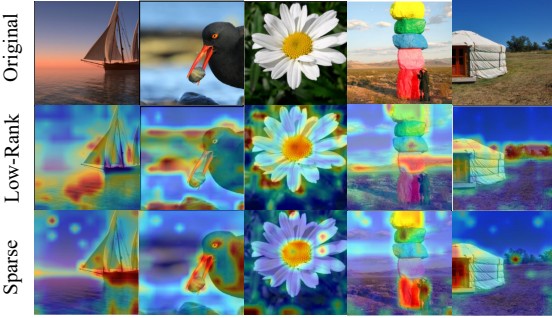

Figure 3. Visualization of the original images and the attention maps from their decomposed low-rank and sparse components. The low-rank component exhibits broader spatial responses over semantically coherent regions, whereas the sparse component highlights more localized and fine-grained discriminative patterns.

However, a small subset of channels often dominates the feature energy, which biases structured factorizations toward a few high-magnitude directions and may lead to unstable or even degenerate estimation of the low-rank subspace. To mitigate this issue, we incorporate a rescaling mechanism (Sun et al., 2024) to balance channel-wise representations before decomposition. Specifically, the latent representation $H_l$ is normalized channel-wise using the scaling matrix $D_l = (\text{diag}(H_l H_l^\top) + \epsilon I))^{-1/2}$, where $I$ is the identity matrix and $\epsilon$ is a small positive constant used for numerical stability. The rescaling representation $D_l H_l$ reduces channel-wise scale imbalance and ensures a well-conditioned decomposition. A detailed analysis of the rescaling strategy is provided in Appendix B.

Finally, the proposed rescaled low- and sparse decomposition is formulated as:

$$\min_{\boldsymbol{L}_l, \boldsymbol{S}_l} \quad \|\boldsymbol{D}_l \boldsymbol{H}_l - \boldsymbol{L}_l - \boldsymbol{S}_l\|_F^2,$$
$$\text{s.t.} \quad \text{rank}(\boldsymbol{L}_l) \leq r, \ \|\boldsymbol{S}_l\|_0 \leq k, \tag{3}$$

where the rank $r$ and sparsity level $k$ control the contributions of the low-rank and sparse components, respectively.

**Alternating Updates with Hard-Thresholding Operator and Regularized Low-Rank Approximation.** To ensure efficient adversarial generation, we solve Eq. 3 using an alternating optimization strategy (Zhang & Papyan, 2025; Zhou & Tao, 2011; Bertsimas et al., 2023). Specifically, the sparse component $\boldsymbol{S}_l$ and the low-rank component $\boldsymbol{L}_l$ are updated alternately, with one fixed while optimizing the other. This reformulates the non-convex problem into two tractable subproblems, each allowing either a closed-form update or an efficient approximation.

*(i) Sparse Component Update.* Fixing the low-rank component $\boldsymbol{L}_l^{(t-1)}$, the update of the sparse component reduces to:

$$\boldsymbol{S}_l^{(t)} = \arg\min_{\|\boldsymbol{S}_l\|_0 \leq k} \|(\boldsymbol{D}_l \boldsymbol{H}_l - \boldsymbol{L}_l^{(t-1)}) - \boldsymbol{S}_l\|_F^2. \tag{4}$$

This subproblem has a closed-form global solution via hard-thresholding (Bulmensath & Davies, 2009):

$$\boldsymbol{S}_l^{(t)} = \boldsymbol{M} \odot (\boldsymbol{D}_l \boldsymbol{H}_l - \boldsymbol{L}_l^{(t-1)}), \tag{5}$$

where $\boldsymbol{M}$ is a binary mask that keeps the $k$ entries with the largest magnitudes and sets all remaining entries to zero.

*(ii) Low-rank Component Update.* Fixing the sparse component $\boldsymbol{S}_l^{(t)}$, the update of the low-rank component reduces to:

$$\boldsymbol{L}_l^{(t)} = \arg\min_{\text{rank}(\boldsymbol{L}_l) \leq r} \|(\boldsymbol{D}_l \boldsymbol{H}_l - \boldsymbol{S}_l^{(t)}) - \boldsymbol{L}_l\|_F^2. \tag{6}$$

Although truncated SVD (Lin et al., 2011; Gu, 2015) provides an exact solution to this subproblem, it is computationally expensive in iterative attack settings. Power iteration (Golub & Van Loan, 2013) reduces the cost by estimating the leading singular components, but such coarse approximation often leads to suboptimal adversarial effectiveness. To achieve a better trade-off between efficiency and effectiveness, we reformulate Eq. 6 as a regularized low-rank factorization problem:

$$\min_{\boldsymbol{U}_l, \boldsymbol{V}_l} \|(\boldsymbol{D}_l \boldsymbol{H}_l - \boldsymbol{S}_l^{(t)}) - \boldsymbol{U}_l \boldsymbol{V}_l^\top\|_F^2 + \lambda \|\boldsymbol{U}_l\|_F^2 + \lambda \|\boldsymbol{V}_l\|_F^2, \tag{7}$$

where $\boldsymbol{U}_l \in \mathbb{R}^{C_l \times r}$, $\boldsymbol{V}_l \in \mathbb{R}^{H_l W_l \times r}$, and $\lambda$ controls the regularization strength. In practice, $\boldsymbol{U}_l$ and $\boldsymbol{V}_l$ are alternately updated with only a few gradient descent steps. The regularization terms in Eq. 7 stabilize the optimization and promote

---

**Algorithm 1** Warm-started Alternating Low-rank Approximation (WALA) Algorithm

---

**Input:** feature representation $\boldsymbol{H}_l$, rescaling matrix $\boldsymbol{D}_l$, sparse component $\boldsymbol{S}_l$, target rank $r$, maximum iterations $n_{\text{iter}}$, tolerance tol, previous factors $\boldsymbol{U}_l^{\text{prev}}, \boldsymbol{V}_l^{\text{prev}}$, perturbation $\epsilon$.
**Output:** Low-rank factors $\boldsymbol{U}_l, \boldsymbol{\Sigma}_l, \boldsymbol{V}_l$.
1: **if** *warm-start* and *previous factors available* **then**
2: $\quad \boldsymbol{U}_l = \text{QR}(\boldsymbol{U}_l^{\text{prev}} + \epsilon \cdot \mathcal{N}(0,1)), \boldsymbol{V}_l = \text{QR}(\boldsymbol{V}_l^{\text{prev}} + \epsilon \cdot \mathcal{N}(0,1))$
3: **else**
4: $\quad \boldsymbol{U}_l = 0, \boldsymbol{V}_l = \text{QR}(\text{randn}(HW, r))$
5: **end if**
6: $\mathcal{L}_0 \leftarrow \infty$
7: **for** $h = 1$ **to** $n_{\text{iter}}$ **do**
8: $\quad$ Update $\boldsymbol{U}_l$ and $\boldsymbol{V}_l$ via Eqs. 8-9
9: $\quad$ Compute the loss: $\mathcal{L}_h = \|(\boldsymbol{D}_l \boldsymbol{H}_l - \boldsymbol{S}_l) - \boldsymbol{U}_l \boldsymbol{V}_l^\top\|_F^2$
10: $\quad$ **if** $|\mathcal{L}_h - \mathcal{L}_{h-1}| <$ tol **then**
11: $\quad\quad$ **break**
12: $\quad$ **end if**
13: **end for**
14: Estimate singular-value matrix: $\boldsymbol{\Sigma}_l = \boldsymbol{U}_l^\top (\boldsymbol{D}_l \boldsymbol{H}_l - \boldsymbol{S}_l) \boldsymbol{V}_l$
15: **return** $\boldsymbol{U}_l, \boldsymbol{\Sigma}_l, \boldsymbol{V}_l$

---

well-conditioned low-rank factors under a limited attack iteration budget. This leads to the following regularized update directions for the low-rank factors:

$$\boldsymbol{U}_l = (\boldsymbol{D}_l \boldsymbol{H}_l - \boldsymbol{S}_l^{(t)}) \boldsymbol{V}_l - \lambda \boldsymbol{U}_l, \tag{8}$$

$$\boldsymbol{V}_l = (\boldsymbol{D}_l \boldsymbol{H}_l - \boldsymbol{S}_l^{(t)})^\top \boldsymbol{U}_l - \lambda \boldsymbol{V}_l, \tag{9}$$

followed by QR orthogonalization. After a few iterations, typically $5 \sim 10$, the low-rank term can be efficiently approximated as $\boldsymbol{L}_l^{(t)} = \boldsymbol{U}_l^* \boldsymbol{\Sigma}_l (\boldsymbol{V}_l^*)^\top$, where $\boldsymbol{\Sigma}_l$ is obtained in closed form via projecting onto the estimated subspaces. See Appendix C.2 for the complete derivation.

To further improve the efficiency, we adopt a *warm-start strategy* that initializes $\boldsymbol{U}_l$ and $\boldsymbol{V}_l$ with the solutions from the previous adversarial iteration. Since adjacent adversarial updates usually induce only mild changes in intermediate representations, the corresponding low-rank subspaces remain relatively stable across iterations. Reusing previously estimated subspaces therefore provides a more reliable initialization than random sampling. We summarize the full procedure as the *Warm-started Alternating Low-rank Approximation (WALA)* algorithm in Algorithm 1, WALA efficiently estimates dominant low-rank structures without full SVD by combining alternating QR-projected updates (Jain et al., 2013) with warm-start initialization. The detail derivation of the WALA algorithm is provided in Appendix C.1.

**Reconstructing the Hidden Representation for Subsequent Forward Propagation.** After the alternating updates converge, we recover the hidden representation by applying the inverse channel-wise rescaling:

$$\widetilde{\boldsymbol{H}}_l = \mathcal{G}(\boldsymbol{H}_l) = \boldsymbol{D}_l^{-1}(\boldsymbol{L}_l^* + \boldsymbol{S}_l^*), \tag{10}$$

where $\boldsymbol{L}_l^*$ and $\boldsymbol{S}_l^*$ denote the converged low-rank and sparse

components, respectively. This step restores the original feature scale, enabling compatibility with subsequent forward propagation through the remaining layers.

### 3.3. Hierarchical Mixture of Robust Low-Rank and Sparse Experts

DNNs are typically over-parameterized, leading to feature redundancy and uneven information distribution across layers (Yosinski et al., 2014; Morcos et al., 2018; Raghu et al., 2017). Such layer-wise heterogeneity makes a single-layer, uniform decomposition strategy inadequate for capturing the diverse structural representations at different depths.

**Integrating Robust Low-Rank/Sparse Experts across Network Layers.** To this end, we extend the single-layer formulation in Eq. 3 to a hierarchical mixture of robust low-rank and sparse experts. This enables layer-adaptive decomposition by applying layer-wise parameters $\theta = (r^{(l)}, k^{(l)})$ across $\{\mathbb{L}_S, \mathbb{L}_M, \mathbb{L}_D\}$ in a structure-aware and cross-layer consistent manner. To be specific, we assign different experts to shallow, middle, and deep layers, corresponding to *sparse-dominant*, *balanced low-rank/sparse*, and *low-rank-dominant* decompositions, respectively.

(i) *Sparse Expert in Shallow Layers* ($l \in \mathbb{L}_S$): Shallow layers mainly encode fine-grained and localized patterns (Zeiler & Fergus, 2014; Mehta et al., 2019). Accordingly, we introduce a *"sparse expert"* to emphasize structured sparsity in these layers. Given the latent feature $\boldsymbol{H}_l$ at layer $l$, the sparse expert applies a sparse decomposition component followed by inverse channel-wise rescaling, yielding a structured representation $\widetilde{\boldsymbol{H}}_l = \mathcal{G}_\theta^S(\boldsymbol{H}_l) = \boldsymbol{D}_l^{-1}\boldsymbol{S}_l$. The resulting feature is then propagated through the remaining network layers to produce the final logits:

$$\widetilde{\boldsymbol{z}}^S = \widetilde{\mathcal{F}}^S(\boldsymbol{x}; l, \theta) = \mathcal{F}_{l+1:L} \circ \mathcal{G}_\theta^S \circ \mathcal{F}_{1:l}(\boldsymbol{x}). \qquad (11)$$

(ii) *Low-Rank and Sparse Expert in Middle Layers* ($l \in \mathbb{L}_M$): Middle layers capture mid-level patterns, such as object parts and textures (Yosinski et al., 2014; Bahri et al., 2017), which often exhibit both sparse activations and low-rank correlations. To jointly preserve these two characteristics, we adopt a *"joint low-rank and sparse expert"* that combines the complementary structural components. Given the latent feature $\boldsymbol{H}_l$ at layer $l$, this expert forms the structured representation by integrating both sparse and low-rank decomposition components: $\widetilde{\boldsymbol{H}}_l = \mathcal{G}_\theta^S(\boldsymbol{H}_l) + \mathcal{G}_\theta^L(\boldsymbol{H}_l) = \boldsymbol{D}_l^{-1}\boldsymbol{S}_l + \boldsymbol{D}_l^{-1}\boldsymbol{L}_l$. The combined feature is subsequently propagated through the remaining network layers to obtain the final logits:

$$\widetilde{\boldsymbol{z}}^M = \widetilde{\mathcal{F}}^M(\boldsymbol{x}; l, \theta) = \mathcal{F}_{l+1:L} \circ (\mathcal{G}_\theta^S + \mathcal{G}_\theta^L) \circ \mathcal{F}_{1:l}(\boldsymbol{x}). \quad (12)$$

(iii) *Low-Rank Expert in Deep Layers* ($l \in \mathbb{L}_D$): Deep layers encode high-level semantic concepts that are more

---

**Algorithm 2** LRS-Attack

**Input:** clean image $\boldsymbol{x}$, ground-truth label $y$, feature extractor $\mathcal{F}_{1:L}$, total iterations $T$, decomposition iterations $n$, perturbation bound $\epsilon$, step size $\alpha$, momentum $\mu \in [0, 1)$, loss function $\mathcal{L}(\cdot, \cdot)$, layer indices for experts $l_s, l_m, l_d$
**Output:** adversarial example $\boldsymbol{x}_{adv}$ with $\|\boldsymbol{x}_{adv} - \boldsymbol{x}\|_\infty \leq \epsilon$
1: Initialize $\boldsymbol{x}_{adv}^{(t)} = \boldsymbol{x}$, $\boldsymbol{g}_0 = \boldsymbol{0}$
2: **for** $t = 0$ **to** $T - 1$ **do**
3:     **for** each depth $l \in \{l_s, l_m, l_d\}$ **do**
4:         Extract latent feature at layer $l$: $\boldsymbol{H}_l \leftarrow \mathcal{F}_{1:l}(\boldsymbol{x}_{adv}^{(t)})$
5:         Initialize sparse/low-rank component: $\boldsymbol{S}_l = \boldsymbol{0}, \boldsymbol{L}_l = \boldsymbol{0}$
6:         **for** $j = 1$ **to** $n$ **do**
7:             Update $\boldsymbol{S}_l$ and $\boldsymbol{L}_l$ via Eq. 5 and Alg. 1
8:         **end for**
9:     **end for**
10:   Obtain original logits: $\widetilde{\boldsymbol{z}}^O \leftarrow \mathcal{F}_{1:L}(\boldsymbol{x}_{adv}^{(t)})$
11:   Obtain logits for all experts $\widetilde{\boldsymbol{z}}^S, \widetilde{\boldsymbol{z}}^M, \widetilde{\boldsymbol{z}}^D$ via Eqs. 11-13
12:   Compute aggregated logits $\widetilde{\boldsymbol{z}}$ via Eq. 14
13:   Compute loss $\mathcal{L}(\widetilde{\boldsymbol{z}}, y)$ via Eq. 15
14:   Update momentum gradient:
      $\boldsymbol{g}_{t+1} = \mu \cdot \boldsymbol{g}_t + \nabla_{\boldsymbol{x}_{adv}}\mathcal{L}(\widetilde{\boldsymbol{z}}, y)/\|\nabla_{\boldsymbol{x}_{adv}}\mathcal{L}(\widetilde{\boldsymbol{z}}, y)\|_1$
15:   Update adversarial example:
      $\boldsymbol{x}_{adv}^{(t+1)} = \text{Clip}_{\boldsymbol{x},\epsilon}(\boldsymbol{x}_{adv}^{(t)} + \alpha \cdot \text{sign}(\boldsymbol{g}_{t+1}))$
16: **end for**
17: **return** $\boldsymbol{x}_{adv}^{(T)}$

---

consistent across samples and architectures (Cheng et al., 2015; Wang et al., 2019). Accordingly, we employ a *"low-rank expert"* to extract compact and semantically consistent representations. Given the latent feature $\boldsymbol{H}_l$ at layer $l$, the low-rank expert retains the low-rank decomposition component and restores the original feature scale, producing a structured representation $\widetilde{\boldsymbol{H}}_l = \mathcal{G}_\theta^L(\boldsymbol{H}_l) = \boldsymbol{D}_l^{-1}\boldsymbol{L}_l$. The structured representation is then passed through the remaining layers to yield the final logits:

$$\widetilde{\boldsymbol{z}}^D = \widetilde{\mathcal{F}}^D(\boldsymbol{x}; l, \theta) = \mathcal{F}_{l+1:L} \circ \mathcal{G}_\theta^L \circ \mathcal{F}_{1:l}(\boldsymbol{x}). \qquad (13)$$

**Depth-Aware Hyperparameter Scheduling.** Across the hierarchical experts, the layer-wise parameters $\theta = (r^{(l)}, k^{(l)})$ are scheduled according to network depth: *as $l$ increases, the sparsity level $k^{(l)}$ gradually decreases, while the rank ratio $r^{(l)}$ increased*. This design is motivated by the hierarchical evolution of feature representations in DNNs. In shallow layers, a larger $k^{(l)}$ allows the sparse component to capture fine-grained, localized features, which are highly sensitive to adversarial perturbations. As representations become more semantics in deeper layers, a larger $r^{(l)}$ encourages perturbations to align with the globally correlated and model-agnostic subspaces. By progressively shifting from *sparsity-dominant* to *low-rank-dominant* modeling, our method effectively balances localized attack strength with global cross-model transferability.

**Hierarchical Mixture of Robust Experts.** Given the expert-specific mappings defined above, we select one representative layer from each set $\mathbb{L}_S$, $\mathbb{L}_M$, and $\mathbb{L}_D$, and assign

the corresponding sparse, low-rank/sparse expert, or low-rank expert to each layer. Preliminary experiments indicate that using one representative layer at each depth provides a favorable trade-off between runtime efficiency and attack effectiveness. The hierarchical strategy can also be naturally extended to more layers when finer-grained feature decomposition is desired.

Rather than relying on a single universal decomposition, our framework aggregates the logits from depth-dependent experts into a unified prediction, forming a *hierarchical mixture of robust experts*:

$$\widetilde{z} = w_O \cdot z + w_S \cdot \widetilde{z}_S + w_M \cdot \widetilde{z}_M + w_D \cdot \widetilde{z}_D, \quad (14)$$

where, without loss of generality, the weights are set to $w_O = w_S = w_M = w_D = 1/4$, and $z = \mathcal{F}(x)$ denotes the original non-robust dense expert. Consequently, the corresponding *hierarchical low-rank and sparse decomposition-based attack* is formulated as:

$$\max_{x_{adv}} \mathcal{L}\big(\mathcal{F}(x_{adv}) + \widetilde{\mathcal{F}}^S(x_{adv}) + \widetilde{\mathcal{F}}^M(x_{adv}) + \widetilde{\mathcal{F}}^D(x_{adv}), y\big)$$
$$\text{s.t.} \quad \|x_{adv} - x\|_\infty \leq \epsilon. \tag{15}$$

As shown above, this hierarchical mixture *exploits the complementary representational properties of shallow, middle, and deep layers*, enriching gradient diversity and yielding more informative perturbation directions for highly transferable adversarial attacks. In practice, the constrained optimization problem (15) is solved using momentum gradient descent, *i.e.*, MI-FGSM (Dong et al., 2018), and the overall optimization procedure is summarized in Algorithm 2.

## 4. Experiments

### 4.1. Experimental Setting

**Datasets.** Following prior work, we conduct experiments on the widely used ImageNet-Compatible benchmark (nip, 2017), introduced in the NIPS 2017 Competition on Adversarial Attacks and Defenses.

**Models.** In this study, we adopt both CNNs (Inception-V3 (Christian Szegedy & Wojna, 2016), Vgg-19 (Simonyan & Zisserman, 2015), DenseNet-121 (Huang et al., 2017)) and ViTs (ViT-B (Wu et al., 2020), DeiT-B(Touvron et al., 2021a), CaiT-S (Touvron et al., 2021b)) as surrogate models. We evaluate attack performance on diverse target models, including nine undefended CNNs (*i.e.*, Inc-v3, Inc-v4, Vgg-16 (Simonyan & Zisserman, 2015), Vgg-19, Desenet-121, IncRes-v2 (He et al., 2016b), Res152-v1, Res50-v1, Res101-v1 (He et al., 2016a)), four defended CNNs (*i.e.*, Inc-v3adv, IncRes-v2adv, Inc-v3ens3, IncRes-v2ens (Tramèr et al., 2018)), eight vanilla ViTs(*i.e.*, PiT-S (Heo et al., 2021), Visformer-S (Chen et al., 2021), Twins-B (Chu et al., 2021), CaiT-S, DeiT-B, ConViT-B (d'Ascoli et al., 2021), ViT-B,

Swin-S (Liu et al., 2021a)), and three defended ViTs (*i.e.*, DeiT-Sadv (Bai et al., 2021), Swin-Badv (Mo et al., 2022), XCiT-Sadv (Debenedetti et al., 2023)).

**Baseline Methods.** To comprehensively evaluate the attack performance, we integrate our method with various attack baselines, including MI-FGSM (Dong et al., 2018), DI-FGSM (Xie et al., 2019), TI-FGSM (Dong et al., 2019), PI-FGSM (Gao et al., 2020), SI-NI-FGSM (Lin et al., 2020), VMI-FGSM (Wang & He, 2021), GI-FGSM (Wang et al., 2024a), SVD (Weng et al., 2024), SIA (Wang et al., 2023), BSR (Wang et al., 2024b), PGN (Ge et al., 2023a), GGS (Niu et al., 2025), BFA (Wang et al., 2024c), and P2FA (Liu et al., 2025). In addition, we evaluate our method on representative ViT-based attacks, including MIM (Dong et al., 2018), PNA (Wei et al., 2022), TGR (Zhang et al., 2023), and ATT (Ming et al., 2024).

**Evaluation Metrics.** We adopt the average attack success rate (ASR, %) as the primary metric, where a Higher ASR indicates stronger transferability. $\text{Avg}^{\text{CNN}}$ denotes the average attack success rate (ASR) across nine standard CNN models under black-box settings, while $\text{Avg}^{\text{ViT}}$ denotes the average ASR across eight standard ViT models. $\text{Avg}^{\text{Def-CNN}}$ and $\text{Avg}^{\text{Def-ViT}}$ represent the average ASR over four defense-enhanced CNN models and three defense-enhanced ViT models, respectively. Furthermore, $\text{Avg}^{\text{Def}}$ reports the average ASR after applying four representative defense methods, including HGD (Liao et al., 2018), JPEG (Guo et al., 2018), R&P (Xie et al., 2018) and NIPS_r3 (Kurakin et al., 2018a).

**Implementation Details.** All experiments were conducted using PyTorch on an NVIDIA 3090 GPU. For a fair comparison, baseline hyperparameters follow prior work (Long et al., 2022). The maximum perturbation is set to $\epsilon = 16$, with the number of iteration $T = 10$ and step size $\alpha = \epsilon/T$. For MI-FGSM, the decay factor is set to $\mu = 1.0$. For DI-FGSM, the transformation probability is set to $p = 0.5$. For TI-FGSM, the kernel size is set to $k = 7$. For PI-FGSM, the amplification factor, project factor, kernel size are set to $\beta = 10$, $\gamma = 16$, and $k_w = 7$, respectively. For SI-NI-FGSM, the number of copies is set to $m_1 = 5$. The specific layer selections is as follows. Inc-v3 uses Mixed_5b, Mixed_5d, and Mixed_6e as the shallow, middle, and deep layers, respectively. VGG19 uses conv2_2, conv3_4, and conv4_4. DenseNet uses transition1, transition2, and transition3. For ViT/DeiT, QKV decompositions are applied at Block-4, Block-7, and Block-10. For CaiT-S, SA-Block-4, SA-Block-12, and SA-Block-20 are selected.

### 4.2. Transfer Attack using CNN Surrogate Models

We evaluate transferability with CNN surrogates by integrating our method with several gradient-based attacks, including MI-FGSM, DI-FGSM, TI-FGSM, PI-FGSM, VMI-FGSM, SI-NI-FGSM and GI-FGSM.

*Table 1.* Average ASR (%) of CNN-based attacks across different models and defense methods. Best results are highlighted in bold.

| Model | Attack | Method | Avg$^{CNN}$ | Avg$^{Def-CNN}$ | Avg$^{ViT}$ | Avg$^{Def-ViT}$ | Avg$^{Def}$ |
|---|---|---|---|---|---|---|---|
| Inc-v3 | MI-FGSM | W/O | 51.7 | 20.6 | 21.0 | 34.7 | 32.0 |
| | | SVD | 55.5 | 22.3 | 21.3 | 34.9 | 33.5 |
| | | LRS | **61.8** | **23.2** | **24.3** | **35.3** | **36.1** |
| | DI-FGSM | W/O | 42.1 | 14.1 | 14.7 | 32.0 | 25.7 |
| | | SVD | 46.3 | 14.4 | 15.0 | 31.9 | 26.9 |
| | | LRS | **49.6** | **15.3** | **16.4** | **32.5** | **28.4** |
| | TI-FGSM | W/O | 29.8 | 11.7 | 12.6 | 31.9 | 21.5 |
| | | SVD | 31.4 | 11.8 | 12.8 | **32.3** | 22.1 |
| | | LRS | **35.2** | **13.3** | **14.1** | **32.3** | **23.7** |
| | PI-FGSM | W/O | 63.4 | 42.4 | 20.7 | 45.4 | 43.0 |
| | | SVD | 65.5 | 44.8 | 21.0 | 46.3 | 44.4 |
| | | LRS | **67.7** | **45.6** | **21.6** | **47.6** | **45.6** |
| | SI-NI-FGSM | W/O | 74.4 | 36.5 | 33.8 | **37.5** | 45.5 |
| | | SVD | 77.7 | 37.2 | 35.0 | 37.1 | 46.8 |
| | | LRS | **80.4** | **39.2** | **37.0** | 37.3 | **48.5** |
| | VMI-FGSM | W/O | 68.0 | 36.0 | 37.5 | 36.1 | 44.4 |
| | | SVD | 74.0 | 41.6 | 39.8 | 36.3 | 47.9 |
| | | LRS | **76.6** | **41.7** | **41.1** | **36.8** | **49.0** |
| | GI-FGSM | W/O | 61.2 | 21.8 | 24.1 | 35.9 | 35.8 |
| | | SVD | 67.5 | 22.5 | 25.9 | 35.9 | 38.0 |
| | | LRS | **73.5** | **24.8** | **28.3** | **36.4** | **40.8** |
| Vgg-19 | MI-FGSM | W/O | 73.1 | 37.8 | 37.5 | 46.7 | 48.8 |
| | | SVD | 74.1 | 37.7 | 35.2 | 46.8 | 48.5 |
| | | LRS | **77.4** | **40.5** | **40.0** | **47.0** | **51.2** |
| | DI-FGSM | W/O | 64.6 | 25.1 | 25.9 | 40.4 | 39.0 |
| | | SVD | 63.3 | 24.6 | 24.4 | 40.5 | 38.2 |
| | | LRS | **68.1** | **27.8** | **27.7** | **40.6** | **41.0** |
| | TI-FGSM | W/O | 56.5 | 26.7 | 23.2 | 40.5 | 36.7 |
| | | SVD | 54.0 | 25.3 | 21.6 | **40.8** | 35.4 |
| | | LRS | **61.4** | **28.8** | **25.8** | 40.6 | **39.1** |
| | PI-FGSM | W/O | 69.3 | 47.4 | 23.2 | 52.8 | 48.2 |
| | | SVD | 69.1 | 47.1 | 23.1 | 53.5 | 48.2 |
| | | LRS | **71.1** | **48.6** | **24.0** | **54.2** | **49.4** |
| | SI-NI-FGSM | W/O | 86.5 | 54.7 | 47.9 | 48.1 | 59.3 |
| | | SVD | 87.2 | 53.7 | 47.3 | **48.3** | 59.1 |
| | | LRS | **89.6** | **58.3** | **51.0** | **48.3** | **61.8** |
| | VMI-FGSM | W/O | 83.6 | 52.5 | 50.6 | 48.0 | 58.7 |
| | | SVD | 84.7 | 52.6 | 49.5 | 47.6 | 58.6 |
| | | LRS | **85.4** | **54.3** | **52.5** | **48.1** | **60.1** |
| | GI-FGSM | W/O | 81.2 | 41.4 | 39.1 | 44.3 | 51.5 |
| | | SVD | 82.6 | 41.7 | 39.1 | 44.5 | 51.9 |
| | | LRS | **83.1** | **42.9** | **40.9** | **44.6** | **52.9** |
| Densenet-121 | MI-FGSM | W/O | 76.3 | 47.5 | 40.7 | 38.8 | 50.8 |
| | | SVD | 77.0 | 45.7 | 37.8 | 39.3 | 49.9 |
| | | LRS | **83.5** | **54.4** | **46.1** | **39.5** | **55.9** |
| | DI-FGSM | W/O | 73.0 | 40.4 | 33.4 | **41.3** | 47.0 |
| | | SVD | 72.7 | 38.3 | 30.2 | 41.1 | 45.6 |
| | | LRS | **79.1** | **43.9** | **36.5** | **41.3** | **50.2** |
| | TI-FGSM | W/O | 56.6 | 33.6 | 26.0 | 41.1 | 39.3 |
| | | SVD | 54.7 | 32.2 | 23.2 | 40.7 | 37.7 |
| | | LRS | **63.7** | **39.1** | **29.1** | **41.4** | **43.3** |
| | PI-FGSM | W/O | 74.1 | 61.8 | 26.4 | 55.7 | 54.5 |
| | | SVD | 74.8 | 61.7 | 25.2 | 55.2 | 54.2 |
| | | LRS | **76.9** | **65.1** | **27.3** | **56.0** | **56.3** |
| | SI-NI-FGSM | W/O | 88.4 | 68.3 | 56.4 | 49.2 | 65.6 |
| | | SVD | 88.8 | 67.1 | 53.1 | 49.0 | 64.5 |
| | | LRS | **91.5** | **73.1** | **60.4** | **49.8** | **68.7** |
| | VMI-FGSM | W/O | 88.0 | 69.1 | 61.4 | 49.4 | 67.0 |
| | | SVD | 89.4 | 70.5 | 60.6 | 49.2 | 67.4 |
| | | LRS | **91.4** | **73.2** | **64.7** | **50.2** | **69.9** |
| | GI-FGSM | W/O | 86.6 | 58.5 | 49.7 | 45.6 | 60.1 |
| | | SVD | 87.7 | 57.1 | 47.6 | 45.7 | 59.5 |
| | | LRS | **91.5** | **65.3** | **55.6** | **46.0** | **64.6** |

*Table 2.* Average ASR (%) of ViT-based attacks across different models and defense methods. Best results are highlighted in bold.

| Model | Attack | Method | Avg$^{CNN}$ | Avg$^{Def-CNN}$ | Avg$^{ViT}$ | Avg$^{Def-ViT}$ | Avg$^{Def}$ |
|---|---|---|---|---|---|---|---|
| ViT-B | MIM | W/O | 51.1 | 34.4 | 64.2 | 48.0 | 49.4 |
| | | SVD | 53.6 | 35.8 | 68.2 | 48.5 | 51.5 |
| | | LRS | **57.7** | **38.6** | **69.8** | **49.4** | **53.9** |
| | PNA | W/O | 52.7 | 33.5 | 69.0 | 45.8 | 50.2 |
| | | SVD | 53.6 | 35.0 | 71.2 | 46.2 | 51.5 |
| | | LRS | **59.8** | **38.3** | **71.8** | **47.2** | **54.3** |
| | TGR | W/O | 59.5 | 40.5 | 68.9 | 51.1 | 55.0 |
| | | SVD | 60.1 | 40.4 | 69.3 | 51.7 | 55.4 |
| | | LRS | **61.1** | **41.3** | **69.7** | **52.7** | **56.2** |
| | ATT | W/O | 65.0 | **46.2** | 77.0 | 51.8 | 60.0 |
| | | SVD | 65.8 | 45.6 | 76.5 | 52.3 | 60.1 |
| | | LRS | **66.3** | **47.2** | **77.2** | **53.1** | **60.5** |
| DeiT-B | MIM | W/O | 60.9 | 40.7 | 78.0 | 48.8 | 57.1 |
| | | SVD | 62.8 | 43.0 | 81.2 | 49.4 | 59.1 |
| | | LRS | **68.1** | **48.2** | **86.6** | **50.7** | **63.4** |
| | PNA | W/O | 63.9 | 42.8 | 98.7 | 98.5 | 76.0 |
| | | SVD | 66.8 | 45.8 | 98.9 | **98.8** | 77.6 |
| | | LRS | **71.2** | **50.1** | **99.4** | **98.8** | **79.9** |
| | TGR | W/O | 78.8 | 58.8 | 92.7 | 54.7 | 71.2 |
| | | SVD | 79.6 | 60.4 | 93.1 | 55.0 | 72.1 |
| | | LRS | **82.2** | **63.1** | **94.1** | **56.7** | **74.0** |
| | ATT | W/O | 81.4 | 63.7 | 95.2 | 54.6 | 73.7 |
| | | SVD | 81.1 | 62.4 | 95.2 | 54.1 | 73.2 |
| | | LRS | **83.4** | **65.2** | **95.8** | **56.2** | **75.2** |
| CaiT-S | MIM | W/O | 68.5 | 47.0 | 83.3 | 49.4 | 62.1 |
| | | SVD | 70.6 | 49.3 | 85.9 | 50.4 | 64.0 |
| | | LRS | **74.6** | **56.0** | **88.4** | **52.4** | **67.8** |
| | PNA | W/O | 65.1 | 43.1 | 81.4 | 45.5 | 58.8 |
| | | SVD | 68.3 | 47.7 | 85.5 | 46.2 | 61.9 |
| | | LRS | **73.7** | **53.9** | **87.4** | **49.1** | **66.0** |
| | TGR | W/O | 80.4 | 60.9 | 90.5 | 54.1 | 71.5 |
| | | SVD | 80.3 | 62.2 | 90.8 | 55.4 | 72.2 |
| | | LRS | **83.1** | **65.7** | **91.7** | **58.0** | **74.6** |
| | ATT | W/O | 81.3 | 66.2 | 91.3 | 53.8 | 73.2 |
| | | SVD | 82.0 | 66.1 | 91.8 | 55.1 | 73.8 |
| | | LRS | **83.0** | **66.8** | **92.0** | **56.9** | **74.7** |

### 4.3. Transfer Attack using ViT Surrogate Models

We further evaluate transferability with ViT surrogates by integrating our method with representative ViT-based attacks, including MIM, PNA, TGR, and ATT.

**ViT-to-ViT Transfer.** We first evaluate attack transferability across ViT architectures. As shown in Table 2, LRS consistently outperforms the baselines. For example, under MIM with DeiT-B as the surrogate, LRS achieves an Avg$^{ViT}$ of 86.6% (vs. 78.0% W/O, 81.2% SVD) and an Avg$^{Def-ViT}$ 50.7% (vs. 48.8% W/O, 49.4% SVD).

**ViT-to-CNN Transfer.** We further assess cross-architecture transfer from ViT surrogates to CNN target models. When ViT-B is used as the surrogate under MIM, LRS improves Avg$^{CNN}$ from 51.1% for W/O and 53.6% for SVD to 57.7%. Under PNA, LRS further increase Avg$^{CNN}$ from 52.7% for W/O and 53.6% for SVD to 59.8%.

The complete results of all ViT-based attacks are provided in Appendix G.2 (Tables 14-16).

### 4.4. Ablation Study

**Effect of Low-rank/Sparsity Hyperparameter on ASR.** To study the interaction between low-rank and sparsity, we evaluate five scheduling strategies ($\mathbf{S}_0 - \mathbf{S}_4$) by varying the sparsity $k$ and rank $r$ across iterations. Specifically, $\mathbf{S}_0$ represents *an opposite trend with decreasing $k$ and increasing $r$*; $\mathbf{S}_1$ and $\mathbf{S}_3$ *fix either $k$ or $r$ as control strategies*; $\mathbf{S}_2$ and $\mathbf{S}_4$ *adopt co-increasing and co-decreasing schedules*, respectively. As illustrated in Fig. 4, $\mathbf{S}_0$ consistently achieves the best ASR across architectures. This indicates that shifting from sparsity-dominant to low-rank-dominant modeling matches hierarchical feature evolution, where early sparsity

**CNN-to-CNN Transfer.** We first verify the effectiveness of our method against the normally trained and defended CNNs. As shown in Table 1, LRS consistently outperforms both the original baselines without decomposition and the SVD-based decomposition method across different surrogate models. For example, with Inc-v3 as the surrogate model, LRS improves average ASR on CNN target models by 6.3% (MI-FGSM), with similar gains on other attacks.

**CNN-to-ViT Transfer.** For cross-architecture evaluation, we test CNN-generated adversarial examples on ViT targets. As reported in Table 1. LRS maintains strong transferability across different ViT architectures. For example, with Inc-v3 used as the surrogate model, LRS improves average ASRs on ViT targets by 3.0% under MI-FGSM.

Due to space limitations, the main paper reports only the average ASR, with complete results provided in Appendix G.1 (Tables 11-13). We further extend LRS to different types of attack methods, with results in Appendix F.3 (Tables 5-7). Additional results using other CNN surrogate models are reported in Appendix F.4 (Tables 8-10).

*Table 3.* Ablation study of different expert configurations using Inc-v3 and ViT-B as surrogate models. *Spa.*, *Mix.*, and *Lrk.* denote Sparse, Mixture, and Low-rank experts, respectively. White-box and black-box ASRs (%) are reported, with the best results highlighted in bold.

| Source | Strategy | Expert | | | Inc-v3 | Vgg19 | Dense121 | CaiT-S | DeiT-B | ViT-B | Avg |
| | | Shallow | Middle | Deep | | | | | | | |
|---|---|---|---|---|---|---|---|---|---|---|---|
| Inc-v3 | w/o | - | - | - | **100.0** | 59.9 | 47.4 | 15.0 | 17.1 | 25.0 | 44.1 |
| | All-Spa. | Spa. | Spa. | Spa. | **100.0** | 65.5 | 57.5 | 18.9 | 19.0 | 25.9 | 47.8 |
| | All-Lrk. | Lrk. | Lrk. | Lrk. | 99.9 | 64.8 | 56.2 | 19.1 | 19.6 | 26.2 | 47.6 |
| | L-M-S | Lrk. | Mix. | Spa. | **100.0** | 61.4 | 51.1 | 17.1 | 18.0 | 24.5 | 45.4 |
| | L-S-M | Lrk. | Spa. | Mix. | **100.0** | 64.2 | 57.0 | 17.8 | 20.3 | **26.5** | 47.6 |
| | S-L-M | Spa. | Lrk. | Mix. | **100.0** | 65.1 | 58.4 | 18.1 | 19.2 | 26.1 | 47.8 |
| | **S-M-L** | **Spa.** | **Mix.** | **Lrk.** | **100.0** | **66.5** | **60.4** | **19.6** | **21.0** | **26.5** | **49.0** |
| ViT-B | w/o | - | - | - | 39.0 | 57.3 | 44.8 | 44.4 | 49.0 | 99.9 | 55.7 |
| | All-Spa. | Spa. | Spa. | Spa. | 52.7 | 69.6 | 59.7 | 75.2 | 79.1 | **100.0** | 72.7 |
| | All-Lrk. | Lrk. | Lrk. | Lrk. | 52.6 | 69.3 | 60.9 | 76.0 | 78.6 | **100.0** | 72.9 |
| | L-M-S | Lrk. | Mix. | Spa. | 53.0 | 70.0 | 60.2 | 76.2 | 77.4 | **100.0** | 72.8 |
| | L-S-M | Lrk. | Spa. | Mix. | 52.2 | 68.0 | 60.3 | 76.6 | 78.1 | **100.0** | 72.5 |
| | S-L-M | Spa. | Lrk. | Mix. | 52.7 | 68.3 | 60.5 | 76.5 | 76.3 | **100.0** | 72.4 |
| | **S-M-L** | **Spa.** | **Mix.** | **Lrk.** | **53.5** | **70.3** | **61.8** | **77.3** | **79.9** | **100.0** | **73.8** |

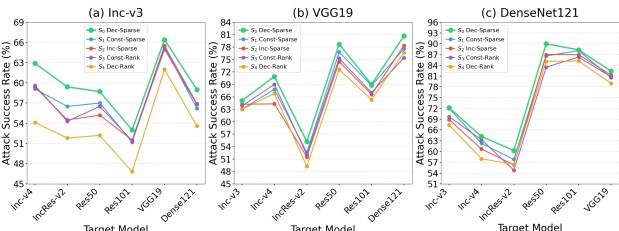

*Figure 4.* Impact of different hyperparameter strategies on ASR. Strategies $S_0$–$S_4$ correspond to different $k$-$r$ configurations.

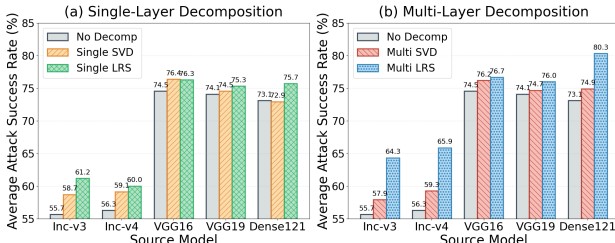

*Figure 5.* Effect of layer configuration across five CNN surrogates. Black-box ASRs are averaged over different target models.

captures local vulnerabilities and later low-rank promotes global semantic alignment. thereby improving transferability. Analyses on ViTs are provided in Appendix F.2 ( Fig. 9).

**Effect of Layer Configuration on ASR.** We evaluate both single-layer and multi-layer variants of SVD and LRS across five CNN surrogate models. As shown in Fig. 5, multi-layer LRS achieves the highest average ASR across target models (including Inc-v3, Inc-v4, Vgg16, Vgg19, and DenseNet121) by better preserving complementary structural components and enhancing black-box transferability.

**Effect of Expert Configuration on ASR.** We evaluate different expert configurations across shallow, middle, and deep stages. As shown in Table 3, all expert-based settings consistently outperform the baseline without experts, confirming the effectiveness of the expert design. Among them, the proposed S-M-L strategy achieves the best performance on both Inc-v3 and ViT-B, with average ASRs of 49.0%

*Table 4.* Feature similarity across models measured by CKA.

| Transfer | Layer | Clean | SVD | LRS |
|---|---|---|---|---|
| Inc-v3 → Inc-v4 | Pool | 0.7986 | 0.4233 | 0.4813 |
| | FC | 0.5626 | 0.1891 | 0.2086 |
| Inc-v3 → IncRes-v2 | Pool | 0.7612 | 0.3927 | 0.4495 |
| | FC | 0.5854 | 0.1873 | 0.2135 |
| Inc-v3 → Vgg-16 | Pool | 0.5065 | 0.4512 | 0.4887 |
| | FC | 0.2944 | 0.1341 | 0.1532 |
| Inc-v3 → Vgg-19 | Pool | 0.5079 | 0.4373 | 0.4767 |
| | FC | 0.2986 | 0.1310 | 0.1541 |

and 73.8%, respectively. This suggests that the sparse-to-low-rank assignment aligns well with the evolution of DNN representations from local textures to global semantics.

### 4.5. Cross-Model Representation Consistency Analysis

To verify whether LRS preserves transferable semantic structures, we measure cross-model feature similarity using Centered Kernel Alignment (CKA). As shown in Table 4, both SVD and LRS reduce CKA values compared with clean samples, indicating adversarial perturbations inevitably distort the original feature distributions. Nevertheless, LRS consistently yields higher cross-model CKA values than SVD across all transfer paths, suggesting better preservation of architecture-shared and model-agnostic representations.

## 5. Conclusion

By moving beyond existing dense and single-layer modeling, we propose LRS-Attack, a structured feature-space adversarial attack that decomposes intermediate representations into low-rank semantic structures and sparse discriminative patterns. Building on this, we further introduce a warm-started alternating low-rank approximation solver and a hierarchical mixture of robust experts to efficiently impose structural constraints across network depths. Extensive experiments on CNNs and ViTs demonstrate that the proposed LRS consistently improves black-box transferability across diverse models and defense settings, while remaining compatible with various attack frameworks.

## Acknowledgments

This work is partially supported by the National Natural Science Foundation of China (Grant No. 62441607, 62472059), the Natural Science Foundation of Chongqing, China (Grant No. CSTB2024NSCQ-MSX0341), the Science and Technology Research Program of Chongqing Municipal Education Commission (Grant No. KJQN202301142), and the Open Research Fund of Key Laboratory of Cyberspace Big Data Intelligent Security, Ministry of Education (Grant No. CBDIS202403).

## Impact Statement

This paper presents a methodological study on adversarial attacks, aiming to advance the understanding of robustness and vulnerability in deep neural networks. By exploring structured perturbation mechanisms in the feature space, our work provides insights into how adversarial examples interact with hierarchical representations in deep learning models. Although adversarial attack techniques may be misused, research in this direction is important for identifying weaknesses in existing systems and motivating the development of more trustworthy machine learning models. We do not foresee immediate negative societal consequences arising uniquely from this work beyond those already associated with adversarial robustness research.

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

# A. Interpretation of Low-Rank and Sparse Decomposition

**Problem Setup.** Given an intermediate feature tensor at layer $l$, we first reshape it into a matrix $\boldsymbol{H}_l \in \mathbb{R}^{C_l \times H_l W_l}$, where each row corresponds to a channel and each column corresponds to a spatial position. We formulate low-rank and sparse decomposition as a constrained matrix approximation problem:

$$\boldsymbol{H}_l \approx \boldsymbol{L}_l + \boldsymbol{S}_l, \tag{16}$$

where the low-rank component $\boldsymbol{L}_l$ is expected to capture globally correlated and shared semantic structures, while the sparse component $\boldsymbol{S}_l$ preserves localized and highly discriminative responses. This decomposition provides a structured view of intermediate representations by separating global semantic regularities from local task-sensitive activations.

**Motivation for Low-Rank and Sparse Feature Modeling.** Unlike nuclear-norm or $\ell_1$-norm relaxations commonly used in robust PCA, we explicitly impose rank and sparsity constraints in the latent feature space. This allows direct control over the allocation between global semantic structures and localized discriminative responses. Such controllability is particularly important for adversarial optimization, where the decomposition is repeatedly performed during iterative gradient updates. *(i) Low-Rank Component:* Deep network features often exhibit strong channel-wise correlations induced by convolutions or self-attention, forming dominant subspaces that encode global semantics (Denil et al., 2013; Denton et al., 2014). *(ii) Sparse Component:* Non-linear operations, such as ReLU, attention gating, can produce localized high-magnitude activations that are naturally sparse and capture fragile yet discriminative patterns (Shazeer et al., 2017; Voita et al., 2019).

**Key Takeaway.** Separating $\boldsymbol{L}_l$ and $\boldsymbol{S}_l$ exposes structurally distinct yet complementary subspaces that can be independently exploited during gradient propagation. In contrast, existing SVD-based approximations often entangle these variations, which may limit attack transferability (Wright et al., 2008; Zhou & Tao, 2011). Different from low-rank and sparse modeling in LLMs, which mainly focuses on weight compression or pruning, our LRS decomposition is not intended for exact reconstruction or compression, but to reveal complementary structural components for generating diverse and highly transferable adversarial perturbations.

# B. Channel-Wise Rescaling and Numerical Stability

**Channel-Wise Scale Imbalance.** Deep intermediate features often exhibit significant channel-wise scale imbalance, where a small subset of channels dominates the overall representation magnitude. Directly applying structured matrix factorization in such an unbalanced feature space leads to an ill-conditioned optimization problem: high-magnitude channels disproportionately influence the estimated low-rank subspace, while informative but low-magnitude channels may be absorbed into the residual components, resulting in unstable or even degenerate decompositions.

**Preconditioning for Stable Decomposition.** To mitigate this issue, we incorporate a *channel-wise rescaling* strategy before low-rank and sparse decomposition, following the conditioning principle introduced in (Sun et al., 2024). Specifically, given the feature matrix $\boldsymbol{H}_l \in \mathbb{R}^{C_l \times (H_l W_l)}$, we normalize each channel by its accumulated spatial magnitudes:

$$\boldsymbol{D}_l = \left(\mathrm{diag}(\boldsymbol{H}_l \boldsymbol{H}_l^\top + \epsilon \boldsymbol{I})\right)^{-1/2}, \tag{17}$$

where $\boldsymbol{I}$ is the identity matrix and $\epsilon$ is a small positive constant used for numerical stability. The rescaled feature $\boldsymbol{D}_l \boldsymbol{H}_l$ balances channel magnitudes and yields a better-conditioned approximation objective. From an optimization perspective, this operation can be interpreted as applying a diagonal preconditioner that reduces anisotropy in the Frobenius-norm objective, thereby stabilizing the alternating updates for estimating the low-rank and sparse components and preventing the decomposition from collapsing toward a few dominant high-magnitude channels.

**Role of Rescaling in Stable Low-Rank and Sparse (LRS) Decomposition.** Importantly, channel-wise rescaling neither introduce new structural assumptions nor acts as an adversarial mechanism by itself. Rather, it serves as a necessary preconditioning step that enables a meaningful and well-conditioned realization of the proposed LRS decomposition. Without rescaling, the decomposition tends to overfit dominant channels, weakening the interpretability and effectiveness of the resulting decomposition. In contrast, with channel-wise rescaling, the rescaled formulation in Eq. 3 more reliably separates globally shared semantic structure from localized discriminative variations. Consequently, the improved adversarial performance does not stem from the rescaling operation itself, but from its role in enabling a stable and reliable implementation of the proposed LRS decomposition.

## C. Efficient Approximation of Low-Rank Components via WALA

### C.1. Low-Rank Subspace Estimation

**Regularized Subspace Iteration.** We provide additional derivation and interpretation of the update rules used in Algorithm 1. The procedure operates at the *subspace level* and serves as an *inner-loop solver* for the low-rank update within the outer-loop low-rank and sparse decomposition. For notational simplicity, we first define the rescaled residual feature matrix at $t$-th inner-loop iteration as:

$$\boldsymbol{A} = \boldsymbol{D}_l \boldsymbol{H}_l - \boldsymbol{S}_l^{(t)}, \tag{18}$$

and focus on estimating its rank-$r$ approximation. Building on this, we then consider the following regularized low-rank approximation objective:

$$\min_{\boldsymbol{U}, \boldsymbol{V}} \mathcal{L}(\boldsymbol{U}, \boldsymbol{V}) = \|\boldsymbol{A} - \boldsymbol{U}\boldsymbol{V}^\top\|_F^2 + \lambda \left( \|\boldsymbol{U}\|_F^2 + \|\boldsymbol{V}\|_F^2 \right), \tag{19}$$

where $\boldsymbol{U} \in \mathbb{R}^{C_l \times r}$ and $\boldsymbol{V} \in \mathbb{R}^{(H_l W_l) \times r}$ denote the low-rank factors, and $\lambda$ controls the Frobenius-norm regularization strength. Consequently, the gradient of Eq. 19 with respect to $\boldsymbol{U}$ is given by

$$\nabla_{\boldsymbol{U}} \mathcal{L}(\boldsymbol{U}, \boldsymbol{V}) = -2\boldsymbol{A}\boldsymbol{V} + 2\boldsymbol{U}\boldsymbol{V}^\top \boldsymbol{V} + 2\lambda \boldsymbol{U}. \tag{20}$$

In WALA, $\boldsymbol{U}$ and $\boldsymbol{V}$ are orthonormalized at each iteration via QR projection. Under this orthogonality-preserving update, $\boldsymbol{V}^\top \boldsymbol{V} \approx \boldsymbol{I}$ holds approximately, which simplifies gradient to

$$\nabla_{\boldsymbol{U}} \mathcal{L}(\boldsymbol{U}, \boldsymbol{V}) \approx -2\boldsymbol{A}\boldsymbol{V} + 2(1 + \lambda)\boldsymbol{U}. \tag{21}$$

Taking a gradient descent step with step size $\eta = \frac{1}{2}$ leads to the update

$$\boldsymbol{U}^{(t+1)} \approx \boldsymbol{U}^{(t)} - \eta \cdot \nabla_{\boldsymbol{U}} \mathcal{L}(\boldsymbol{U}, \boldsymbol{V})|_{\boldsymbol{U}=\boldsymbol{U}^{(t)}, \boldsymbol{V}=\boldsymbol{V}^{(t)}} = \boldsymbol{A}\boldsymbol{V}^{(t)} - \lambda \boldsymbol{U}^{(t)} = (\boldsymbol{D}_l \boldsymbol{H}_l - \boldsymbol{S}_l^{(t)})\boldsymbol{V}^{(t)} - \lambda \boldsymbol{U}^{(t)}, \tag{22}$$

followed by QR projection to preserve orthogonality:

$$\boldsymbol{U}^{(t+1)} \leftarrow \text{QR}(\boldsymbol{U}^{(t+1)}). \tag{23}$$

This yields the update rule in Eq. 8. An analogous derivation with respect to $\boldsymbol{V}$ gives the update rule in Eq. 9. When $\lambda = 0$, the updates reduce to a projected *power iteration* that tracks the dominant rank-$r$ subspace of $\boldsymbol{A}$. The regularization term further introduces a damping effect, stabilizing subspace updates across adversarial iterations and preventing numerical drift.

**Warm-Started Strategy.** Across consecutive adversarial iterations, the perturbation is updated gradually, so the intermediate feature matrix $\boldsymbol{A}$ changes only mildly. Thus, the previously estimated subspaces provide informative initialization for the current iteration. At each iteration $t$, WALA initializes $\boldsymbol{U}$ and $\boldsymbol{V}$ with the factors from the previous attack iteration (*i.e.*, $\boldsymbol{U}^{(t-1)}, \boldsymbol{V}^{(t-1)}$), allowing the low-rank subspace to be refined with fewer inner updates. This enables efficient tracking of slowly varying feature subspaces without repeatedly computing exact truncated SVDs.

### C.2. Singular-Value Matrix Estimation

The subspace iteration described above focuses on estimating the dominant rank-$r$ left and right subspaces of $\boldsymbol{A}$. After convergence, we further estimate the corresponding coefficient matrix to complete the low-rank reconstruction. Given the orthonormal subspaces $\boldsymbol{U}_l$ and $\boldsymbol{V}_l$, the remaining problem reduces to estimating the coefficient matrix $\boldsymbol{\Sigma}_l$ that minimizes the reconstruction error:

$$\min_{\boldsymbol{\Sigma}_l} \|\boldsymbol{A} - \boldsymbol{U}_l \boldsymbol{\Sigma}_l \boldsymbol{V}_l^\top\|_F^2. \tag{24}$$

Using the identity $\|\boldsymbol{M}\|_F^2 = \text{Tr}(\boldsymbol{M}^\top \boldsymbol{M})$, the objective in Eq. 24 can be rewritten as

$$\begin{aligned}
\|\boldsymbol{A} - \boldsymbol{U}_l \boldsymbol{\Sigma}_l \boldsymbol{V}_l^\top\|_F^2 &= \text{Tr}\left((\boldsymbol{A} - \boldsymbol{U}_l \boldsymbol{\Sigma}_l \boldsymbol{V}_l^\top)^\top (\boldsymbol{A} - \boldsymbol{U}_l \boldsymbol{\Sigma}_l \boldsymbol{V}_l^\top)\right) \\
&= \text{Tr}(\boldsymbol{A}^\top \boldsymbol{A}) - 2\,\text{Tr}(\boldsymbol{\Sigma}_l^\top \boldsymbol{U}_l^\top \boldsymbol{A}\boldsymbol{V}_l) + \text{Tr}(\boldsymbol{\Sigma}_l^\top \boldsymbol{\Sigma}_l),
\end{aligned} \tag{25}$$

where the last equality follows from the cyclic invariance of the trace and the orthonormality conditions $\boldsymbol{U}_l^\top \boldsymbol{U}_l = \boldsymbol{V}_l^\top \boldsymbol{V}_l = \boldsymbol{I}$.

Taking the derivative with respect to $\boldsymbol{\Sigma}_l$ and setting it to zero yields

$$\frac{\partial}{\partial \boldsymbol{\Sigma}_l} \left( \text{Tr}(\boldsymbol{\Sigma}_l^\top \boldsymbol{\Sigma}_l) - 2 \text{Tr}(\boldsymbol{\Sigma}_l^\top \boldsymbol{U}_l^\top \boldsymbol{A} \boldsymbol{V}_l) \right) = 2\boldsymbol{\Sigma}_l - 2\boldsymbol{U}_l^\top \boldsymbol{A} \boldsymbol{V}_l = 0, \tag{26}$$

which leads to the closed-form solution

$$\boldsymbol{\Sigma}_l = \boldsymbol{U}_l^\top \boldsymbol{A} \boldsymbol{V}_l = \boldsymbol{U}_l^\top (\boldsymbol{D}_l \boldsymbol{H}_l - \boldsymbol{S}_l) \boldsymbol{V}_l. \tag{27}$$

This subspace projection provides the optimal low-rank reconstruction within the estimated subspaces. Importantly, $\boldsymbol{\Sigma}_l$ is computed only once after subspace estimation and therefore does not affect the convergence of the alternating subspace iteration.

## D. Derivation and Convergence Analysis of LRS-Attack

In this section, we provide a rigorous derivation of the proposed Low-Rank and Sparse (LRS) attack approach and analyze the convergence properties of its alternating optimization algorithm. Our analysis builds upon the theoretical framework of GoDec (Zhou & Tao, 2011), while extending it to the rescaled feature-space decomposition used in our attack formulation.

### D.1. Derivation of Alternating Updates for Low-Rank and Sparse Components

We revisit the rescaled low-rank and sparse decomposition objective:

$$\min_{\boldsymbol{L}_l, \boldsymbol{S}_l} \|\boldsymbol{D}_l \boldsymbol{H}_l - \boldsymbol{L}_l - \boldsymbol{S}_l\|_F^2, \quad \text{s.t. } \text{rank}(\boldsymbol{L}_l) \leq r, \ \|\boldsymbol{S}_l\|_0 \leq k. \tag{28}$$

Although the joint optimization is non-convex due to the rank and sparsity constraints, it can be efficiently solved via alternating minimization by updating one component while fixing the other.

**Sparse Component Update.** Fixing the low-rank component $\boldsymbol{L}_l^{(t-1)}$, the optimization with respect to the sparse component reduces to

$$\boldsymbol{S}_l^{(t)} = \underset{\|\boldsymbol{S}_l\|_0 \leq k}{\arg\min} \ \|(\boldsymbol{D}_l \boldsymbol{H}_l - \boldsymbol{L}_l^{(t-1)}) - \boldsymbol{S}_l\|_F^2. \tag{29}$$

This subproblem has a closed-form solution given by the hard-thresholding operator:

$$\boldsymbol{S}_l^{(t)} = \mathcal{H}_k(\boldsymbol{D}_l \boldsymbol{H}_l - \boldsymbol{L}_l^{(t-1)}), \tag{30}$$

where $\mathcal{H}_k(\cdot)$ preserves the $k$ largest-magnitude entries and sets all others to zero. As a result, this update gives the globally optimal solution for the sparse subproblem under the $\ell_0$ constraint.

**Low-Rank Component Update.** Fixing the sparse component $\boldsymbol{S}_l^{(t)}$, the optimization with respect to the low-rank component reduces to

$$\boldsymbol{L}_l^{(t)} = \underset{\text{rank}(\boldsymbol{L}_l) \leq r}{\arg\min} \ \|(\boldsymbol{D}_l \boldsymbol{H}_l - \boldsymbol{S}_l^{(t)}) - \boldsymbol{L}_l\|_F^2. \tag{31}$$

The optimal solution is given by the rank-$r$ truncated SVD of $(\boldsymbol{D}_l \boldsymbol{H}_l - \boldsymbol{S}_l^{(t)})$. However, in practice, computing truncated SVD at each iteration is computationally expensive. Therefore, we adopt the proposed Warm-started Alternating Low-rank Approximation (WALA) algorithm to efficiently approximate the dominant low-rank structure without explicit SVD computation. Thus, this replaces the exact low-rank projection with an efficient approximation, substantially reducing computational overhead while preserving the intended low-rank update objective.

### D.2. Convergence Analysis of Alternating Optimization

We now analyze the convergence properties of the alternating low-rank and sparse updates. Let the decomposition objective be defined as

$$J(\boldsymbol{L}_l, \boldsymbol{S}_l) = \|\boldsymbol{D}_l \boldsymbol{H}_l - \boldsymbol{L}_l - \boldsymbol{S}_l\|_F^2. \tag{32}$$

At iteration $t$, the sparse component $S_l^{(t)}$ is obtained by minimizing $J(L_l^{(t-1)}, S_l)$ with $L_l^{(t-1)}$ fixed. Since the hard-thresholding operator gives the globally optimal solution to the sparse subproblem under the $\ell_0$ constraint, we have

$$J(L_l^{(t-1)}, S_l^{(t)}) \leq J(L_l^{(t-1)}, S_l^{(t-1)}). \tag{33}$$

Similarly, with $S_l^{(t)}$ fixed, the low-rank component $L_l^{(t)}$ is obtained by minimizing $J(L_l, S_l^{(t)})$. The proposed WALA algorithm approximates this rank-constrained update through regularized subspace iteration. When the inner WALA updates converge or provide a sufficiently accurate descent step, the objective does not increase after the low-rank update. Therefore, the following inequality holds:

$$J(L_l^{(t)}, S_l^{(t)}) \leq J(L_l^{(t-1)}, S_l^{(t)}). \tag{34}$$

Combining the above inequalities gives the monotonic descent relation:

$$J(L_l^{(t-1)}, S_l^{(t-1)}) \geq J(L_l^{(t-1)}, S_l^{(t)}) \geq J(L_l^{(t)}, S_l^{(t)}). \tag{35}$$

Therefore, over $T$ iterations, the objective value satisfies

$$J(L_l^{(1)}, S_l^{(1)}) \geq J(L_l^{(2)}, S_l^{(2)}) \geq \cdots \geq J(L_l^{(T)}, S_l^{(T)}). \tag{36}$$

Since $J(L_l, S_l)$ is a non-negative squared reconstruction error, this monotonically decreasing objective sequence is lower-bounded by zero and thus converges to a local minimum.

## E. Rationale for Hierarchical Low-Rank and Sparse Experts

This section provides theoretical and empirical motivation for adopting multi-layer low-rank and sparse decomposition experts. The rationale is grounded in the hierarchical nature of deep representations, layer-wise feature redundancy and sparsity, and their connection to adversarial transferability.

**Depth-Dependent Feature Structures.** Deep networks learn hierarchical representations whose properties vary with depth. Shallow layers encode localized patterns such as edges and textures (Zeiler & Fergus, 2014; Mehta et al., 2019), often exhibiting activation sparsity and high spatial variability (Glorot et al., 2011). Intermediate layers combine local discriminative responses with emerging global structures (Yosinski et al., 2014; Bahri et al., 2017). Deep layers capture abstract semantic concepts (Cheng et al., 2015; Wang et al., 2019) and concentrate information into a low-dimensional subspace due to strong channel correlations and over-parameterization (Gunasekar et al., 2017). This hierarchy implies distinct statistical and geometric characteristics at different network depths.

**Hierarchical Expert Specialization.** Dense expert attacks or single-layer attacks often generate gradients biased toward limited feature regimes, which restricts cross-architecture generalization. A straightforward extension is to apply the same decomposition strategy across multiple layers. However, this still implicitly assumes homogeneous feature structures and overlooks the hierarchical nature of deep representations. To address this limitation, we introduce depth-aware decomposition experts aligned with different network stages: (i) a sparse-dominant expert for shallow layers to capture high-frequency localized presentations (Ilyas et al., 2019); (ii) a balanced low-rank plus sparse expert for intermediate layers to model both local and global structures; (iii) a low-rank-dominant expert for deep layers to emphasize architecture-invariant semantic pattern (Sanyal et al., 2018). These experts are structured transformations applied to a shared backbone rather than independent models, enabling systematic exploration of depth-dependent vulnerabilities. On the other hand, each depth-aware expert defines a distinct perturbation pathway and emphasizes different structural cues in the feature space. Aggregating these heterogeneous gradients enriches the optimization direction and reduces overfitting to a single feature regime of the surrogate model, thereby improving black-box transferability.

**Empirical Evidence.** Single-layer decomposition shows strong layer sensitivity, where no single layer consistently maximizes ASR across target models (Fig. 6). In contrast, multi-layer decomposition that combines shallow, intermediate, and deep layers consistently achieves higher and more stable ASR (Fig. 7), demonstrating the benefit of aggregating features from multiple semantic levels. These results suggest that multi-layer decomposition experts effectively leverage hierarchical feature structures and expose complementary vulnerabilities that are difficult to capture with a single expert, thereby improving adversarial transferability.

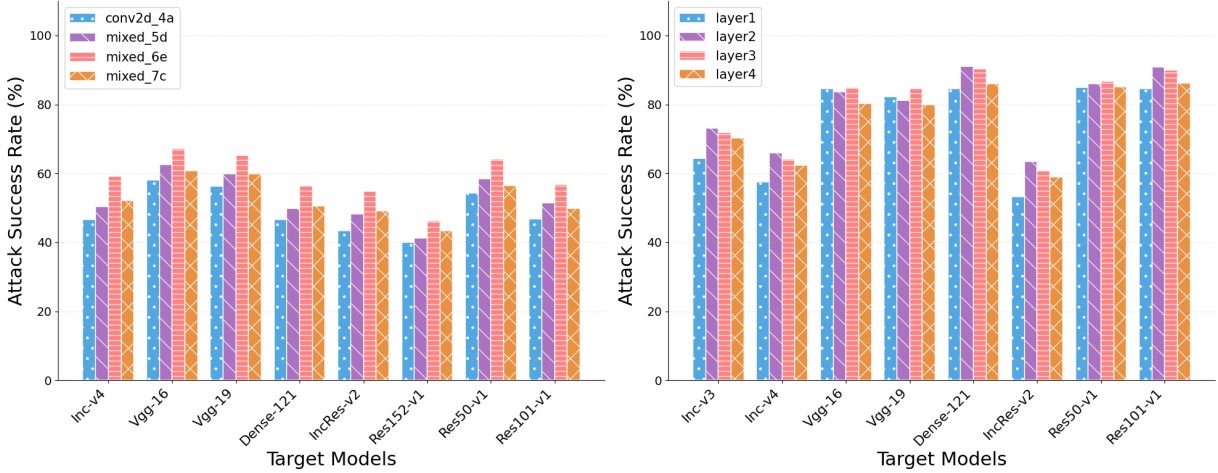

*Figure 6.* Effect of single-layer selection on black-box ASR. Results are reported using Inc-v3 and ResNet-152-v1 as surrogate models and evaluated on multiple target architectures.

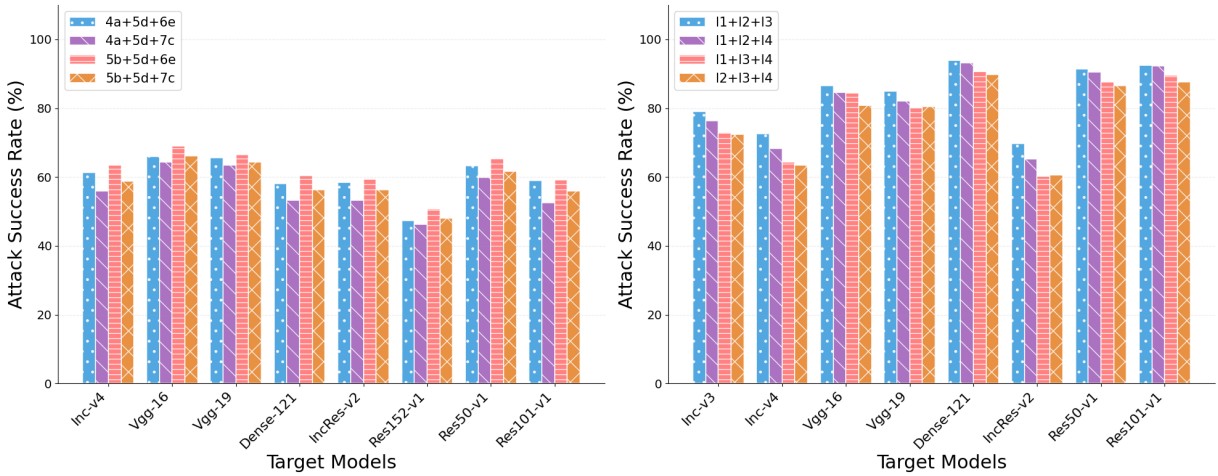

*Figure 7.* Effect of multi-layer selection on black-box ASR. Results are reported using Inc-v3 and ResNet-152-v1 as surrogate models and evaluated across multiple target architectures. Shallow, intermediate, and deep layers are jointly decomposed (*e.g.*, $4a + 5d + 6e$ for Inc-v3 and $l_1 + l_2 + l_3$ for ResNet-152-v1).

## F. Additional Experiments

In this section, we conduct four sets of experiments to comprehensively evaluate LRS-Attack. First, we provide qualitative visualizations of iterative attacks over 10 steps, illustrating the evolution of intermediate adversarial examples and prediction confidence. This analysis offers insights into the attack dynamics and highlights the stability and early misclassification behavior induced by LRS-Attack. Second, we analyze the hyperparameter ablation strategies on Vision Transformer (ViT) surrogate models to examine whether the parameter selection trends observed on CNNs generalize to transformer-based architectures. Third, we integrate our approach with several transformation-based and feature-based attacks to assess its generality beyond standard gradient-based baselines. Finally, we introduce additional CNN surrogate models, including Inc-v4 (Christian Szegedy & Alemi, 2017), Res152-v1 (He et al., 2016a), VGG-16 (Simonyan & Zisserman, 2015), and IncRes-v2 (He et al., 2016b), enabling a more comprehensive quantitative comparison with existing methods.

### F.1. Qualitative Visualization of Iterative Attacks

To further examine the effectiveness of the proposed LRS attack method, we visualize the evolution of adversarial examples generated by an iterative attack process over 10 iterations under three settings: (i) without decomposition, (ii) SVD-based decomposition, and (iii) the proposed LRS decomposition. All attacks are conducted within the MI-FGSM framework.

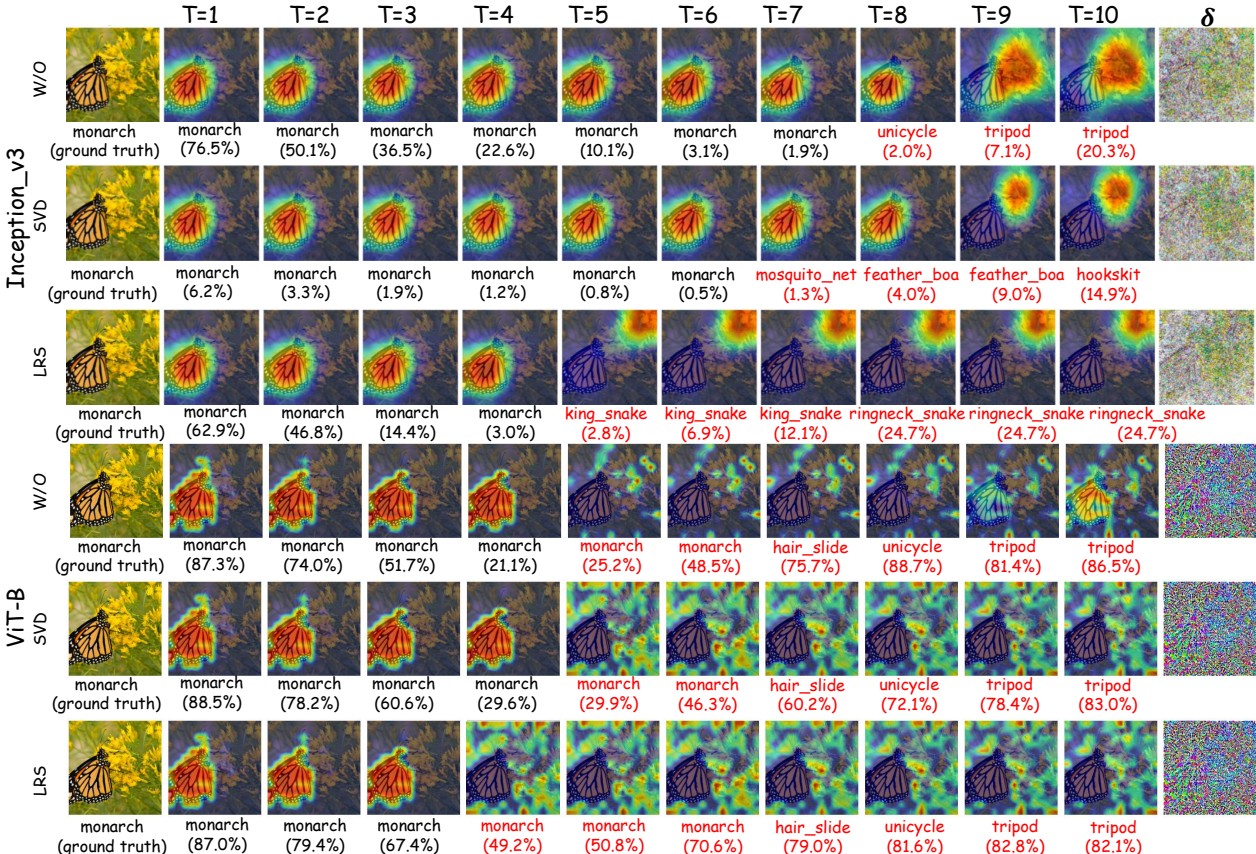

*Figure 8.* Qualitative comparison of iterative adversarial attacks across CNN and ViT surrogate models. We show adversarial evolution over 10 iterations using Inception-v3 and ViT-B under three settings: without decomposition (W/O), SVD-based decomposition, and the proposed low-rank and sparse decomposition (LRS). Predicted classes and softmax confidences are annotated, with misclassifications highlighted in red. The rightmost column ($\delta$) visualizes the magnified final perturbation for visualization.

We use Inception-v3 as a representative CNN surrogate and ViT-B as a representative ViT surrogate to assess the cross-architecture behavior of different decomposition strategies.

As shown in Fig. 8, LRS consistently requires fewer iterations to induce misclassification across both surrogate models compared with the baseline without decomposition (W/O) and the SVD-based variant. For instance, under the ViT-B surrogate, LRS successfully triggers misclassification within $T = 4$ iterations, whereas the baseline methods require more iterations to reach comparable confidence on an incorrect class. The overlaid attention heatmaps further reveal that LRS more effectively redirects the model's focus away from the salient regions of the original class (*e.g.,monarch butterfly*). At later iterations, LRS produces more dispersed or background-oriented activation patterns, whereas baseline methods tend to retain concentrated attention on class-discriminative regions. Despite its stronger attack performance, the final perturbations ($\delta$) generated by LRS do not introduce visible structured artifacts or excessive noise. The perturbations remain high-frequency and visually imperceptible, satisfying the $L_\infty$ constraint while achieving improved transferability.

### F.2. Additional Hyperparameter Analysis on Vision Transformers

In the main paper, we analyze the impact of different hyperparameter selection strategies on CNN surrogate models. To further examine whether the observed trends generalize to vision transformer-based architectures, we additionally conduct the same hyperparameter ablation on ViT models.

Fig. 9 reports the impact of different hyperparameter selection strategies on the attack success rate when using ViT models as surrogates. For a fair comparison, we adopt the same five strategies ($S_0$–$S_4$) defined in the main paper. Consistent with the results on CNN-based models, the $S_0$ strategy, which gradually decreases the sparsity while increasing the rank, consistently achieves the highest ASR across different target architectures.

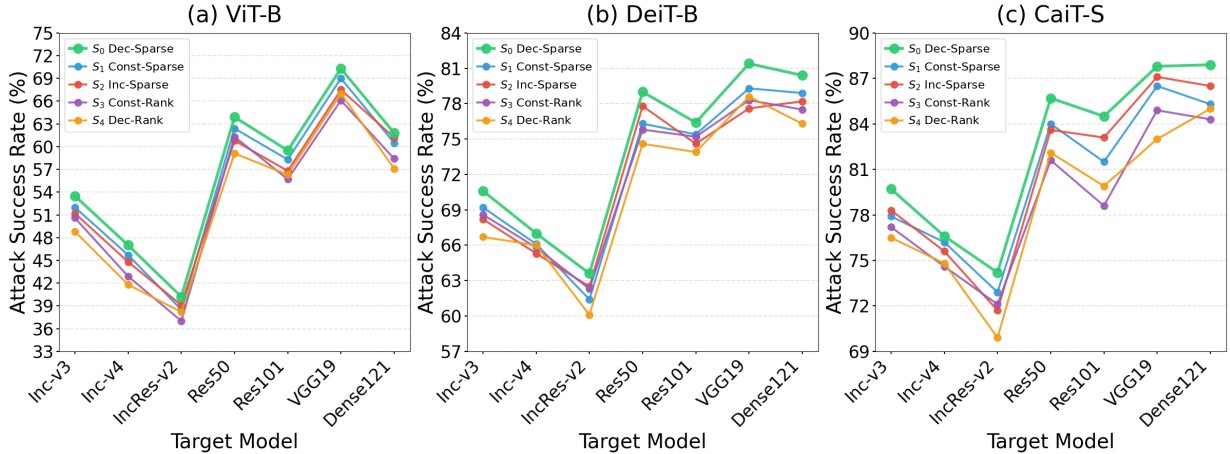

*Figure 9.* Impact of different hyperparameter strategies on ASR for ViT surrogate models.

### F.3. Combination with Different Types of Attack Methods

To verify the generalizability and compatibility of the proposed LRS method, we integrate it with different types of attack methods, including optimization-based methods (PGN (Ge et al., 2023a), GGS (Niu et al., 2025)), input transformations-based methods (SIA (Wang et al., 2023), BSR (Wang et al., 2024b)), and feature-based methods (BFA (Wang et al., 2024c), P2FA (Liu et al., 2025)). We compare the original attacks (W/O) and the SVD-based decomposition with our proposed LRS-based decomposition across diverse evaluation scenarios.

As summarized in Table 5, LRS consistently improves the ASR across various CNN target models. Although some strong baselines, such as GGS and BFA, already approach saturation on certain models with ASRs close to 100%, LRS still maintains or slightly further improves their performance. More importantly, in more challenging cross-architecture transfer tasks, such as transferring from Inc-v3 to Res152, LRS provides substantial gain over the SVD baseline. These results suggests that LRS can better isolates transferable adversarial components that are generalize across different convolutional architectures.

In addition to standard CNN targets, we further evaluate the robustness of the proposed LRS method against defended CNN models, including Inc-v3$_{adv}$, Inc-v3$_{ens3}$, Inc-v3$_{ens4}$, and IncRes-v2$_{ens}$, as well as widely adopted input preprocessing defenses, such as HGD, JPEG compression, random resizing and padding (R&P), and the NIPS-r3 defense. Table 6 reports the detailed attack performance under these defended settings, where LRS-Attack achieves the best ASR in most cases.

To further examine the cross-architecture transferability, we extend the evaluation to a broad range of Vision Transformer (ViT) models with diverse architectural designs. Specifically, we consider standard ViT models, including PiT (Heo et al., 2021), CaiT (Touvron et al., 2021b), DeiT (Touvron et al., 2021a), Swin (Liu et al., 2021a), Visformer (Chen et al., 2021), ConViT (d'Ascoli et al., 2021), Twins (Chu et al., 2021), and XCiT, as well as three defended ViTs, including DeiT-S (adv) (Bai et al., 2021), Swin-B (adv) (Mo et al., 2022), XCiT-S (adv) (Debenedetti et al., 2023). Table 7 reports ASRs on these ViT models. Similar to the observations on CNN-based targets, incorporating LRS consistently improves the transferability of different types of attack methods across most ViT target models and defended ViT target models.

### F.4. Comparison of Attack Transferability Using Additional CNN Surrogate Models

To further examine the generality of our LRS-Attack, we extend the evaluation to additional CNN surrogate models. Table 8 reports the ASR of different attack methods on various CNN target models using Inc-v4, Res152-v1, Vgg-16, and IncRes-v2 as additional surrogates. Table 9 further presents the corresponding ASR of the same attack methods under various defended CNN target models and defense methods.

Across all surrogate settings and attack variants, the proposed LRS attacks consistently outperform both the original attacks without decomposition (W/O) and their SVD-based counterparts. Specifically, LRS achieves the highest ASR on CNN models for almost all attack frameworks, including MI-FGSM (Dong et al., 2018), DI-FGSM (Xie et al., 2019), TI-FGSM (Dong et al., 2019), PI-FGSM (Gao et al., 2020), SI-NI-FGSM (Lin et al., 2020), VT-MI-FGSM (Wang & He,

2021) and GI-FGSM (Wang et al., 2024a).

Beyond CNN target models, we further evaluate attack transferability on a diverse set of Vision Transformer (ViT) models. As shown in Table 10, LRS-based attacks consistently achieve higher ASRs than competing methods across almost all evaluated ViT architectures.

## G. Complete Per-Model Experimental Results

In this section, we present the complete per-model experimental results corresponding to the main paper. While the main text reports averaged performance for clarity, this section presents a detailed breakdown of ASRs across different target architectures and defense settings.

### G.1. CNN-Based Surrogate Models

Tables 11-13 provide the detailed per-model results corresponding to Table 1 in the main paper. Specifically, Table 11 reports the complete ASRs of different attacks evaluated on various CNN target models, Table 12 presents the complete results on various defended CNN target models and defense methods, and Table 13 reports the complete results on various ViT target models and defended ViT target models. The full results are presented below.

### G.2. ViT-Based Surrogate Models

Tables 14-16 provide the detailed per-model results corresponding to Table 2 in the main paper. Specifically, Table 14 reports the complete ASRs of different ViT-based attack methods evaluated on various CNN target models, Table 15 presents the complete results on various defended CNN target models and four defense strategies, and Table 16 reports the complete results on various ViT target models and defended ViT target models. The full results are shown below.

*Table 5.* ASR (%) on various CNN target models for original (W/O), SVD, and LRS approaches combined with different types of attacks, including optimization-based, input transformation-based, and feature-based methods. Best results are highlighted in bold.

| Model | Attack | Method | Inc-v3 | Inc-v4 | Vgg-16 | Vgg-19 | Dense-121 | IncRes-v2 | Res152 | Res50 | Res101 |
|---|---|---|---|---|---|---|---|---|---|---|---|
| Inc-v3 | SIA | W/O | 99.6 | 88.8 | 88.4 | 89.4 | 85.1 | 86.8 | 78.8 | 86.1 | 81.3 |
| | | SVD | 99.7 | 92.7 | 92.7 | 92.7 | 89.0 | 91.0 | 84.0 | 90.0 | 87.2 |
| | | LRS | **100.0** | **93.3** | **93.3** | **93.8** | **90.3** | **91.9** | **85.2** | **91.0** | **89.5** |
| | BSR | W/O | 99.6 | 92.7 | 91.8 | 92.8 | 90.7 | 89.4 | 81.0 | 88.1 | 84.9 |
| | | SVD | 99.8 | 94.2 | 95.2 | 95.0 | 92.3 | 91.3 | **87.6** | 90.9 | **89.5** |
| | | LRS | **100.0** | **95.5** | **96.0** | **95.9** | **93.2** | **92.5** | 86.8 | **92.2** | 89.3 |
| | PGN | W/O | **100.0** | 90.6 | 87.9 | 88.6 | 87.9 | 90.1 | 81.9 | 85.2 | 80.9 |
| | | SVD | 99.8 | 91.4 | 90.3 | 91.0 | **90.1** | 90.5 | 85.3 | 88.6 | **86.0** |
| | | LRS | 99.9 | **93.2** | **90.6** | **91.5** | 89.3 | **91.4** | **87.0** | **90.7** | 85.8 |
| | GGS | W/O | **100.0** | 96.4 | 91.9 | 92.1 | 91.3 | 95.7 | 87.7 | 90.2 | 89.3 |
| | | SVD | **100.0** | 97.0 | **95.1** | 93.8 | 92.6 | 95.8 | 90.6 | 93.0 | 92.1 |
| | | LRS | **100.0** | **97.5** | 95.0 | **94.9** | **93.4** | **97.3** | **91.3** | **93.6** | **92.6** |
| | BFA | W/O | **100.0** | 96.6 | 94.9 | 95.5 | 92.9 | 95.4 | 90.7 | 94.0 | 92.1 |
| | | SVD | 99.8 | 96.6 | 95.1 | 95.0 | 93.5 | 96.0 | **92.0** | 94.2 | 93.2 |
| | | LRS | **100.0** | **97.4** | **95.5** | **96.0** | **94.3** | **96.6** | 91.7 | **94.7** | **93.9** |
| | P2FA | W/O | **100.0** | 97.0 | **95.5** | 95.1 | **93.1** | 96.1 | 87.1 | 96.5 | 93.2 |
| | | SVD | **100.0** | 96.6 | 96.4 | 95.5 | 92.0 | 95.7 | 85.3 | 96.6 | 93.5 |
| | | LRS | **100.0** | **97.3** | **95.5** | **95.8** | 92.0 | **96.6** | **87.6** | **97.0** | **94.3** |
| Vgg-19 | SIA | W/O | 87.4 | 90.8 | **99.9** | **99.9** | 94.7 | 81.9 | 86.6 | 97.0 | 94.2 |
| | | SVD | 89.3 | 92.6 | 99.8 | 99.8 | 95.2 | 83.1 | 85.8 | 97.3 | 95.0 |
| | | LRS | **89.9** | **94.1** | **99.9** | **99.9** | **95.8** | **84.6** | **87.6** | **98.0** | **95.7** |
| | BSR | W/O | 90.3 | 93.7 | 99.8 | **100.0** | 95.1 | 85.3 | 85.7 | 97.6 | 95.2 |
| | | SVD | 91.0 | **94.8** | 99.8 | **100.0** | 95.5 | **87.6** | 88.1 | 98.0 | 96.0 |
| | | LRS | **92.3** | **94.8** | **99.9** | **100.0** | **96.7** | 87.4 | **88.6** | **98.6** | **96.2** |
| | PGN | W/O | 84.6 | 84.0 | **100.0** | **100.0** | 91.8 | 73.5 | 82.8 | 92.3 | 88.4 |
| | | SVD | 85.5 | 87.5 | 99.9 | **100.0** | 92.9 | **76.6** | 86.0 | 93.3 | 91.0 |
| | | LRS | **86.1** | **88.4** | 99.8 | **100.0** | **93.5** | 76.4 | **86.6** | **94.6** | **91.7** |
| | GGS | W/O | 92.1 | 93.4 | 99.7 | **100.0** | 96.4 | 82.9 | 89.8 | 97.7 | 94.4 |
| | | SVD | 92.4 | **95.5** | 99.7 | 99.9 | 97.0 | 84.1 | 90.1 | 98.6 | 96.1 |
| | | LRS | **92.9** | 95.2 | **99.8** | **100.0** | **97.6** | **86.3** | **91.8** | **98.9** | **96.4** |
| | BFA | W/O | 91.2 | 94.5 | 99.8 | **100.0** | 96.5 | 87.6 | 90.7 | **98.2** | 96.4 |
| | | SVD | 92.3 | **95.4** | 99.7 | **100.0** | 96.4 | 86.9 | 92.4 | 98.0 | 96.0 |
| | | LRS | **92.6** | 95.3 | **99.9** | **100.0** | **96.6** | **88.3** | **92.5** | **98.2** | 96.3 |
| | P2FA | W/O | 84.9 | 90.5 | **99.9** | **100.0** | **94.5** | 78.8 | 86.6 | **97.9** | 96.1 |
| | | SVD | 63.8 | 68.8 | 94.4 | 96.0 | 72.8 | 58.9 | 68.6 | 85.6 | 81.5 |
| | | LRS | **85.0** | **90.9** | **99.9** | **100.0** | 94.0 | **78.9** | **87.6** | **97.9** | **96.5** |
| Densenet-121 | SIA | W/O | 93.3 | 91.3 | 95.5 | 95.9 | **100.0** | 89.4 | 96.8 | 96.9 | 96.4 |
| | | SVD | 91.6 | 92.1 | 97.5 | 97.6 | **100.0** | 87.3 | 95.9 | 98.3 | 97.1 |
| | | LRS | **94.5** | **94.7** | **98.6** | **98.2** | **100.0** | **90.6** | **97.5** | **99.1** | **98.5** |
| | BSR | W/O | 94.0 | 92.9 | 97.7 | 96.9 | **100.0** | 89.8 | 96.0 | 98.3 | **98.2** |
| | | SVD | 92.5 | 92.0 | 97.5 | 96.6 | **100.0** | 86.8 | 94.9 | 98.1 | 97.0 |
| | | LRS | **95.2** | **94.5** | **98.5** | **98.1** | **100.0** | **92.8** | **97.0** | **99.0** | **98.2** |
| | PGN | W/O | 93.3 | 91.3 | 95.5 | 95.9 | **100.0** | 89.4 | 96.8 | 96.9 | 96.4 |
| | | SVD | 91.4 | 90.9 | 95.2 | 95.9 | **100.0** | 84.2 | 89.8 | **97.8** | 95.8 |
| | | LRS | **94.4** | **92.9** | **96.6** | **96.9** | **100.0** | **91.0** | **97.5** | 97.7 | **97.3** |
| | GGS | W/O | 98.2 | 97.3 | 99.0 | 98.8 | **100.0** | 97.4 | 99.0 | 98.2 | 99.0 |
| | | SVD | 96.8 | 97.2 | 98.7 | 98.8 | **100.0** | 95.8 | 99.0 | 98.6 | 99.2 |
| | | LRS | **98.6** | **97.6** | **99.3** | **99.2** | **100.0** | **97.5** | **99.2** | **99.6** | **99.3** |
| | BFA | W/O | **90.7** | 88.7 | 97.2 | 96.1 | **100.0** | **91.2** | 96.4 | 97.6 | 96.5 |
| | | SVD | 89.7 | 87.3 | 97.5 | 96.1 | **100.0** | 88.0 | 95.6 | 97.2 | 96.2 |
| | | LRS | **90.7** | **89.6** | **97.6** | **96.2** | **100.0** | 91.0 | **96.9** | **97.7** | **96.7** |
| | P2FA | W/O | 94.8 | 95.3 | 99.5 | 98.5 | **100.0** | 91.6 | 99.0 | **99.6** | 98.3 |
| | | SVD | 89.0 | 92.2 | 98.8 | 98.3 | **100.0** | 83.4 | 97.2 | 99.1 | 98.4 |
| | | LRS | **94.8** | **95.6** | **99.7** | **99.1** | **100.0** | **92.6** | **99.4** | 99.5 | **98.9** |

*Table 6.* ASR (%) on various defended CNN target models and defense methods for original (W/O), SVD, and LRS approaches combined with different types of attacks, including optimization-based, input transformation-based, and feature-based methods. Best results are highlighted in bold.

| Model | Attack | Method | Inc-v3$_{adv}$ | Inc-v3$_{ens3}$ | Inc-v3$_{ens4}$ | IncRes-v2$_{ens}$ | HGD | JPEG | R&P | NIPS_r3 |
|---|---|---|---|---|---|---|---|---|---|---|
| Inc-v3 | SIA | W/O | 50.1 | 47.1 | 46.1 | 26.1 | 27.4 | 70.2 | 55.8 | 60.1 |
| | | SVD | 53.8 | 50.8 | 49.1 | 26.2 | 28.6 | 75.6 | 62.1 | 65.4 |
| | | LRS | **54.3** | **51.3** | **50.7** | **26.9** | **29.5** | **77.9** | **62.8** | **65.9** |
| | BSR | W/O | 52.3 | 52.8 | 50.8 | 30.6 | 36.9 | 75.6 | 64.7 | 53.1 |
| | | SVD | 54.5 | 57.6 | 55.0 | 32.9 | 40.0 | 78.9 | 69.1 | 72.2 |
| | | LRS | **56.3** | **59.1** | **57.1** | **34.4** | **41.4** | **80.9** | **71.2** | **74.5** |
| | PGN | W/O | 70.6 | 64.5 | 65.0 | 45.5 | 38.6 | 93.4 | 69.3 | 72.6 |
| | | SVD | 72.8 | 68.1 | 67.9 | **48.0** | **40.2** | 85.8 | 72.3 | **76.6** |
| | | LRS | **73.7** | **68.8** | **68.8** | 47.9 | 39.8 | **94.8** | **74.1** | **76.6** |
| | GGS | W/O | 70.1 | 62.7 | 61.1 | **39.0** | 30.9 | 80.5 | 73.3 | 77.4 |
| | | SVD | 70.6 | **63.0** | 63.5 | 38.9 | 30.0 | 84.0 | 77.2 | 81.0 |
| | | LRS | **70.8** | 62.3 | **63.7** | **39.0** | **31.2** | **86.2** | **78.2** | **81.9** |
| | BFA | W/O | 73.1 | 59.3 | 58.3 | **33.7** | **37.3** | 85.1 | **75.8** | **49.1** |
| | | SVD | 71.9 | 58.0 | 57.9 | 32.8 | 35.0 | 86.2 | 73.5 | 47.0 |
| | | LRS | **73.9** | **61.9** | **58.7** | 33.0 | 36.9 | **86.9** | 74.7 | 48.1 |
| | P2FA | W/O | 44.2 | **27.2** | 26.5 | 14.3 | **5.9** | 81.4 | 59.1 | 64.3 |
| | | SVD | 44.6 | 25.9 | 24.5 | 13.2 | 4.9 | 79.5 | 56.7 | 63.7 |
| | | LRS | **45.7** | 27.0 | **27.1** | **14.5** | 4.8 | **81.9** | **60.0** | **65.2** |
| Vgg-19 | SIA | W/O | 71.0 | 64.7 | 60.3 | 48.6 | 70.6 | 78.5 | 51.5 | 54.0 |
| | | SVD | 73.2 | 65.0 | 60.5 | 49.0 | 69.4 | 79.3 | 53.4 | 57.7 |
| | | LRS | **73.7** | **67.5** | **63.6** | **51.8** | **74.7** | **82.1** | **55.5** | **59.2** |
| | BSR | W/O | 75.5 | 69.5 | 64.4 | 54.8 | 74.3 | 79.4 | 57.8 | 61.6 |
| | | SVD | 77.4 | **73.0** | 67.4 | 55.9 | 75.1 | 82.8 | 60.2 | 63.9 |
| | | LRS | **77.9** | 72.0 | **67.5** | **57.8** | **78.0** | **83.1** | **60.7** | **64.3** |
| | PGN | W/O | 70.3 | 64.4 | 63.8 | 52.1 | 62.4 | 82.2 | 58.4 | 63.7 |
| | | SVD | 71.2 | 66.0 | 65.2 | **53.4** | 64.9 | 84.2 | **62.2** | 65.2 |
| | | LRS | **72.6** | **66.9** | **65.3** | 53.0 | **65.9** | **85.6** | 61.0 | **66.0** |
| | GGS | W/O | 73.9 | 67.4 | 64.6 | 53.9 | 71.1 | 85.3 | 60.1 | 65.2 |
| | | SVD | 75.0 | **69.9** | 66.5 | 52.8 | 73.1 | 87.9 | **61.3** | 65.4 |
| | | LRS | **76.0** | 67.9 | **67.2** | **57.0** | **75.5** | **88.2** | 60.6 | **66.4** |
| | BFA | W/O | 78.1 | **72.4** | 66.6 | 55.1 | 73.1 | 84.6 | 58.9 | 62.3 |
| | | SVD | 78.1 | 71.7 | 66.7 | 55.2 | 75.7 | **85.3** | 58.9 | **62.6** |
| | | LRS | **78.2** | 71.9 | **67.7** | **55.7** | **75.8** | 85.0 | **60.0** | 61.2 |
| | P2FA | W/O | 57.3 | 45.5 | 36.9 | **28.8** | 51.3 | 66.8 | 35.7 | 37.5 |
| | | SVD | 41.3 | 32.4 | 27.1 | 22.6 | 34.5 | 50.6 | 29.1 | 29.0 |
| | | LRS | **58.0** | **46.1** | **37.6** | 28.6 | **53.2** | **67.6** | **36.4** | **38.9** |
| Densenet-121 | SIA | W/O | **89.1** | 87.1 | **87.1** | **79.9** | 88.2 | 93.5 | 75.1 | 80.7 |
| | | SVD | 84.3 | 80.7 | 79.4 | 69.5 | 85.7 | 92.4 | 72.9 | 78.5 |
| | | LRS | **89.1** | **87.8** | 85.2 | 77.1 | **90.6** | **94.7** | **77.9** | **84.7** |
| | BSR | W/O | 89.6 | 87.0 | 86.8 | 76.7 | 90.7 | 75.6 | 80.0 | 84.5 |
| | | SVD | 86.8 | 82.7 | 81.6 | 71.7 | 87.3 | 80.1 | 74.8 | 80.9 |
| | | LRS | **92.1** | **88.7** | **89.3** | **81.5** | **93.3** | **81.2** | **83.5** | **89.2** |
| | PGN | W/O | 89.1 | 87.1 | 87.1 | 79.9 | 89.2 | 76.9 | 83.9 | 83.4 |
| | | SVD | 62.9 | 64.0 | 62.8 | 46.0 | 72.8 | 80.5 | 73.1 | 66.8 |
| | | LRS | **89.9** | **89.2** | **88.5** | **83.8** | **90.6** | **82.1** | **85.4** | **86.5** |
| | GGS | W/O | 95.3 | 93.8 | 92.5 | 86.5 | 94.7 | 97.1 | **89.2** | 90.7 |
| | | SVD | 92.4 | 92.2 | 89.8 | 82.4 | 95.4 | 97.2 | 86.0 | 89.4 |
| | | LRS | **95.4** | **94.0** | **93.0** | **87.7** | **96.5** | **97.3** | 88.6 | **92.2** |
| | BFA | W/O | 77.6 | 72.4 | 68.2 | 56.1 | 75.7 | 89.7 | 35.3 | **64.7** |
| | | SVD | 74.9 | 70.4 | 65.6 | 54.6 | 73.7 | 88.8 | 33.2 | 62.0 |
| | | LRS | **78.3** | **72.6** | **69.5** | **57.4** | **75.9** | **89.9** | **36.8** | **64.7** |
| | P2FA | W/O | 76.1 | 70.2 | **62.1** | 49.7 | 74.1 | 89.8 | 55.0 | 58.9 |
| | | SVD | 73.3 | 62.1 | 55.5 | 46.2 | 67.1 | 84.6 | 49.8 | 52.3 |
| | | LRS | **78.1** | **71.0** | 61.5 | **50.3** | **79.8** | **90.0** | **58.4** | **64.8** |

*Table 7.* ASR (%) on various ViT target models an defended ViT target models for original (W/O), SVD, and LRS approaches combined with different types of attacks, including optimization-based, input transformation-based, and feature-based methods. Best results are highlighted in bold.

| Model | Attack | Method | PiT-S | CaiT-S | DeiT-B | Swin-S | ViT-B | Visformer-S | ConViT-B | Twins-B | DeiT-S (adv) | Swin-B (adv) | XCiT-S (adv) |
|---|---|---|---|---|---|---|---|---|---|---|---|---|---|
| Inc-v3 | SIA | W/O | 53.7 | 41.2 | **35.6** | 45.9 | 39.6 | 55.2 | 34.9 | 51.7 | 29.7 | 19.9 | 61.4 |
| | | SVD | 55.8 | 43.4 | 35.5 | 50.2 | 42.9 | **60.0** | 37.6 | 57.1 | 29.8 | 20.2 | 61.0 |
| | | LRS | **58.6** | **44.0** | 35.3 | **51.2** | **43.0** | 59.6 | **37.9** | **57.7** | **30.9** | **20.5** | **62.4** |
| | BSR | W/O | 58.0 | 45.3 | 37.4 | 49.2 | 41.7 | 60.1 | 37.0 | 55.3 | 30.6 | 20.6 | 62.3 |
| | | SVD | 62.2 | 49.6 | 40.6 | 51.3 | **46.3** | **65.3** | 40.6 | 60.0 | 31.3 | 21.0 | 62.3 |
| | | LRS | **62.4** | **49.9** | **41.1** | **54.2** | 46.1 | 64.8 | **41.1** | **60.3** | **31.4** | **21.5** | **63.2** |
| | PGN | W/O | 62.8 | 53.0 | 46.9 | 54.5 | 48.5 | 63.2 | 45.2 | 60.5 | 34.2 | 23.8 | 64.0 |
| | | SVD | 65.4 | 58.3 | 50.1 | 58.5 | **52.3** | **68.9** | 48.5 | 64.0 | **36.3** | 25.4 | **64.5** |
| | | LRS | **67.8** | **58.4** | **50.4** | **59.8** | **52.3** | 68.0 | **50.4** | **65.6** | **36.3** | **26.6** | **64.5** |
| | GGS | W/O | 66.7 | 56.0 | 48.4 | 61.7 | 47.5 | 70.5 | 46.9 | 66.5 | 33.6 | 22.3 | 62.3 |
| | | SVD | 69.5 | 60.7 | 49.5 | 63.0 | **51.5** | **73.1** | 47.7 | 70.6 | 33.5 | 22.7 | 62.8 |
| | | LRS | **69.6** | **61.2** | **50.2** | **63.2** | 50.3 | 72.7 | **47.8** | **73.1** | **34.1** | **22.8** | **63.0** |
| | BFA | W/O | 65.8 | **58.5** | 44.9 | 62.1 | 50.2 | 76.0 | **43.8** | 72.4 | 30.5 | **21.0** | 61.5 |
| | | SVD | 65.5 | 57.3 | 43.4 | 62.8 | 49.9 | 75.3 | 42.9 | 70.9 | 30.0 | 20.8 | **61.9** |
| | | LRS | **65.9** | 57.2 | **45.0** | **63.1** | **50.5** | **76.5** | **43.8** | **73.1** | **30.8** | **21.0** | **61.9** |
| | P2FA | W/O | 49.7 | **38.0** | **28.9** | 43.7 | **42.8** | 60.7 | 28.8 | 51.7 | 28.3 | 19.5 | 61.3 |
| | | SVD | 48.7 | 36.5 | 27.9 | 42.0 | 41.7 | 59.1 | 27.9 | 50.2 | 29.2 | 19.1 | 60.1 |
| | | LRS | **50.0** | 37.9 | **28.9** | **44.3** | 42.6 | **61.5** | **29.8** | **52.6** | **29.5** | **19.8** | **61.4** |
| Vgg-19 | SIA | W/O | 62.7 | 53.0 | 45.4 | 60.8 | 47.6 | 75.8 | 43.5 | 72.5 | 40.9 | 32.4 | **73.6** |
| | | SVD | 62.8 | 53.2 | 43.4 | 58.9 | **51.1** | 76.9 | 43.1 | 72.0 | 41.1 | 32.5 | 73.5 |
| | | LRS | **65.3** | **56.7** | **47.8** | **62.6** | 50.5 | **78.9** | **46.3** | **74.3** | **41.9** | **32.7** | **73.6** |
| | BSR | W/O | 62.5 | 53.0 | 44.1 | 53.3 | 52.0 | 71.3 | 42.3 | 64.8 | 41.7 | 32.2 | **73.4** |
| | | SVD | 63.3 | 53.6 | 42.7 | 54.7 | **54.1** | 73.5 | **44.2** | 65.4 | 41.8 | 32.7 | 73.3 |
| | | LRS | **65.0** | **55.7** | **45.1** | **57.6** | 52.4 | **76.3** | 42.7 | **67.5** | **41.9** | **33.0** | **73.4** |
| | PGN | W/O | 59.1 | 51.6 | 47.1 | 54.6 | 52.3 | 65.4 | 43.7 | 62.5 | 43.8 | 34.4 | 74.4 |
| | | SVD | **61.9** | **53.6** | 48.7 | 55.5 | **54.4** | 67.7 | 46.2 | 66.5 | **44.9** | 35.5 | **75.4** |
| | | LRS | **61.9** | 51.9 | **49.7** | **58.5** | 53.9 | **69.0** | **48.4** | **68.1** | 44.7 | **35.7** | 75.0 |
| | GGS | W/O | 64.7 | 55.3 | 43.9 | 63.1 | 52.2 | 80.4 | 44.0 | 73.3 | 34.7 | 21.7 | **66.2** |
| | | SVD | 67.7 | 59.3 | 46.1 | 67.1 | **55.5** | **82.2** | 45.4 | 79.8 | 34.5 | 22.7 | 65.0 |
| | | LRS | **69.9** | **60.0** | **46.5** | **70.4** | 54.2 | 81.7 | **48.1** | **81.8** | **34.9** | **23.6** | 65.7 |
| | BFA | W/O | 45.1 | 36.4 | 29.5 | 46.9 | **36.5** | 63.4 | 28.6 | 58.4 | 32.6 | 23.9 | 65.0 |
| | | SVD | 17.6 | 13.0 | 12.7 | 18.2 | 23.7 | 24.9 | 11.3 | 19.3 | 30.4 | 22.0 | 64.4 |
| | | LRS | **45.3** | **36.6** | **30.4** | **47.4** | 35.8 | **63.9** | **29.2** | **58.8** | **33.2** | **24.2** | **65.2** |
| | P2FA | W/O | 47.1 | 38.2 | 31.0 | **54.9** | 46.1 | **74.0** | 28.6 | 67.5 | 37.1 | **30.0** | 72.6 |
| | | SVD | 47.0 | 37.6 | 30.6 | 53.9 | 46.4 | 72.9 | 27.2 | 66.0 | 35.3 | 28.9 | 72.0 |
| | | LRS | **48.4** | **38.7** | **32.5** | 54.8 | **47.3** | **74.0** | **28.9** | **68.8** | **37.7** | 29.6 | **73.4** |
| Densenet-121 | SIA | W/O | 77.0 | 71.1 | 64.0 | 67.9 | 61.5 | 85.1 | 60.9 | 78.0 | 43.5 | 35.0 | 73.3 |
| | | SVD | 73.3 | 65.5 | 59.2 | 62.3 | 58.2 | 82.4 | 56.1 | 71.8 | 43.3 | 34.1 | 73.1 |
| | | LRS | **79.9** | **74.5** | **65.9** | **71.8** | **65.5** | **87.3** | **64.2** | **80.5** | **44.6** | **35.5** | **74.0** |
| | BSR | W/O | 78.2 | 67.7 | 60.7 | 63.0 | 61.6 | 83.3 | 57.6 | 71.5 | 44.3 | **35.0** | **73.4** |
| | | SVD | 71.5 | 62.7 | 56.4 | 55.8 | 59.9 | 78.2 | 54.7 | 64.6 | 43.7 | 34.1 | 72.7 |
| | | LRS | **80.7** | **73.7** | **63.5** | **65.5** | **66.7** | **85.6** | **62.9** | **75.7** | **44.6** | 34.9 | 73.3 |
| | PGN | W/O | 81.9 | 75.3 | 67.8 | 72.5 | 66.8 | 84.4 | 65.5 | 78.6 | 48.4 | 39.3 | 74.8 |
| | | SVD | 64.5 | 54.7 | 49.8 | 57.2 | 53.5 | 71.8 | 46.4 | 69.3 | 36.5 | 24.7 | 63.9 |
| | | LRS | **82.6** | **76.6** | **70.9** | **76.1** | **69.9** | **87.0** | **70.0** | **81.6** | **49.7** | **41.0** | **75.2** |
| | GGS | W/O | 88.3 | 84.0 | **76.0** | 83.2 | **72.5** | **94.7** | 73.4 | 90.6 | 41.5 | 29.1 | 67.9 |
| | | SVD | 83.7 | 78.6 | 66.6 | 76.5 | 67.8 | 91.4 | 65.2 | 84.5 | 41.2 | 27.3 | 66.9 |
| | | LRS | **88.6** | **84.3** | 75.6 | **85.2** | 72.1 | 94.6 | **74.4** | **90.9** | **42.0** | **29.7** | **68.6** |
| | BFA | W/O | 64.0 | 56.0 | 47.6 | 56.0 | 51.2 | 78.3 | **45.8** | **65.5** | 35.2 | 24.2 | 66.2 |
| | | SVD | 60.3 | 51.9 | 44.3 | 54.9 | 49.7 | 76.7 | 42.2 | 63.3 | 34.2 | 23.2 | 65.8 |
| | | LRS | **64.4** | **57.0** | **49.5** | **56.5** | **52.3** | **78.4** | 45.7 | 65.1 | **35.4** | **24.4** | **66.9** |
| | P2FA | W/O | 64.5 | 57.8 | **45.3** | **61.3** | 59.1 | 82.1 | **40.1** | 65.2 | 30.7 | 22.8 | 65.6 |
| | | SVD | 57.9 | 54.7 | 42.4 | 58.6 | 55.0 | 76.4 | 34.3 | 65.4 | 30.3 | 21.8 | 63.8 |
| | | LRS | **64.6** | **58.6** | 44.9 | 59.8 | **61.6** | **82.7** | 39.4 | **66.8** | **30.9** | **23.0** | **67.1** |

*Table 8.* ASR (%) of different attacks evaluated on various CNN target models using additional CNN surrogate models. Best results are highlighted in bold.

| Model | Attack | Method | Inc-v3 | Inc-v4 | Vgg-16 | Vgg-19 | Dense-121 | IncRes-v2 | Res152 | Res50 | Res101 |
|---|---|---|---|---|---|---|---|---|---|---|---|
| Inc-v4 | MI-FGSM | W/O | 60.5 | 99.9 | 64.9 | 63.3 | 47.5 | 46.9 | 40.3 | 57.1 | 51.0 |
| | | SVD | 61.9 | **100.0** | 68.5 | 66.4 | 53.7 | 48.4 | 42.1 | 59.9 | 54.7 |
| | | LRS | **74.4** | 99.8 | **74.9** | **73.1** | **58.3** | **59.2** | **48.6** | **66.5** | **60.1** |
| | DI-FGSM | W/O | 54.6 | 99.2 | 52.3 | 50.1 | 34.8 | 36.0 | 27.5 | 43.4 | 37.5 |
| | | SVD | 57.1 | **99.5** | 57.0 | 58.9 | 38.2 | 39.6 | 30.3 | 47.2 | 41.7 |
| | | LRS | **62.9** | 99.3 | **61.0** | **59.3** | **41.0** | **42.7** | **33.7** | **50.7** | **47.3** |
| | TI-FGSM | W/O | 35.2 | 99.8 | 45.4 | 43.1 | 29.7 | 18.4 | 23.5 | 28.3 | 25.2 |
| | | SVD | 35.3 | **100.0** | 48.9 | 46.5 | 33.0 | 19.8 | 25.0 | 29.6 | 25.3 |
| | | LRS | **42.8** | 99.6 | **52.0** | **51.0** | **36.6** | **24.0** | **29.2** | **33.4** | **28.7** |
| | PI-FGSM | W/O | 60.8 | **98.7** | 79.8 | 80.2 | 66.5 | 40.0 | 54.9 | 66.2 | 62.1 |
| | | SVD | 67.2 | **98.7** | 83.5 | 82.1 | 71.7 | **41.8** | 56.9 | 67.8 | 63.7 |
| | | LRS | 63.7 | **98.7** | **84.2** | **83.9** | **74.9** | 40.4 | **57.4** | **69.5** | **65.5** |
| | SI-NI-FGSM | W/O | 86.1 | **100.0** | 83.2 | 83.1 | 78.8 | 78.0 | 69.5 | 77.3 | 74.3 |
| | | SVD | 88.1 | **100.0** | 87.7 | 87.0 | 80.9 | 81.2 | 72.6 | 81.3 | 77.3 |
| | | LRS | **90.0** | **100.0** | **87.8** | **88.7** | **84.9** | **84.3** | **74.6** | **83.2** | **80.9** |
| | VMI-FGSM | W/O | 77.6 | **99.9** | 75.5 | 75.5 | 66.6 | 69.0 | 60.3 | 68.9 | 65.4 |
| | | SVD | 82.4 | **99.9** | 79.3 | 79.5 | 72.4 | 73.3 | 63.5 | 75.4 | 69.7 |
| | | LRS | **85.3** | 99.8 | **83.4** | **81.7** | **75.9** | **78.4** | **66.5** | **76.2** | **73.0** |
| | GI-FGSM | W/O | 70.1 | **100.0** | 71.7 | 72.0 | 55.8 | 59.0 | 48.9 | 65.8 | 59.0 |
| | | SVD | 73.7 | **100.0** | 79.4 | 78.2 | 61.9 | 64.8 | 50.9 | 71.8 | 66.0 |
| | | LRS | **81.4** | 99.9 | **82.2** | **81.9** | **67.6** | **73.3** | **57.1** | **75.9** | **70.5** |
| Res152-v1 | MI-FGSM | W/O | 67.7 | 62.0 | 78.3 | 78.5 | 89.2 | 58.0 | **100.0** | 86.5 | 87.2 |
| | | SVD | 72.4 | 65.4 | 86.5 | 84.6 | 90.6 | 62.2 | **100.0** | 88.9 | 89.4 |
| | | LRS | **79.0** | **72.6** | **86.6** | **84.9** | **93.8** | **69.7** | **100.0** | **91.4** | **92.5** |
| | DI-FGSM | W/O | 66.5 | 61.2 | 77.5 | 76.8 | 88.0 | 59.3 | **100.0** | 87.2 | 91.1 |
| | | SVD | 68.6 | 65.7 | 81.0 | 79.5 | 89.2 | 62.0 | **100.0** | 88.7 | 90.9 |
| | | LRS | **72.5** | **71.0** | **83.1** | **82.1** | **92.2** | **67.5** | **100.0** | **91.5** | **94.1** |
| | TI-FGSM | W/O | 47.0 | 40.9 | 58.5 | 57.7 | 74.7 | 36.1 | **100.0** | 71.3 | 73.7 |
| | | SVD | 50.3 | 47.4 | 65.6 | 64.4 | 78.2 | 40.0 | **100.0** | 75.5 | 74.4 |
| | | LRS | **57.2** | **52.7** | **69.2** | **67.9** | **83.6** | **47.7** | **100.0** | **80.2** | **82.4** |
| | PI-FGSM | W/O | 66.3 | 65.6 | 82.6 | 82.4 | 84.7 | 51.7 | 99.6 | 87.3 | 86.3 |
| | | SVD | 67.5 | 67.1 | **84.5** | 83.3 | 86.8 | 53.0 | 99.4 | **89.7** | 88.0 |
| | | LRS | **70.0** | **67.5** | 84.1 | **83.8** | **88.7** | **56.0** | 99.9 | 88.9 | 88.0 |
| | SI-NI-FGSM | W/O | 87.2 | 81.8 | 90.5 | 90.5 | 97.6 | 81.5 | 100.0 | 94.9 | 95.7 |
| | | SVD | 90.4 | 85.5 | 93.5 | 92.6 | 98.0 | **84.2** | 100.0 | 96.5 | 96.4 |
| | | LRS | **91.2** | **87.9** | **94.3** | **93.4** | **98.6** | **86.1** | 100.0 | **97.1** | **97.5** |
| | VMI-FGSM | W/O | 84.2 | 78.6 | 87.8 | 87.8 | 94.8 | 79.0 | **100.0** | 93.1 | 93.9 |
| | | SVD | 89.1 | 84.2 | **93.3** | 92.5 | 96.6 | 83.5 | **100.0** | 95.5 | 95.7 |
| | | LRS | **89.8** | **84.5** | 92.0 | **92.7** | **97.9** | **85.2** | **100.0** | **95.7** | **96.9** |
| | GI-FGSM | W/O | 75.7 | 70.7 | 87.2 | 86.0 | 93.8 | 67.3 | **100.0** | 91.7 | 92.4 |
| | | SVD | 83.9 | 77.5 | 93.2 | 91.8 | 96.3 | 75.9 | **100.0** | 95.3 | 96.0 |
| | | LRS | **87.2** | **81.7** | **93.4** | **92.2** | **96.8** | **79.6** | **100.0** | **97.2** | **97.0** |
| Vgg-16 | MI-FGSM | W/O | 67.2 | 66.6 | 99.9 | 98.9 | 79.2 | 55.2 | 62.2 | 85.0 | 75.7 |
| | | SVD | 69.0 | 69.8 | 99.9 | 99.3 | 80.9 | 55.6 | 63.1 | 87.1 | **80.7** |
| | | LRS | **70.9** | **72.0** | 99.9 | **99.6** | **82.9** | **58.5** | **66.1** | **87.6** | 80.5 |
| | DI-FGSM | W/O | 57.8 | 59.8 | 99.9 | 98.6 | 67.8 | 44.7 | 52.2 | 80.6 | 69.1 |
| | | SVD | 57.2 | 59.0 | **100.0** | 98.8 | 67.7 | 43.9 | 52.0 | 79.8 | 69.4 |
| | | LRS | **62.9** | **65.1** | 99.9 | **99.3** | **72.8** | **47.9** | **58.5** | **84.9** | **75.1** |
| | TI-FGSM | W/O | 48.5 | 49.8 | **99.9** | 95.2 | 64.1 | 34.9 | 44.5 | 67.4 | 57.1 |
| | | SVD | 48.8 | 48.5 | **99.9** | 95.6 | 60.1 | 33.4 | 44.4 | 66.3 | 55.9 |
| | | LRS | **52.1** | **54.6** | **99.9** | **97.0** | **67.9** | **38.2** | **47.1** | **71.4** | **62.1** |
| | PI-FGSM | W/O | **63.1** | **64.5** | 99.9 | 95.2 | **78.6** | 40.9 | 63.8 | 87.3 | 80.2 |
| | | SVD | 53.2 | 56.6 | 99.5 | 87.1 | 74.1 | 35.5 | 60.8 | 82.1 | 75.1 |
| | | LRS | 62.5 | 64.1 | **100.0** | **96.4** | 78.0 | **41.5** | 63.3 | **84.9** | 78.9 |
| | SI-NI-FGSM | W/O | 82.1 | 82.8 | **100.0** | 99.7 | 90.7 | 72.5 | 79.1 | 93.4 | 91.1 |
| | | SVD | 83.5 | 86.2 | **100.0** | **99.9** | 90.9 | 74.9 | 79.7 | 94.4 | 90.5 |
| | | LRS | **85.5** | **87.0** | **100.0** | 99.7 | **91.8** | **76.3** | **80.9** | **95.0** | **92.5** |
| | VMI-FGSM | W/O | 79.6 | 79.9 | **99.9** | 99.4 | 89.3 | 67.7 | 77.5 | 92.8 | 87.4 |
| | | SVD | 81.8 | 80.1 | **99.9** | 99.8 | **90.8** | 68.8 | 80.7 | **95.0** | 88.8 |
| | | LRS | **82.5** | **81.7** | **99.9** | **99.9** | 90.0 | **69.5** | **81.7** | 93.2 | 87.9 |
| | GI-FGSM | W/O | 72.2 | 76.2 | 99.9 | 99.6 | 86.1 | 63.2 | 70.9 | 92.4 | 86.1 |
| | | SVD | 76.8 | 79.4 | **100.0** | **100.0** | 89.2 | 64.1 | 73.2 | 94.7 | **90.3** |
| | | LRS | **77.9** | **81.1** | 99.9 | 99.9 | 89.0 | **66.3** | **73.8** | **94.8** | 89.0 |
| IncRes-v2 | MI-FGSM | W/O | 60.3 | 51.8 | 62.9 | 59.8 | 49.4 | 98.9 | 45.8 | 57.3 | 52 |
| | | SVD | 71.8 | 62.3 | **71.4** | 69.3 | 58.6 | **99.7** | 51.9 | 65.5 | 59.7 |
| | | LRS | **77.3** | **68.6** | 70.6 | **70.8** | **62.6** | 99.6 | **54.3** | **66.8** | **60** |
| | DI-FGSM | W/O | 53.6 | 51.0 | 50.3 | 48.5 | 38.4 | 97.7 | 34.3 | 45.3 | 39.6 |
| | | SVD | 64.9 | 59.5 | **59.5** | 57.5 | 45.6 | **99.4** | 39.3 | **54.1** | 46.5 |
| | | LRS | **67.6** | **62.0** | 56.5 | 57.3 | **47.0** | 98.6 | **43.2** | 54.1 | **49.3** |
| | TI-FGSM | W/O | 34.4 | 32.6 | 42.6 | 39.8 | 35.9 | 98.6 | 29 | 29.8 | 26.2 |
| | | SVD | 38.3 | 37.3 | 48.6 | 47.0 | 39.1 | **99.7** | 33.6 | 33.6 | 31.0 |
| | | LRS | **44.9** | **40.5** | **51.8** | **48.3** | **43.5** | 99.3 | **34.4** | **34.9** | **33.9** |
| | PI-FGSM | W/O | 62.4 | 56.0 | 78.6 | 78.1 | 70 | 95.6 | 56.2 | 66.6 | 62.6 |
| | | SVD | 67.3 | 63.6 | 82.0 | 81.1 | **76.8** | **98.1** | 63.2 | **71.9** | 66.9 |
| | | LRS | **68.4** | **65.8** | **82.5** | **82.3** | 76.4 | 96.8 | **63.3** | 70.5 | **67.6** |
| | SI-NI-FGSM | W/O | 86.0 | 83.7 | 80.9 | 80.9 | 80.3 | 99.9 | 74.2 | 77.5 | 74.3 |
| | | SVD | 91.0 | 87.5 | **87.6** | 86.2 | **85.9** | **100.0** | 79.2 | **84.9** | 81.6 |
| | | LRS | **92.3** | **89.5** | 85.9 | **87.5** | 85.1 | **100.0** | 79.2 | 83.9 | **81.8** |
| | VMI-FGSM | W/O | 80.5 | 79.9 | 72.3 | 71.4 | 68.8 | 96.0 | 64.9 | 68.4 | 64.4 |
| | | SVD | 84.3 | 81.9 | 78.1 | 78.4 | 74.7 | 99.3 | 71.6 | 76.7 | 71.2 |
| | | LRS | **86.4** | **82.9** | **81.7** | **80.9** | **78.7** | **99.7** | **72.4** | **79.2** | **74.5** |
| | GI-FGSM | W/O | 71.2 | 67.3 | 71.3 | 69.5 | 59.7 | 99.3 | 52.7 | 65.7 | 60.8 |
| | | SVD | 81.6 | 78.7 | 81.5 | 80.4 | **71.3** | 99.7 | 62.0 | 77.7 | 70.5 |
| | | LRS | **84.2** | **80.0** | **82.1** | **81.8** | 70.1 | **100.0** | **62.9** | **78.6** | **73.9** |

*Table 9.* ASR (%) of different attacks evaluated on various defended CNN target models and defense methods using additional CNN surrogate models. Best results are highlighted in bold.

| Model | Attack | Method | Inc-v3$_{adv}$ | Inc-v3$_{ens3}$ | Inc-v3$_{ens4}$ | IncRes-v2$_{ens}$ | HGD | JPEG | R&P | NIPS_r3 |
|---|---|---|---|---|---|---|---|---|---|---|
| Inc-v4 | MI-FGSM | W/O | 24.4 | 19.2 | 17.6 | 11.1 | 7.3 | 36.4 | 22.1 | 22.5 |
| | | SVD | 25.6 | 20.9 | 19.5 | 10.7 | 8.0 | 37.7 | 24.8 | 23.9 |
| | | LRS | **27.1** | **24.0** | **23.2** | **12.9** | **9.3** | **43.9** | **28.8** | **29.8** |
| | DI-FGSM | W/O | 13.0 | 13.6 | 13.1 | 6.5 | 6.6 | 20.9 | 10.9 | 10.4 |
| | | SVD | 13.9 | 14.3 | 14.5 | 7.5 | 7.5 | 23.7 | 11.8 | 12.8 |
| | | LRS | **14.9** | **15.4** | **15.5** | **9.0** | **7.8** | **25.0** | **13.4** | **13.9** |
| | TI-FGSM | W/O | 11.8 | 12.4 | 10.9 | 6.8 | 5.9 | 20.8 | 9.4 | 9.7 |
| | | SVD | 12.1 | 11.8 | 12.8 | 7.0 | 5.8 | 20.7 | 9.4 | 8.9 |
| | | LRS | **14.0** | **12.9** | **13.4** | **8.1** | **7.2** | **24.4** | **12.3** | **12.0** |
| | PI-FGSM | W/O | 41.9 | 43.4 | 43.4 | 33.5 | 31.8 | 61.8 | 38.9 | 39.7 |
| | | SVD | 42.6 | 43.6 | 45.6 | 34.0 | 33.6 | 64.5 | 41.0 | **43.5** |
| | | LRS | **44.1** | **44.8** | **48.1** | **36.4** | **34.4** | **65.2** | **41.7** | 42.8 |
| | SI-NI-FGSM | W/O | 46.9 | 47.8 | 43.1 | **29.2** | **25.9** | 60.5 | 51.3 | 52.8 |
| | | SVD | 44.6 | 47.8 | 42.3 | 26.9 | 24.2 | 65.7 | 51.6 | 52.6 |
| | | LRS | **47.6** | **49.1** | **44.4** | 28.6 | 23.8 | **67.1** | **55.7** | **57.0** |
| | VMI-FGSM | W/O | 38.6 | 38.3 | 38.0 | 24.7 | 21.7 | 50.3 | 42.6 | 40.9 |
| | | SVD | 41.4 | 44.1 | 42.6 | 28.7 | 26.9 | 54.6 | 44.4 | 45.5 |
| | | LRS | **43.5** | **45.4** | **44.7** | **29.8** | **29.6** | **58.5** | **49.6** | **50.2** |
| | GI-FGSM | W/O | 27.9 | 19.9 | 18.7 | 12.0 | 3.4 | 44.9 | 29.4 | 29.7 |
| | | SVD | 29.8 | 21.2 | 20.9 | 12.3 | 3.5 | 48.0 | 32.1 | 31.6 |
| | | LRS | **33.6** | **23.5** | **23.8** | **12.4** | **5.0** | **50.3** | **36.6** | **37.2** |
| Res152-v1 | MI-FGSM | W/O | 52.1 | 49.5 | 44.8 | 38.2 | 70.0 | 72.5 | 40.7 | 40.7 |
| | | SVD | 55.5 | 52.4 | 49.4 | 41.2 | 67.2 | 79.1 | 43.4 | 43.1 |
| | | LRS | **61.6** | **59.9** | **54.0** | **46.2** | **78.4** | **82.9** | **48.1** | **49.2** |
| | DI-FGSM | W/O | 47.3 | 45.6 | 40.6 | 30.4 | 71.4 | 59.3 | 30.5 | 31.6 |
| | | SVD | 49.7 | 46.7 | 42.4 | 32.9 | 66.6 | 61.4 | 31.1 | 32.7 |
| | | LRS | **56.1** | **48.6** | **45.1** | **34.5** | **77.5** | **64.9** | **32.8** | **34.9** |
| | TI-FGSM | W/O | 37.6 | 34.6 | 32.1 | 24.7 | 52.9 | 57.5 | 25.4 | 26.0 |
| | | SVD | 39.8 | 37.1 | 34.1 | 24.3 | 49.8 | 61.5 | 27.0 | 27.8 |
| | | LRS | **45.1** | **40.3** | **37.4** | **31.2** | **62.3** | **64.6** | **30.4** | **31.9** |
| | PI-FGSM | W/O | 63.4 | 60.8 | 68.0 | 59.3 | 67.3 | 84.5 | 64.0 | 62.6 |
| | | SVD | 62.9 | 64.5 | 69.9 | 60.2 | 68.6 | 85.4 | 64.7 | 64.0 |
| | | LRS | **65.0** | **67.6** | **70.8** | **62.8** | **73.0** | **86.8** | **66.7** | **68.0** |
| | SI-NI-FGSM | W/O | 75.9 | 72.3 | 70.5 | 59.3 | 86.2 | 88.6 | 62.1 | 64.1 |
| | | SVD | 77.9 | 75.9 | 72.2 | 63.1 | 86.0 | 91.6 | 63.6 | 64.8 |
| | | LRS | **80.8** | **78.3** | **74.8** | **66.2** | **89.5** | **92.2** | **67.2** | **68.3** |
| | VMI-FGSM | W/O | 72.8 | 70.2 | 69.2 | 62.1 | 86.0 | 86.5 | 61.2 | 61.7 |
| | | SVD | **79.3** | 76.2 | 74.7 | 68.5 | 86.9 | 90.2 | 69.1 | 69.7 |
| | | LRS | 79.0 | **77.2** | **74.9** | **69.1** | **89.5** | **91.3** | 68.6 | **70.4** |
| | GI-FGSM | W/O | 58.1 | 56.5 | 54.9 | 43.9 | 77.7 | 82.9 | 49.4 | 52.1 |
| | | SVD | 66.4 | 61.1 | 59.7 | 48.0 | 79.6 | 87.5 | 55.4 | 56.3 |
| | | LRS | **70.6** | **66.0** | **62.3** | **53.5** | **85.5** | **89.2** | **59.4** | **60.4** |
| Vgg-16 | MI-FGSM | W/O | 45.6 | 41.3 | 38.8 | 27.4 | 40.9 | 57.1 | 33.2 | **35.6** |
| | | SVD | 46.0 | 41.4 | 36.9 | **29.4** | 41.2 | 59.5 | 34.5 | 34.4 |
| | | LRS | **47.9** | **45.0** | **41.9** | 28.7 | **44.6** | **61.2** | **34.9** | **35.6** |
| | DI-FGSM | W/O | 34 | 29 | 27.5 | 17.8 | 29.3 | 37.7 | **21.2** | 21.6 |
| | | SVD | 33.2 | 28.9 | 26.1 | 17.7 | 27.1 | 40.0 | 20.1 | 21.2 |
| | | LRS | **36.3** | **32.6** | **28.4** | **19.4** | **33.9** | **40.3** | 20.9 | **22.9** |
| | TI-FGSM | W/O | 33.4 | 29.6 | 28 | 19.1 | 27.2 | 43.3 | 23.3 | 24.3 |
| | | SVD | 31.9 | 28.6 | 27.3 | 19.4 | 25.4 | 44.0 | 23.8 | 23.3 |
| | | LRS | **34.3** | **31.6** | **30.5** | **21.3** | **30.1** | **47.9** | **25.0** | **26.5** |
| | PI-FGSM | W/O | 49.1 | 49.8 | 55.1 | 41.9 | 46.8 | 69.5 | 48.3 | **48.4** |
| | | SVD | 46.6 | **51.4** | **56.4** | **44.7** | 45.3 | 72.6 | **50.2** | 46.8 |
| | | LRS | **49.8** | 50.5 | 55.7 | 41.4 | **46.4** | **74.3** | 46.3 | 46.1 |
| | SI-NI-FGSM | W/O | 64.6 | 56.6 | 53.6 | 42.1 | 61.0 | 73.6 | 47.4 | 49.1 |
| | | SVD | 66.0 | 57.8 | 53.2 | 42.5 | 60.3 | 74.4 | 48.6 | 50.7 |
| | | LRS | **67.9** | **60.1** | **56.0** | **45.8** | **64.4** | **76.2** | **51.4** | **53.1** |
| | VMI-FGSM | W/O | 59.9 | 54.5 | 51.9 | 42.3 | 58.0 | 71.6 | 46.4 | 49.5 |
| | | SVD | 62.2 | 55.5 | 53.5 | 43.3 | 59.1 | 73.7 | **50.1** | 51.5 |
| | | LRS | **65.5** | **57.5** | **54.5** | **44.2** | **62.8** | **77.2** | 49.6 | **51.8** |
| | GI-FGSM | W/O | 50.2 | 43.5 | 40.8 | 29.4 | 46.3 | 65.4 | 38.3 | 38.2 |
| | | SVD | 52.1 | 43.4 | 43.0 | 30.5 | 47.9 | 66.3 | **40.7** | 40.4 |
| | | LRS | **53.6** | **47.2** | **43.5** | **32.2** | **52.7** | **66.9** | 40.2 | **41.4** |
| IncRes-v2 | MI-FGSM | W/O | 27.8 | 21.4 | 21.8 | 13.3 | 12.1 | 37.8 | 25.6 | 25.3 |
| | | SVD | 31.1 | 26.9 | 25.5 | 17.1 | 13.9 | 44.9 | 32.0 | 33.6 |
| | | LRS | **32.5** | **29.4** | **27.4** | **19.0** | **17.3** | **46.8** | **34.7** | **34.6** |
| | DI-FGSM | W/O | 15.8 | 14.9 | 15.2 | 9.8 | 9.8 | 24.0 | 14.1 | 14.4 |
| | | SVD | 17.2 | 15.8 | 17.1 | 9.9 | 11.5 | 27.6 | 15.1 | 14.9 |
| | | LRS | **17.5** | **17.9** | **17.9** | **11.9** | **12.1** | **30.2** | **16.7** | **16.1** |
| | TI-FGSM | W/O | 14.1 | 13.0 | 13.4 | 8.8 | 7.3 | 23.8 | 11.8 | 12.7 |
| | | SVD | 14.6 | 14.1 | 14.1 | 9.7 | 7.9 | 26.2 | 14.0 | 14.3 |
| | | LRS | **16.8** | **15.9** | **14.4** | **12.3** | **10.0** | **29.1** | **16.0** | **16.6** |
| | PI-FGSM | W/O | 50.1 | 49.0 | 49.9 | 44.2 | 39.9 | 65.2 | 47.7 | 49.5 |
| | | SVD | 53.9 | **55.3** | **56.5** | **51.4** | 45.0 | 70.9 | **54.7** | 55.8 |
| | | LRS | **55.3** | 54.9 | 56.2 | 50.5 | **45.4** | **71.3** | 53.9 | **56.0** |
| | SI-NI-FGSM | W/O | 57.5 | 55.6 | 48.5 | 40.2 | 37.3 | 64.7 | 56.9 | 58.3 |
| | | SVD | 60.2 | 59.3 | 52.6 | 42.1 | 38.6 | 70.2 | 63.3 | 64.3 |
| | | LRS | **62.1** | **60.5** | **53.9** | **44.0** | **42.0** | **71.5** | **64.0** | **66.4** |
| | VMI-FGSM | W/O | 43.2 | 45.2 | 40.2 | 35.0 | 31.6 | 52.6 | 45.1 | 47.8 |
| | | SVD | 50.5 | 53.8 | 46.6 | 40.6 | 39.4 | 60.5 | 53.9 | 52.4 |
| | | LRS | **53.4** | **55.3** | **52.1** | **43.2** | **43.1** | **64.2** | **54.5** | **54.7** |
| | GI-FGSM | W/O | 31.8 | 24.8 | 22.5 | 15.1 | 7.3 | 46.1 | 33.6 | 34.4 |
| | | SVD | **38.1** | 28.8 | 25.9 | 17.1 | 9.3 | 54.0 | 42.7 | 43.0 |
| | | LRS | 37.9 | **29.0** | **28.4** | **18.1** | **10.8** | **54.8** | **44.1** | **45.8** |

*Table 10.* ASR (%) of different attacks evaluated on various ViT target models and defended ViT target models using additional CNN surrogate models. Best results are highlighted in bold.

| Model | Attack | Method | PiT-S | CaiT-S | DeiT-B | Swin-S | ViT-B | Visformer-S | ConViT-B | Twins-B | DeiT-S (adv) | Swin-B (adv) | XCiT-S (adv) |
|---|---|---|---|---|---|---|---|---|---|---|---|---|---|
| Inc-v4 | MI-FGSM | W/O | 23.6 | 17.1 | 18.9 | 21.6 | 25.3 | 26.6 | 15.9 | 22.6 | **27.6** | 17.2 | 59.7 |
| | | SVD | 24.5 | 18.1 | 18.5 | 21.9 | 26.1 | 28.5 | **16.5** | 25.0 | 27.3 | 17.8 | **60.3** |
| | | LRS | **27.6** | **20.5** | **21.1** | **26.7** | **27.3** | **33.2** | 16.5 | **29.0** | 27.3 | **18.1** | 59.9 |
| | DI-FGSM | W/O | 15.8 | 10.6 | 11.5 | 17.0 | 17.5 | 19.7 | 10.1 | 17.0 | 23.3 | 15.8 | 56.2 |
| | | SVD | 17.0 | 11.0 | 11.3 | 16.3 | **18.4** | 18.9 | **10.8** | 17.5 | 23.2 | **16.0** | **56.3** |
| | | LRS | **17.1** | **12.2** | **11.8** | **18.1** | 17.9 | **20.7** | 9.9 | **19.3** | **23.8** | **16.0** | 56.2 |
| | TI-FGSM | W/O | 14.7 | 9.5 | 12.0 | 13.9 | 18.9 | 14.9 | 9.2 | 12.9 | 23.7 | 15.5 | **56.6** |
| | | SVD | 13.7 | 9.9 | 11.7 | 12.7 | 18.8 | 15.1 | **10.4** | 12.8 | **23.9** | 15.9 | 56.3 |
| | | LRS | **15.3** | **11.1** | **12.9** | **15.3** | **20.0** | **16.7** | 10.0 | **15.0** | **23.9** | **16.2** | 56.5 |
| | PI-FGSM | W/O | 21.6 | 17.6 | 18.8 | 15.6 | 37.4 | 23.7 | 16.3 | 18.2 | 42.2 | 29.1 | 66.8 |
| | | SVD | 23.0 | 17.2 | 18.9 | **18.6** | 37.3 | 24.1 | 16.4 | 18.3 | 43.0 | **29.9** | 67.3 |
| | | LRS | **24.6** | **17.8** | **19.4** | 16.3 | **38.4** | **24.8** | **16.9** | **18.7** | **43.3** | 29.6 | **68.6** |
| | SI-NI-FGSM | W/O | 47.2 | 35.4 | **33.1** | 42.5 | 38.2 | 53.3 | 32.6 | 47.7 | **30.1** | 19.7 | 60.8 |
| | | SVD | 46.3 | 36.2 | 31.3 | 41.7 | 38.7 | 52.2 | 32.5 | 47.1 | 29.7 | 20.1 | 61.5 |
| | | LRS | **48.0** | **36.6** | 31.3 | **43.2** | **39.8** | **55.6** | **32.7** | **49.4** | 30.0 | **20.8** | **62.3** |
| Res152-v1 | MI-FGSM | W/O | 44.7 | 37.6 | 32.7 | 39.0 | 36.0 | 47.6 | 32.2 | 46.6 | 35.0 | 26.4 | 66.5 |
| | | SVD | 44.1 | 35.1 | 29.6 | 36.8 | 37.0 | 48.5 | 29.5 | 44.8 | 34.0 | 25.9 | **66.8** |
| | | LRS | **51.1** | **44.6** | **38.3** | **46.0** | **40.6** | **56.9** | **39.4** | **53.4** | **35.8** | **27.1** | 66.7 |
| | DI-FGSM | W/O | 41.1 | 30.7 | 26.9 | 34.0 | 29.0 | 45.0 | 27.4 | 42.1 | 32.7 | 24.6 | 66.4 |
| | | SVD | 42.2 | 31.5 | 27.9 | 35.6 | 30.6 | 46.7 | 28.1 | 42.2 | 32.6 | **24.7** | 66.5 |
| | | LRS | **44.1** | **35.5** | **30.5** | **36.4** | **32.6** | **51.9** | **28.7** | **46.6** | **33.4** | 24.7 | **66.7** |
| | TI-FGSM | W/O | 27.7 | 21.8 | 20.1 | 21.8 | 27.2 | 29.8 | 20.9 | 26.4 | 28.3 | **20.2** | **59.8** |
| | | SVD | 27.7 | 21.7 | 20.6 | 22.3 | 27.3 | 30.0 | 21.8 | 25.3 | 29.0 | **20.2** | **59.8** |
| | | LRS | **31.9** | **27.6** | **23.2** | **25.9** | **28.4** | **36.5** | **25.1** | **34.0** | **29.8** | 20.2 | 59.5 |
| | PI-FGSM | W/O | 28.0 | 24.2 | 21.0 | **20.3** | 50.0 | 29.0 | 23.2 | **25.1** | 49.6 | 42.9 | 75.0 |
| | | SVD | 29.2 | 23.2 | 22.3 | **20.3** | 50.4 | 29.6 | 22.6 | 22.3 | 50.1 | 43.7 | 75.1 |
| | | LRS | **31.5** | **25.3** | **23.8** | 20.1 | **51.3** | **31.8** | **24.8** | 24.6 | **51.1** | **44.1** | **75.3** |
| | SI-NI-FGSM | W/O | 59.3 | 52.0 | 45.1 | 50.9 | 45.1 | 65.4 | 47.1 | 60.4 | 38.1 | 29.0 | **68.3** |
| | | SVD | 62.0 | 56.5 | **48.3** | 52.9 | 48.5 | 67.2 | **49.5** | 62.2 | **39.8** | **29.6** | 68.1 |
| | | LRS | **64.1** | **59.0** | 47.5 | **54.4** | **48.8** | **70.8** | 49.0 | **63.4** | 39.7 | 29.1 | 68.0 |
| Vgg-16 | MI-FGSM | W/O | 38.2 | 31.6 | 27.5 | 38.6 | 34.4 | 51.9 | 29.5 | 47.5 | **38.2** | 29.4 | 71.8 |
| | | SVD | 39.1 | 31.6 | 25.7 | 38.1 | 34.9 | 50.6 | 27.7 | 45.7 | **38.2** | 29.7 | **72.1** |
| | | LRS | **42.4** | **34.3** | **29.1** | **43.4** | **36.4** | **55.2** | 29.5 | **50.2** | 38.1 | **29.8** | 71.7 |
| | DI-FGSM | W/O | 27.2 | 20.0 | 18.6 | 28.1 | 24.3 | 36.8 | 18.0 | 34.2 | 31.9 | **23.7** | 65.6 |
| | | SVD | 25.7 | 18.3 | 16.2 | 27.0 | 25.2 | 35.4 | 15.9 | 31.4 | **32.1** | **23.7** | 65.8 |
| | | LRS | **30.2** | **23.1** | **19.0** | **32.6** | **25.5** | **42.6** | **19.3** | **38.6** | 31.8 | 23.7 | **65.9** |
| | TI-FGSM | W/O | 24.4 | 21.4 | 18.4 | 23.5 | 25.1 | 32.8 | 19.2 | 29.0 | **33.0** | 23.5 | 66.0 |
| | | SVD | 23.6 | 19.6 | 18.2 | 21.9 | 24.9 | 28.7 | 18.3 | 25.1 | 32.6 | 23.5 | **66.3** |
| | | LRS | **28.3** | **23.3** | 24.2 | 26.8 | 27.0 | 36.2 | 20.3 | 31.2 | **33.0** | 23.9 | 66.1 |
| | PI-FGSM | W/O | 24.5 | 17.5 | 17.1 | 17.4 | **42.7** | 23.2 | 17.8 | 20.0 | **47.7** | 39.4 | 73.9 |
| | | SVD | 18.8 | 13.2 | 14.0 | 12.1 | 41.7 | 18.9 | 15.3 | 14.1 | 46.1 | 39.2 | 72.7 |
| | | LRS | **26.9** | **20.8** | **19.8** | **18.5** | 41.8 | **26.7** | **19.9** | **22.5** | **47.7** | 39.3 | 73.4 |
| | SI-NI-FGSM | W/O | 51.5 | 44.8 | 36.8 | 50.3 | 45.3 | 67.0 | 36.3 | 60.9 | 39.9 | 30.7 | 72.7 |
| | | SVD | 52.8 | 44.1 | 37.1 | 49.1 | 44.6 | 68.1 | 36.0 | 60.2 | **40.5** | **31.4** | 72.9 |
| | | LRS | **55.2** | **46.6** | **40.0** | **53.5** | **45.6** | **69.7** | **39.5** | **63.3** | 40.3 | 31.1 | **73.4** |
| IncRes-v2 | MI-FGSM | W/O | 24.5 | 20.1 | 18.8 | 20.8 | 25.5 | 26.0 | 16.8 | 24.9 | 27.1 | 17.6 | 59.1 |
| | | SVD | 28.4 | 20.4 | 19.7 | 23.5 | **27.8** | 29.8 | 19.4 | 27.6 | **27.9** | 18.3 | **59.7** |
| | | LRS | **29.8** | **23.3** | **22.0** | **26.2** | 27.3 | **32.0** | **21.3** | **28.9** | 27.4 | 18.3 | 59.6 |
| | DI-FGSM | W/O | 17.4 | 11.5 | 11.0 | 15.4 | 17.6 | 17.7 | 11.2 | 17.3 | 24.1 | 15.9 | **56.5** |
| | | SVD | 19.5 | 13.1 | 12.0 | 16.9 | 18.9 | 21.7 | 11.8 | 19.2 | **24.3** | **16.5** | **56.5** |
| | | LRS | **21.6** | **14.3** | **13.6** | **19.9** | **19.1** | **22.2** | **12.8** | **22.2** | 24.0 | **16.5** | **56.5** |
| | TI-FGSM | W/O | 15.0 | 10.2 | 11.3 | 12.6 | 19.2 | 14.8 | 10.6 | 13.7 | 24.4 | 16.1 | 56.6 |
| | | SVD | 16.7 | 12.0 | 12.3 | 12.6 | 19.3 | 16.0 | 11.3 | 14.2 | 24.7 | 16.3 | **57.1** |
| | | LRS | **19.2** | **13.0** | **13.5** | **16.1** | **21.4** | **18.8** | **12.7** | **17.5** | **24.9** | **16.4** | 56.4 |
| | PI-FGSM | W/O | 21.2 | 17.2 | **19.7** | 15.2 | 38.9 | 22.7 | 18.4 | 18.5 | 42.4 | 29.5 | 67.0 |
| | | SVD | **25.3** | **19.4** | 19.3 | 16.8 | **41.3** | 26.8 | **19.3** | 18.2 | **44.3** | 30.6 | **68.6** |
| | | LRS | 23.5 | 19.1 | 19.3 | **17.7** | 40.5 | **27.4** | 18.0 | **20.5** | 44.1 | **30.9** | 67.6 |
| | SI-NI-FGSM | W/O | 48.5 | 40.0 | 31.7 | 39.4 | 38.4 | 49.3 | 33.7 | 44.5 | 31.3 | **21.4** | 62.6 |
| | | SVD | 49.5 | 41.9 | 33.5 | 42.9 | 41.3 | 53.6 | **35.3** | 48.7 | 31.9 | 21.3 | **62.7** |
| | | LRS | **50.2** | **42.6** | **34.7** | **43.2** | **42.1** | **54.0** | 35.0 | **49.9** | **32.5** | 21.2 | 62.6 |

*Table 11.* Complete per-model ASR (%) of different attacks evaluated on various CNN target models using CNN-based surrogate models. Best results are highlighted in bold.

| Model | Attack | Method | Inc-v3 | Inc-v4 | Vgg-16 | Vgg-19 | Dense-121 | IncRes-v2 | Res152 | Res50 | Res101 |
|---|---|---|---|---|---|---|---|---|---|---|---|
| Inc-v3 | MI-FGSM | W/O | **100.0** | 50.0 | 60.3 | 58.2 | 50.3 | 45.6 | 45.9 | 54.6 | 48.9 |
| | | SVD | 99.9 | 54.0 | 64.0 | 62.3 | 53.1 | 52.0 | 43.9 | 58.9 | 55.4 |
| | | LRS | **100.0** | **63.4** | **69.0** | **66.5** | **60.4** | **59.5** | **50.7** | **65.4** | **59.1** |
| | DI-FGSM | W/O | 99.7 | 47.9 | 48.5 | 47.1 | 40.0 | 39.1 | 34.4 | 41.9 | 37.5 |
| | | SVD | 99.7 | 51.2 | 56.4 | 51.7 | 44.5 | 41.2 | 36.5 | 47.7 | 41.5 |
| | | LRS | **99.9** | **56.5** | **57.8** | **55.9** | **46.3** | **45.6** | **38.7** | **50.6** | **45.1** |
| | TI-FGSM | W/O | 99.9 | 28.5 | 41.0 | 39.3 | 32.7 | 19.6 | 24.9 | 27.8 | 24.2 |
| | | SVD | 99.7 | 29.8 | 44.5 | 40.8 | 34.9 | 20.0 | 27.8 | 28.3 | 24.9 |
| | | LRS | **100.0** | **35.4** | **47.9** | **44.9** | **37.2** | **26.1** | **29.8** | **32.2** | **28.1** |
| | PI-FGSM | W/O | 98.9 | 57.5 | 79.1 | 78.0 | 70.5 | 40.5 | 55.6 | 65.3 | 60.8 |
| | | SVD | 98.9 | 60.4 | 80.5 | 80.2 | 74.2 | 41.1 | 56.8 | 67.5 | 63.2 |
| | | LRS | **99.6** | **63.4** | **83.0** | **82.0** | **75.6** | **43.5** | **58.7** | **69.9** | **65.7** |
| | SI-NI-FGSM | W/O | **100.0** | 77.7 | 78.0 | 77.2 | 76.2 | 75.3 | 67.4 | 74.5 | 68.9 |
| | | SVD | **100.0** | 79.5 | 82.1 | 79.9 | 78.3 | 79.0 | 70.7 | 78.1 | 74.3 |
| | | LRS | **100.0** | **81.5** | **84.3** | **83.1** | **81.9** | **82.1** | **72.0** | **81.2** | **76.9** |
| | VMI-FGSM | W/O | **100.0** | 73.8 | 72.7 | 70.7 | 66.8 | 68.9 | 60.1 | 67.7 | 63.1 |
| | | SVD | 99.6 | 78.2 | 77.7 | 77.8 | 73.7 | 75.1 | 65.0 | 75.3 | 69.2 |
| | | LRS | **100.0** | **81.1** | **79.5** | **80.7** | **75.9** | **77.9** | **68.3** | **77.5** | **72.0** |
| | GI-FGSM | W/O | **100.0** | 63.8 | 67.7 | 67.5 | 56.5 | 60.1 | 50.6 | 65.2 | 58.1 |
| | | SVD | **100.0** | 69.0 | 75.6 | 73.9 | 64.5 | 66.3 | 52.2 | 73.6 | 64.9 |
| | | LRS | **100.0** | **76.9** | **80.0** | **77.5** | **70.4** | **73.7** | **60.8** | **76.4** | **72.0** |
| Vgg-19 | MI-FGSM | W/O | 65.8 | 68.7 | 99.0 | 99.9 | 77.4 | 52.7 | 61.4 | 84.2 | 75.9 |
| | | SVD | 65.8 | 69.8 | 99.2 | **100.0** | 79.4 | 53.7 | 63.0 | 84.1 | 77.9 |
| | | LRS | **71.5** | **72.5** | **99.6** | 99.9 | **81.7** | **57.8** | **66.8** | **89.0** | **79.9** |
| | DI-FGSM | W/O | 56.0 | 60.1 | 98.5 | 99.9 | 64.1 | 41.0 | 50.7 | 77.7 | 68.5 |
| | | SVD | 54.0 | 59.2 | 98.3 | **100.0** | 63.0 | 40.2 | 48.2 | 76.9 | 66.9 |
| | | LRS | **60.9** | **62.9** | **99.1** | **100.0** | **70.4** | **44.0** | **53.7** | **82.1** | **71.5** |
| | TI-FGSM | W/O | 48.3 | 50.0 | 96.2 | 99.9 | 59.2 | 32.3 | 43.8 | 66.2 | 56.3 |
| | | SVD | 45.6 | 46.6 | 95.2 | **100.0** | 56.9 | 30.4 | 40.3 | 63.6 | 53.4 |
| | | LRS | **53.5** | **54.5** | **97.6** | 99.9 | **64.3** | **38.1** | **48.2** | **72.4** | **62.8** |
| | PI-FGSM | W/O | 60.5 | 62.4 | 94.5 | 99.9 | 76.1 | 38.7 | 61.1 | 83.4 | 77.7 |
| | | SVD | 60.5 | 61.2 | 94.4 | **100.0** | 75.1 | 39.4 | 60.3 | 84.1 | 77.7 |
| | | LRS | **62.9** | **63.5** | **96.5** | **100.0** | **76.9** | **41.1** | **63.5** | **85.5** | **78.5** |
| | SI-NI-FGSM | W/O | 82.8 | 85.0 | 99.5 | **100.0** | 91.2 | 72.7 | 77.6 | 93.6 | 89.4 |
| | | SVD | 83.2 | 86.8 | 99.5 | **100.0** | 90.3 | 74.6 | 80.1 | 92.6 | 90.2 |
| | | LRS | **86.1** | **89.0** | **99.9** | **100.0** | **93.9** | **79.0** | **83.0** | **94.2** | **91.6** |
| | VMI-FGSM | W/O | 79.0 | 80.7 | 99.5 | 99.9 | 87.9 | 67.9 | 76.5 | 91.1 | 86.4 |
| | | SVD | 80.0 | **83.4** | 99.5 | **100.0** | 88.6 | 68.3 | 77.6 | 92.5 | 87.5 |
| | | LRS | **80.9** | **83.4** | **99.8** | 99.9 | **88.9** | **70.2** | **78.3** | **93.7** | **87.9** |
| | GI-FGSM | W/O | 73.8 | 78.7 | 99.7 | **100.0** | 87.3 | 61.8 | 71.7 | 91.2 | 85.6 |
| | | SVD | 75.7 | 80.0 | 99.7 | **100.0** | 88.6 | 64.0 | 73.0 | **93.3** | 86.2 |
| | | LRS | **75.8** | **80.5** | **99.8** | 99.7 | **88.8** | **66.2** | **73.5** | **93.3** | **86.9** |
| Densenet-121 | MI-FGSM | W/O | 67.5 | 62.8 | 84.7 | 83.2 | **100.0** | 57.2 | 82.5 | 87.3 | 85.3 |
| | | SVD | 69.2 | 63.4 | 86.5 | 84.7 | **100.0** | 57.5 | 81.4 | 87.9 | 85.6 |
| | | LRS | **77.1** | **72.4** | **89.4** | **88.9** | **100.0** | **66.0** | **89.8** | **93.0** | **91.3** |
| | DI-FGSM | W/O | 63.1 | 62.8 | 78.8 | 77.4 | **100.0** | 53.4 | 77.6 | 86.9 | 83.9 |
| | | SVD | 64.1 | 61.5 | 82.6 | 79.9 | **100.0** | 51.6 | 74.6 | 86.5 | 80.5 |
| | | LRS | **68.9** | **70.0** | **84.7** | **85.5** | **100.0** | **60.9** | **83.8** | **91.0** | **88.3** |
| | TI-FGSM | W/O | 48.9 | 46.3 | 64.6 | 62.2 | **100.0** | 36.6 | 60.3 | 68.5 | 65.3 |
| | | SVD | 46.6 | 44.7 | 65.3 | 61.7 | **100.0** | 33.3 | 57.8 | 66.5 | 61.3 |
| | | LRS | **54.6** | **55.1** | **72.8** | **68.5** | **100.0** | **42.5** | **67.8** | **76.0** | **72.3** |
| | PI-FGSM | W/O | 66.2 | 65.8 | 86.3 | 84.1 | **100.0** | 49.5 | 72.2 | 86.7 | 82.2 |
| | | SVD | 66.1 | 66.5 | 86.9 | 85.1 | **100.0** | 49.2 | 72.8 | 88.2 | 83.8 |
| | | LRS | **69.8** | **67.8** | **87.5** | **87.7** | **100.0** | **53.6** | **75.2** | **89.4** | **84.3** |
| | SI-NI-FGSM | W/O | 83.7 | 81.7 | 91.9 | 91.4 | **100.0** | 78.8 | 93.2 | 93.4 | 93.3 |
| | | SVD | 85.2 | 81.8 | 92.7 | 91.9 | **100.0** | 77.6 | 92.8 | 94.4 | 93.7 |
| | | LRS | **88.6** | **85.8** | **94.3** | **93.9** | **100.0** | **82.9** | **95.6** | **95.5** | **95.6** |
| | VMI-FGSM | W/O | 82.9 | 81.6 | 92.1 | 90.9 | **100.0** | 77.3 | 92.8 | 93.6 | 92.7 |
| | | SVD | 85.1 | 84.4 | 93.8 | 93.2 | **100.0** | 78.8 | 92.1 | 94.3 | 93.7 |
| | | LRS | **86.7** | **86.6** | **94.6** | **93.7** | **100.0** | **82.6** | **95.4** | **96.3** | **95.3** |
| | GI-FGSM | W/O | 81.5 | 78.7 | 92.6 | 91.0 | **100.0** | 71.7 | 91.0 | 93.6 | 92.5 |
| | | SVD | 82.2 | 79.3 | 94.5 | 92.8 | **100.0** | 73.2 | 91.9 | 94.2 | 93.7 |
| | | LRS | **87.8** | **85.3** | **95.8** | **94.7** | **100.0** | **80.9** | **94.8** | **96.9** | **95.7** |

*Table 12.* Complete per-model ASR (%) of different attacks evaluated on various defended CNN target models and defense methods using CNN-based surrogate models. Best results are highlighted in bold.

| Model | Attack | Method | Inc-v3$_{adv}$ | Inc-v3$_{ens3}$ | Inc-v3$_{ens4}$ | IncRes-v2$_{ens}$ | HGD | JPEG | R&P | NIPS_r3 |
|---|---|---|---|---|---|---|---|---|---|---|
| Inc-v3 | MI-FGSM | W/O | 26.5 | 23.0 | 22.5 | 10.4 | 8.1 | 38.9 | 23.0 | 24.7 |
| | | SVD | 29.7 | 23.3 | 23.8 | 12.5 | 7.2 | 41.0 | 24.8 | 27.4 |
| | | LRS | **30.4** | **24.3** | **25.6** | **12.6** | **9.6** | **46.6** | **27.6** | **30.6** |
| | DI-FGSM | W/O | 15.9 | 15.8 | 17.1 | 7.4 | 7.4 | 26.8 | 11.4 | 13.1 |
| | | SVD | 16.1 | 16.3 | 17.6 | 7.6 | 7.6 | 29.5 | 12.5 | 14.3 |
| | | LRS | **17.2** | **17.8** | **18.2** | **8.1** | **9.5** | **31.0** | **15.6** | **15.3** |
| | TI-FGSM | W/O | 13.8 | 12.9 | 13.3 | 6.6 | 5.3 | 23.3 | 9.3 | 9.8 |
| | | SVD | 13.1 | 13.7 | 13.9 | 6.5 | 5.5 | 25.4 | 9.4 | 10.6 |
| | | LRS | **15.4** | **14.3** | **15.6** | **7.8** | **7.5** | **28.3** | **11.2** | **12.2** |
| | PI-FGSM | W/O | 45.4 | 44.9 | 46.2 | 33.1 | 30.3 | 63.7 | 41.2 | 42.9 |
| | | SVD | 47.0 | 46.9 | 49.3 | 35.9 | 33.6 | 67.0 | 41.0 | 44.8 |
| | | LRS | **48.4** | **47.2** | **51.0** | **35.7** | **36.6** | **69.1** | **44.6** | **47.5** |
| | SI-NI-FGSM | W/O | 45.6 | 38.5 | 38.4 | 23.4 | 18.7 | 61.2 | 44.6 | 48.5 |
| | | SVD | 46.0 | 39.9 | 40.1 | 22.8 | 17.7 | 65.3 | 46.0 | 50.1 |
| | | LRS | **49.2** | **40.9** | **41.8** | **24.8** | **19.4** | **68.8** | **50.0** | **52.6** |
| | VMI-FGSM | W/O | 42.2 | 38.7 | 39.6 | 23.4 | 21.1 | 53.4 | 41.8 | 43.2 |
| | | SVD | 49.0 | 44.2 | 46.1 | **26.9** | 26.3 | 60.6 | 45.5 | 48.2 |
| | | LRS | **48.2** | **45.6** | **46.5** | 26.3 | **28.0** | **62.4** | **46.8** | **50.4** |
| | GI-FGSM | W/O | 30.7 | 22.9 | 23.0 | 10.7 | 5.1 | 63.4 | 29.4 | 31.4 |
| | | SVD | 32.8 | 23.3 | 22.8 | 11.1 | 4.1 | **65.0** | 31.9 | 35.3 |
| | | LRS | **34.5** | **26.8** | **24.7** | **13.2** | **5.7** | 64.9 | **38.6** | **38.9** |
| Vgg-19 | MI-FGSM | W/O | 43.3 | 41.0 | 37.9 | 29.0 | 42.5 | 56.4 | 32.7 | 34.6 |
| | | SVD | 45.7 | 40.7 | 35.9 | 28.6 | 40.5 | 55.6 | 32.8 | 33.7 |
| | | LRS | **47.9** | **43.9** | **40.4** | **29.8** | **47.3** | **59.5** | **35.0** | **36.5** |
| | DI-FGSM | W/O | 30.9 | 27.3 | 25.3 | 16.7 | 27.8 | 38.0 | 19.4 | 20.3 |
| | | SVD | 29.8 | 27.1 | 24.5 | 17.0 | 26.0 | 36.5 | 19.2 | 21.1 |
| | | LRS | **34.5** | **30.4** | **27.6** | **18.5** | **32.8** | **39.0** | **21.0** | **22.2** |
| | TI-FGSM | W/O | 30.6 | 28.3 | 29.0 | 18.8 | 26.8 | 42.4 | 22.7 | 23.2 |
| | | SVD | 30.2 | 27.8 | 24.9 | 18.3 | 23.2 | 42.4 | 22.5 | 23.6 |
| | | LRS | **34.8** | **30.7** | **29.4** | **20.1** | **29.6** | **45.8** | **24.9** | **26.1** |
| | PI-FGSM | W/O | 48.0 | 49.2 | 52.2 | 40.0 | 45.4 | 70.7 | 46.1 | **46.4** |
| | | SVD | 47.5 | 48.5 | 51.7 | 40.6 | 43.9 | 71.9 | 45.9 | 45.2 |
| | | LRS | **50.7** | **50.2** | **52.6** | **40.9** | **45.6** | **73.0** | **46.3** | 45.9 |
| | SI-NI-FGSM | W/O | 64.9 | 58.6 | 53.2 | 42.2 | 60.2 | 71.5 | 47.8 | 50.2 |
| | | SVD | 63.9 | 56.3 | 51.6 | 43.1 | 58.8 | 71.4 | 45.8 | 49.2 |
| | | LRS | **68.7** | **62.7** | **55.7** | **45.9** | **64.4** | **74.7** | **49.7** | **51.9** |
| | VMI-FGSM | W/O | 60.3 | 54.3 | 53.1 | 42.3 | 57.5 | 72.5 | 46.1 | 50.6 |
| | | SVD | **61.8** | 54.8 | 51.0 | 42.9 | 56.5 | 73.1 | 45.6 | 48.8 |
| | | LRS | 61.7 | **56.8** | **54.3** | **44.2** | **61.1** | **74.7** | **48.3** | **51.7** |
| | GI-FGSM | W/O | 50.6 | 44.4 | 41.5 | 28.9 | 45.6 | 63.4 | 37.6 | 40.2 |
| | | SVD | 51.3 | 44.7 | 41.6 | 29.0 | 43.4 | 64.8 | 38.7 | 38.9 |
| | | LRS | **53.1** | **45.3** | **42.6** | **30.5** | **48.3** | **65.0** | **39.1** | **40.7** |
| Densenet-121 | MI-FGSM | W/O | 51.7 | 50.4 | 49.6 | 38.3 | 53.9 | 72.7 | 40.2 | 41.3 |
| | | SVD | 51.8 | 48.6 | 46.7 | 35.5 | 49.5 | 71.7 | 41.3 | 40.2 |
| | | LRS | **60.9** | **57.0** | **56.1** | **43.6** | **63.3** | **79.7** | **46.2** | **50.3** |
| | DI-FGSM | W/O | 46.5 | 43.5 | 41.0 | 30.7 | 49.4 | 57.1 | 23.4 | 36.1 |
| | | SVD | 45.0 | 40.2 | 39.4 | 28.7 | 44.2 | 55.0 | 30.4 | 34.9 |
| | | LRS | **50.5** | **46.8** | **45.5** | **32.8** | **56.4** | **61.2** | **33.2** | **40.9** |
| | TI-FGSM | W/O | 37.9 | 35.1 | 35.7 | 25.5 | 35.9 | 23.3 | 28.5 | 27.5 |
| | | SVD | 35.7 | 33.8 | 34.7 | 24.6 | 31.7 | 25.4 | 27.0 | 26.8 |
| | | LRS | **43.1** | **43.3** | **40.8** | **29.3** | **41.9** | **28.3** | **31.9** | **31.9** |
| | PI-FGSM | W/O | 60.9 | 63.8 | 67.0 | 55.4 | 61.1 | 81.2 | 61.6 | 60.0 |
| | | SVD | 60.4 | 63.0 | 67.8 | 55.6 | 60.3 | 81.1 | 61.8 | 60.3 |
| | | LRS | **63.2** | **65.8** | **71.2** | **60.3** | **62.6** | **82.3** | **64.5** | **62.9** |
| | SI-NI-FGSM | W/O | 74.7 | 71.4 | 68.9 | 58.1 | 76.3 | 87.3 | 61.6 | 66.2 |
| | | SVD | 74.3 | 70.5 | 67.1 | 56.4 | 73.3 | 86.8 | 60.1 | 63.4 |
| | | LRS | **79.4** | **76.3** | **72.8** | **64.0** | **79.5** | **89.9** | **66.2** | **71.6** |
| | VMI-FGSM | W/O | 73.6 | 71.6 | 69.8 | 61.5 | 77.8 | 86.0 | 64.1 | 67.4 |
| | | SVD | 76.1 | 74.1 | 69.6 | 62.0 | 76.2 | 86.2 | 63.7 | 67.8 |
| | | LRS | **78.5** | **75.9** | **72.8** | **65.7** | **80.8** | **88.2** | **67.0** | **73.2** |
| | GI-FGSM | W/O | 66.0 | 61.2 | 60.0 | 46.6 | 66.0 | 83.2 | 53.8 | 56.2 |
| | | SVD | 65.1 | 60.2 | 59.2 | 43.9 | 64.5 | 82.9 | 54.1 | 53.1 |
| | | LRS | **73.1** | **70.5** | **63.7** | **53.9** | **75.6** | **87.4** | **58.8** | **63.4** |

*Table 13.* Complete per-model ASR (%) of different attacks evaluated on various ViT target models and defended ViT target models using CNN-based surrogate models. Best results are highlighted in bold.

| Model | Attack | Method | PiT-S | Visformer-S | Twins-B | CaiT-S | DeiT-B | ConViT-B | ViT-B | Swin-S | DeiT-S(adv) | Swin-B(adv) | XCiT-S(adv) |
|---|---|---|---|---|---|---|---|---|---|---|---|---|---|
| Inc-v3 | MI-FGSM | W/O | 25.7 | 23.5 | 22.3 | 16.5 | 17.9 | 16.4 | 24.8 | 20.7 | 26.9 | 17.8 | 59.4 |
| | | SVD | 23.8 | 25.8 | 22.2 | 17.3 | 18.8 | 17.3 | 24.7 | 20.8 | **27.4** | 17.6 | 59.6 |
| | | LRS | **28.4** | **28.7** | **26.5** | **19.6** | **21.0** | **19.4** | **26.5** | **23.9** | 27.1 | **18.6** | **60.3** |
| | DI-FGSM | W/O | 17.1 | 16.3 | 17.2 | 10.6 | 12.2 | 10.3 | 18.1 | 15.7 | 23.6 | 15.8 | 56.6 |
| | | SVD | 17.2 | 18.2 | 16.1 | 11.1 | 11.6 | 11.5 | 18.4 | 15.8 | **23.9** | **16.1** | **56.9** |
| | | LRS | **18.6** | **19.2** | **19.0** | **13.5** | **12.6** | **12.3** | **18.6** | **17.3** | 23.8 | 16.0 | 56.7 |
| | TI-FGSM | W/O | 13.7 | 14.2 | 11.9 | 9.3 | 10.9 | 9.8 | 18.1 | 12.6 | 23.5 | 15.8 | 56.3 |
| | | SVD | 14.2 | 13.7 | 11.1 | 9.5 | 12.3 | 9.7 | 19.3 | 12.7 | 24.3 | 15.9 | **56.6** |
| | | LRS | **15.8** | **15.5** | **14.7** | **10.4** | **12.7** | **10.6** | 19.2 | **14.2** | **24.4** | **16.1** | 56.5 |
| | PI-FGSM | W/O | 22.7 | 22.2 | 16.9 | **17.3** | 18.5 | 17.0 | 36.0 | 14.8 | 41.2 | 28.4 | 66.7 |
| | | SVD | 22.8 | 22.7 | 16.6 | **17.3** | **19.0** | 16.3 | **38.2** | 15.3 | **42.2** | 28.6 | **68.0** |
| | | LRS | **23.5** | **24.7** | **17.7** | 17.1 | 17.1 | **16.5** | 36.1 | **16.5** | 42.0 | **30.0** | 67.8 |
| | SI-NI-FGSM | W/O | 39.7 | 41.5 | 36.3 | 31.2 | 27.2 | 27.5 | 34.4 | 32.4 | **30.5** | **19.8** | **62.2** |
| | | SVD | 41.3 | 43.5 | 38.0 | 31.0 | 28.2 | 27.3 | **36.6** | 34.3 | 29.7 | 19.5 | 62.0 |
| | | LRS | **43.4** | **46.9** | **40.2** | **34.1** | **28.5** | **30.4** | 36.1 | **36.5** | 30.2 | 19.6 | 62.1 |
| | VMI-FGSM | W/O | 37.8 | 40.9 | 37.1 | 61.1 | 27.7 | 28.1 | 31.6 | 35.4 | 28.3 | 19.0 | **61.1** |
| | | SVD | 41.4 | 44.8 | 41.6 | 60.7 | 30.6 | 27.6 | 34.6 | 37.3 | 28.4 | 19.7 | 60.7 |
| | | LRS | **43.4** | **45.5** | **43.8** | 60.7 | **30.8** | **30.5** | **34.9** | **38.8** | **29.5** | **20.1** | 60.7 |
| | GI-FGSM | W/O | 27.9 | 28.8 | 27.2 | 19.4 | 20.6 | 18.9 | 26.3 | 23.7 | 28.5 | 18.4 | 60.8 |
| | | SVD | 28.6 | 30.8 | 28.2 | 22.2 | 21.4 | 19.6 | 29.3 | 27.0 | 28.1 | 18.6 | **61.1** |
| | | LRS | **31.6** | **34.8** | **31.3** | **25.2** | **22.4** | **23.1** | **30.4** | **27.9** | **28.9** | **19.4** | 60.8 |
| Vgg-19 | MI-FGSM | W/O | 38.7 | 51.4 | 46.8 | 32.5 | 29.5 | 27.9 | 34.0 | 39.3 | 37.8 | 29.8 | 72.4 |
| | | SVD | 35.5 | 48.5 | 43.7 | 29.5 | 27.1 | 26.6 | 34.4 | 36.1 | 38.0 | **30.1** | 72.4 |
| | | LRS | **41.7** | **56.3** | **50.7** | **33.8** | **30.2** | **29.0** | **36.7** | **41.4** | **38.3** | 29.9 | **72.7** |
| | DI-FGSM | W/O | 27.2 | 36.8 | 34.2 | 20.0 | 18.6 | 18.0 | 24.3 | 28.1 | 31.9 | **23.7** | 65.6 |
| | | SVD | 25.7 | 35.4 | 31.4 | 18.3 | 16.2 | 15.9 | **25.2** | 27.0 | **32.1** | **23.7** | 65.8 |
| | | LRS | **29.5** | **40.7** | **37.0** | **21.2** | **19.0** | **18.3** | 24.8 | **31.2** | 31.8 | 23.2 | **66.0** |
| | TI-FGSM | W/O | 25.0 | 30.9 | 26.6 | 19.1 | 19.2 | 17.3 | 25.1 | 22.3 | 32.1 | 23.7 | 65.6 |
| | | SVD | 23.7 | 27.2 | 23.7 | 18.2 | 16.8 | 17.3 | 25.7 | 19.9 | 32.4 | **24.0** | 66.1 |
| | | LRS | **27.1** | **33.9** | **31.5** | **21.0** | **20.8** | **19.7** | **27.6** | **24.4** | **32.5** | 23.5 | 65.7 |
| | PI-FGSM | W/O | 24.2 | **26.1** | 20.8 | 18.5 | 17.8 | **19.5** | 41.4 | 17.6 | 46.5 | 38.8 | 73.7 |
| | | SVD | 24.5 | 25.6 | 21.2 | 18.2 | 18.0 | 18.2 | **41.7** | **17.7** | 46.8 | **40.1** | **74.1** |
| | | LRS | **25.7** | 26.0 | **21.8** | **19.7** | **18.1** | 17.6 | 41.6 | **17.7** | **47.0** | 39.1 | 73.9 |
| | SI-NI-FGSM | W/O | 49.9 | 67.6 | 58.1 | 42.8 | 37.1 | 35.6 | 44.4 | 47.9 | 40.3 | 31.1 | 72.8 |
| | | SVD | 49.1 | 65.7 | 56.6 | 41.6 | 35.0 | 36.5 | 45.7 | 47.9 | **40.5** | 31.3 | **73.1** |
| | | LRS | **54.1** | **71.4** | **62.9** | **44.4** | **38.2** | **38.1** | **46.0** | **53.0** | **40.5** | **31.4** | 72.9 |
| | VMI-FGSM | W/O | 54.0 | 66.5 | 61.3 | 46.1 | 39.9 | 39.3 | 44.4 | 53.3 | 39.4 | 31.3 | 73.2 |
| | | SVD | 52.2 | 63.6 | 60.1 | 45.4 | 38.5 | 38.6 | 44.1 | 53.4 | 38.6 | 31.4 | 72.8 |
| | | LRS | **57.4** | **68.0** | **65.0** | **47.5** | **41.6** | **40.3** | **44.6** | **55.9** | **39.5** | **31.7** | 73.0 |
| | GI-FGSM | W/O | 41.0 | 54.7 | 49.4 | 34.5 | 29.4 | 29.5 | 34.5 | 39.5 | 35.8 | 28.3 | 68.8 |
| | | SVD | 42.8 | 56.4 | 47.3 | 33.8 | 29.7 | 27.1 | **36.6** | 39.1 | 35.8 | 28.8 | 68.8 |
| | | LRS | **43.8** | **57.0** | **51.6** | **35.6** | **30.5** | **29.9** | 36.3 | **42.4** | **36.0** | **29.0** | **68.9** |
| Densenet-121 | MI-FGSM | W/O | 44.2 | 53.1 | 48.5 | 36.5 | 34.6 | 31.5 | 37.6 | 39.2 | 31.3 | 21.3 | 63.9 |
| | | SVD | 41.6 | 49.9 | 42.2 | 33.4 | 32.7 | 30.4 | 35.5 | 36.6 | 31.7 | 21.8 | 64.3 |
| | | LRS | **52.5** | **60.1** | **53.1** | **42.5** | **39.2** | **36.9** | **40.3** | **44.4** | **32.2** | **21.9** | **64.5** |
| | DI-FGSM | W/O | 38.1 | 46.0 | 39.8 | 29.9 | **27.9** | 24.7 | 29.4 | 31.5 | 33.0 | 24.4 | **66.4** |
| | | SVD | 35.4 | 40.9 | 33.5 | 26.6 | 24.9 | 23.1 | 28.9 | 28.1 | **33.2** | 24.2 | 66.0 |
| | | LRS | **42.5** | **51.9** | **42.5** | **33.5** | 27.5 | **27.9** | **30.9** | **35.1** | 32.9 | **24.7** | 66.3 |
| | TI-FGSM | W/O | 28.7 | 34.4 | 28.0 | 21.7 | 22.7 | 21.8 | 28.0 | 22.9 | 32.8 | 24.3 | 66.1 |
| | | SVD | 25.2 | 29.2 | 22.7 | 20.0 | 20.6 | 20.3 | 27.5 | 20.3 | 32.6 | 24.0 | 65.6 |
| | | LRS | **33.2** | **37.1** | **30.8** | **26.7** | **25.1** | **23.8** | **29.7** | **26.7** | **33.5** | **24.5** | **66.2** |
| | PI-FGSM | W/O | 28.1 | 30.1 | 22.7 | 21.7 | 19.6 | **21.1** | 48.5 | **19.5** | 49.4 | 42.8 | 75.0 |
| | | SVD | 26.1 | 28.0 | 21.5 | 19.9 | 18.8 | 20.3 | 48.3 | 19.0 | 48.5 | 42.3 | 74.8 |
| | | LRS | **30.6** | **31.2** | **23.0** | **22.3** | **20.9** | **21.1** | **49.9** | 19.2 | **49.6** | **43.1** | **75.2** |
| | SI-NI-FGSM | W/O | 61.3 | 69.9 | 61.6 | 55.8 | 49.2 | 47.0 | 51.4 | 54.9 | 42.4 | 32.4 | 72.8 |
| | | SVD | 59.2 | 67.6 | 55.7 | 52.1 | 45.4 | 42.5 | 53.6 | 51.0 | 41.5 | 32.6 | 72.9 |
| | | LRS | **66.3** | **76.0** | **67.1** | **60.3** | **52.4** | **50.5** | **53.7** | **56.6** | **43.1** | **33.0** | **73.4** |
| | VMI-FGSM | W/O | 67.6 | 74.6 | 68.4 | 59.9 | 54.3 | 53.2 | 53.7 | 59.3 | 41.7 | 33.2 | 73.2 |
| | | SVD | 67.7 | 74.1 | 66.8 | 58.3 | 53.5 | 51.9 | 53.3 | 59.4 | 41.6 | 33.1 | 72.8 |
| | | LRS | **71.6** | **78.0** | **70.7** | **63.8** | **58.7** | **55.8** | **55.3** | **63.7** | **43.2** | **33.8** | **73.6** |
| | GI-FGSM | W/O | 55.8 | 66.1 | 55.6 | 47.8 | 40.3 | 39.3 | 44.6 | 47.7 | 38.5 | 29.4 | 68.8 |
| | | SVD | 52.0 | 62.6 | 52.5 | 44.6 | 38.7 | 39.4 | 45.2 | 46.0 | 38.5 | 29.2 | **69.4** |
| | | LRS | **62.3** | **71.9** | **62.3** | **53.8** | **47.3** | **43.7** | **49.0** | **54.4** | **39.1** | **29.8** | 69.0 |

*Table 14.* Complete per-model ASR (%) of different attacks evaluated on various CNN target models using ViT-based surrogate models. Best results are highlighted in bold.

| Model | Attack | Method | Inc-v3 | Inc-v4 | Vgg-16 | Vgg-19 | Dense-121 | IncRes-v2 | Res152 | Res50 | Res101 |
|---|---|---|---|---|---|---|---|---|---|---|---|
| ViT-B | MIM | W/O | 45.1 | 41.1 | 66.5 | 64.2 | 54.5 | 34.8 | 45.4 | 57.0 | 51.5 |
| | | SVD | 48.2 | 44.1 | 66.9 | 67.2 | 58.4 | 36.7 | 47.9 | 58.1 | 55.0 |
| | | LRS | **53.5** | **47.0** | **72.1** | **70.3** | **61.8** | **40.2** | **51.1** | **63.9** | **59.5** |
| | PNA | W/O | 49.5 | 42.4 | 64.4 | 61.1 | 59.0 | 37.5 | 48.6 | 57.8 | 53.9 |
| | | SVD | 49.4 | 43.9 | 65.6 | 62.7 | 58.3 | 38.3 | 49.8 | 58.9 | 55.5 |
| | | LRS | **54.8** | **49.8** | **73.5** | **68.0** | **65.6** | **43.1** | **55.3** | **66.3** | **61.8** |
| | TGR | W/O | 53.4 | 47.8 | 73.2 | 71.4 | 64.0 | 42.6 | 54.3 | 66.1 | 62.6 |
| | | SVD | 54.2 | 48.0 | 74.9 | 71.6 | **66.0** | **42.7** | 54.6 | 66.2 | 62.6 |
| | | LRS | **54.9** | **48.5** | **77.4** | **74.9** | 64.8 | 40.3 | **56.1** | **68.9** | **64.3** |
| | ATT | W/O | 59.8 | 54.7 | 76.8 | 75.4 | 70.1 | 49.2 | 60.0 | 70.8 | 68.1 |
| | | SVD | 60.7 | **56.5** | 78.1 | 76.2 | **72.3** | **49.3** | 60.7 | 71.2 | 67.3 |
| | | LRS | **61.0** | 54.2 | **80.4** | **77.1** | 71.8 | 48.8 | **60.9** | **73.3** | **69.2** |
| DeIT-B | MIM | W/O | 56.1 | 49.5 | 74.5 | 73.1 | 65.3 | 43.2 | 54.0 | 69.5 | 63.2 |
| | | SVD | 58.5 | 53.2 | 74.9 | 72.3 | 68.3 | 45.9 | 57.4 | 69.0 | 65.8 |
| | | LRS | **63.8** | **56.2** | **78.9** | **76.3** | **75.7** | **52.2** | **64.5** | **73.0** | **72.5** |
| | PNA | W/O | 59.8 | 54.7 | 73.3 | 69.1 | 72.0 | 48.7 | 61.3 | 70.0 | 66.2 |
| | | SVD | 62.2 | 57.1 | 75.4 | 72.1 | 75.5 | 51.3 | 65.0 | 72.8 | 70.2 |
| | | LRS | **66.0** | **62.2** | **78.6** | **75.5** | **80.2** | **56.9** | **70.3** | **77.2** | **73.5** |
| | TGR | W/O | 73.2 | 67.6 | 87.1 | 84.8 | 86.2 | 64.9 | 77.0 | 84.7 | 83.9 |
| | | SVD | 73.9 | 69.3 | 88.2 | 85.3 | 86.4 | 64.8 | 78.1 | 85.9 | 84.5 |
| | | LRS | **76.1** | **71.5** | **90.5** | **88.6** | **88.2** | **67.7** | **81.4** | **88.8** | **86.7** |
| | ATT | W/O | 76.2 | 72.3 | 88.7 | 86.3 | 87.4 | 69.2 | 80.5 | 86.6 | 85.6 |
| | | SVD | 77.1 | 71.2 | 88.6 | 86.4 | 87.4 | 68.7 | 79.5 | 86.4 | 84.6 |
| | | LRS | **77.8** | **74.3** | **90.7** | **88.8** | **90.1** | **70.1** | **82.1** | **89.1** | **87.8** |
| CaiT-S | MIM | W/O | 64.2 | 59.5 | 80.0 | 78.1 | 72.2 | 53.6 | 64.3 | 73.6 | 71.1 |
| | | SVD | 66.2 | 60.9 | 80.8 | 79.4 | 75.1 | 57.1 | 66.1 | 75.3 | 74.1 |
| | | LRS | **70.6** | **67.0** | **82.4** | **81.4** | **80.4** | **63.6** | **70.5** | **79.0** | **76.4** |
| | PNA | W/O | 59.9 | 56.2 | 74.9 | 73.9 | 70.5 | 53.0 | 62.0 | 68.0 | 67.7 |
| | | SVD | 64.8 | 60.6 | 75.7 | 74.8 | 74.3 | 57.2 | 65.2 | 72.5 | 69.5 |
| | | LRS | **70.0** | **67.3** | **80.1** | **80.0** | **78.5** | **63.6** | **72.1** | **76.6** | **74.8** |
| | TGR | W/O | 76.0 | 70.8 | 88.0 | 86.9 | 86.6 | 69.0 | 78.2 | 84.6 | 83.5 |
| | | SVD | 74.9 | 71.1 | 88.2 | 86.6 | 86.0 | 70.2 | 77.7 | 84.0 | 83.9 |
| | | LRS | **78.8** | **74.6** | **90.4** | **89.2** | **88.0** | **71.5** | **81.5** | **86.6** | **86.9** |
| | ATT | W/O | 78.2 | 75.8 | 86.4 | 85.7 | 84.9 | 73.2 | 79.7 | 84.3 | 83.5 |
| | | SVD | 78.9 | 75.9 | 88.1 | 86.6 | 85.0 | 73.6 | 80.2 | **85.9** | 84.1 |
| | | LRS | **79.7** | **76.6** | **89.2** | **87.8** | **87.9** | **74.2** | **81.6** | 85.7 | **84.5** |

*Table 15.* Complete per-model ASR (%) of different attacks evaluated on various CNN target models and defense methods using ViT-based surrogate models. Best results are highlighted in bold.

| Model | Attack | Method | Inc-v3$_{adv}$ | Inc-v3$_{ens3}$ | Inc-v3$_{ens4}$ | IncRes-v2$_{ens}$ | HGD | JPEG | R&P | NIPS_r3 |
|---|---|---|---|---|---|---|---|---|---|---|
| ViT-B | MIM | W/O | 37.6 | 35.2 | 35.6 | 29.2 | 28.2 | 53.5 | 34.4 | 34.6 |
| | | SVD | 39.3 | 37.3 | 36.8 | 29.9 | 31.7 | 55.1 | 37.1 | 36.6 |
| | | LRS | **42.9** | **40.9** | **39.4** | **31.3** | **33.3** | **58.7** | **38.8** | **38.9** |
| | PNA | W/O | 37.9 | 34.1 | 35.9 | 26.1 | 30.4 | 54.6 | 33.5 | 34.4 |
| | | SVD | 39.6 | 37.4 | 36.1 | 26.8 | 32.1 | 54.8 | 34.9 | 34.2 |
| | | LRS | **44.2** | **38.9** | **40.4** | **29.5** | **34.3** | **60.3** | **37.3** | **38.6** |
| | TGR | W/O | 45.2 | 41.3 | 42.7 | 32.9 | **39.3** | 62.1 | 41.8 | 42.2 |
| | | SVD | 45.6 | 41.3 | **42.8** | **33.0** | 39.2 | 62.2 | 42.1 | 42.0 |
| | | LRS | **47.0** | **42.4** | 41.7 | 32.3 | **39.3** | **65.7** | **42.7** | **43.1** |
| | ATT | W/O | 49.9 | **47.5** | **48.4** | **39.0** | 46.5 | 66.4 | 46.4 | 45.9 |
| | | SVD | **51.5** | 47.0 | 47.5 | 38.2 | 44.9 | 66.9 | 46.6 | 46.5 |
| | | LRS | 50.5 | 46.3 | 47.5 | 35.7 | **46.9** | **67.0** | **46.8** | **47.2** |
| DeIT-B | MIM | W/O | 45.8 | 41.5 | 40.3 | 35.2 | 38.6 | 61.3 | 40.6 | 38.2 |
| | | SVD | 48.7 | 43.9 | 44.0 | 35.3 | 40.3 | 62.8 | 41.0 | 41.0 |
| | | LRS | **53.9** | **48.6** | **49.1** | **41.0** | **47.6** | **68.1** | **46.9** | **46.2** |
| | PNA | W/O | 48.9 | 44.4 | 43.6 | 34.1 | 43.8 | 99.5 | 99.1 | 99.1 |
| | | SVD | 51.4 | 48.2 | 47.6 | 36.1 | 45.7 | 99.8 | 99.2 | 98.9 |
| | | LRS | **57.6** | **52.5** | **50.4** | **40.0** | **52.4** | **99.9** | **99.5** | **99.2** |
| | TGR | W/O | 66.1 | 59.1 | 59.4 | 50.7 | 62.2 | 79.7 | 57.6 | 59.0 |
| | | SVD | 67.6 | 62.1 | 60.8 | 51.2 | 63.2 | 80.6 | 59.0 | 60.0 |
| | | LRS | **69.5** | **63.8** | **63.9** | **55.0** | **67.3** | **82.3** | **61.6** | **64.2** |
| | ATT | W/O | 69.4 | 65.1 | 64.7 | 55.4 | 66.1 | 81.0 | 61.3 | 62.6 |
| | | SVD | 69.3 | 63.5 | 63.1 | 53.6 | 66.0 | 80.3 | 60.4 | 60.6 |
| | | LRS | **71.0** | **66.3** | **66.6** | **56.8** | **69.4** | **83.7** | **64.4** | **64.9** |
| CaiT-S | MIM | W/O | 52.9 | 48.2 | 45.8 | 41.1 | 47.0 | 63.9 | 46.3 | 44.1 |
| | | SVD | 53.8 | 50.9 | 49.8 | 42.6 | 50.6 | 66.9 | 48.7 | 47.3 |
| | | LRS | **62.2** | **57.6** | **55.3** | **48.8** | **57.7** | **72.1** | **55.4** | **54.5** |
| | PNA | W/O | 48.6 | 44.1 | 43.7 | 35.9 | 43.4 | 60.5 | 40.1 | 39.3 |
| | | SVD | 52.5 | 49.5 | 47.6 | 41.3 | 47.9 | 64.6 | 44.1 | 43.7 |
| | | LRS | **60.0** | **55.8** | **54.2** | **45.4** | **54.8** | **70.6** | **50.5** | **52.5** |
| | TGR | W/O | 66.6 | 62.6 | 61.5 | 52.8 | 61.0 | 79.8 | 59.4 | 58.8 |
| | | SVD | 67.2 | 63.8 | 62.9 | 54.9 | 62.8 | 79.7 | 61.3 | 59.8 |
| | | LRS | **71.1** | **66.4** | **67.8** | **57.5** | **68.3** | **82.6** | **65.5** | **67.9** |
| | ATT | W/O | 70.1 | 68.2 | 66.7 | **59.9** | 65.6 | 81.5 | 64.6 | 64.8 |
| | | SVD | 70.2 | 67.5 | 67.7 | 59.0 | 67.6 | 80.8 | 64.4 | 65.6 |
| | | LRS | **71.1** | **68.5** | **68.1** | 59.4 | **68.2** | **82.1** | **66.0** | **66.5** |

*Table 16.* Complete per-model ASR (%) of different attacks evaluated on various ViT target models and defended ViT target models using ViT-based surrogate models. Best results are highlighted in bold.

| Model | Attack | Method | PiT-S | CaiT-S | DeiT-B | ViT-B | Visformer-S | ConViT-B | Twins-B | DeiT-Sadv | Swin-Badv | XCiT-Sadv |
|---|---|---|---|---|---|---|---|---|---|---|---|---|
| ViT-B | MIM | W/O | 47.9 | 68.7 | 74.2 | **100.0** | 46.0 | 71.0 | 41.4 | 41.9 | 30.8 | 71.2 |
| | | SVD | 51.7 | 74.9 | 78.8 | **100.0** | 50.8 | 76.3 | 44.8 | 42.0 | 32.1 | 71.4 |
| | | LRS | **55.2** | **77.3** | **79.9** | **100.0** | **51.9** | **78.9** | **45.6** | **43.1** | **33.0** | **72.0** |
| | PNA | W/O | 53.8 | 76.7 | 77.9 | 99.4 | 52.2 | 76.6 | 46.6 | 38.6 | 28.9 | 69.8 |
| | | SVD | 56.0 | 78.9 | 80.6 | 99.7 | 53.9 | **80.7** | **48.8** | 39.0 | 29.8 | 69.9 |
| | | LRS | **57.7** | **79.5** | **81.9** | **100.0** | **55.4** | 79.8 | 48.2 | **41.0** | **30.4** | **70.3** |
| | TGR | W/O | 56.3 | 71.7 | 78.2 | **99.9** | 54.4 | 75.3 | 46.7 | 45.5 | 34.3 | 73.5 |
| | | SVD | 57.9 | **72.1** | 78.4 | **99.9** | **55.4** | **76.7** | 47.8 | 46.8 | 34.9 | 73.5 |
| | | LRS | **58.0** | 70.8 | 75.7 | **99.9** | 54.1 | 75.9 | 47.1 | **47.9** | **35.6** | **74.5** |
| | ATT | W/O | 67.3 | 81.8 | 85.2 | 99.9 | **64.7** | **83.0** | 57.1 | 46.3 | 35.5 | 73.5 |
| | | SVD | 65.8 | **81.3** | 84.6 | 99.9 | 64.6 | **83.0** | 56.4 | 47.9 | 35.7 | 73.2 |
| | | LRS | **68.8** | 79.7 | **85.7** | **100.0** | 64.2 | 81.5 | **58.3** | **48.8** | **36.2** | **74.3** |
| DeIT-B | MIM | W/O | 67.7 | 90.6 | **100.0** | 70.8 | 65.1 | 92.1 | 59.5 | 42.3 | 32.5 | 71.7 |
| | | SVD | 72.4 | 92.8 | **100.0** | 75.3 | 68.5 | 94.4 | 64.7 | 43.6 | 32.9 | 71.8 |
| | | LRS | **81.8** | **94.6** | **100.0** | **84.1** | **77.5** | **95.8** | **72.2** | **44.8** | **34.5** | **72.7** |
| | PNA | W/O | 97.7 | 99.6 | **100.0** | 99.5 | 97.0 | 99.3 | 97.8 | 96.4 | 99.3 | **99.7** |
| | | SVD | 97.8 | 99.2 | 99.9 | 99.6 | 97.7 | **99.7** | 98.4 | **97.3** | **99.4** | **99.7** |
| | | LRS | **98.7** | **99.8** | **100.0** | **99.7** | **98.7** | **99.7** | **99.5** | **97.3** | **99.4** | **99.7** |
| | TGR | W/O | 90.3 | 98.6 | **100.0** | 94.2 | 86.2 | 99.0 | 80.5 | 51.0 | 37.9 | 75.1 |
| | | SVD | 91.1 | 98.9 | **100.0** | 94.2 | 87.2 | **98.9** | 81.7 | 50.8 | 39.0 | 75.3 |
| | | LRS | **91.9** | **99.1** | **100.0** | **95.3** | **89.0** | **98.9** | **84.2** | **53.5** | **41.1** | **75.5** |
| | ATT | W/O | 94.0 | 99.3 | **100.0** | 94.5 | 91.2 | **99.5** | 87.7 | 50.4 | 38.6 | 74.9 |
| | | SVD | 94.2 | 99.2 | **100.0** | 95.1 | 91.1 | 99.4 | 87.5 | 50.1 | 37.8 | 74.3 |
| | | LRS | **94.5** | 98.7 | **100.0** | **96.0** | **92.7** | **99.5** | **89.4** | **52.6** | **40.8** | **75.2** |
| CaiT-S | MIM | W/O | 74.6 | **100.0** | 94.9 | 75.6 | 71.9 | 93.0 | 72.9 | 42.8 | 34.0 | 71.5 |
| | | SVD | 80.2 | **100.0** | 95.5 | 80.0 | 73.9 | 94.7 | 77.1 | 44.0 | 35.1 | 72.2 |
| | | LRS | **84.1** | 99.6 | **95.9** | **84.1** | **78.9** | **95.5** | **80.8** | **47.4** | **36.4** | **73.3** |
| | PNA | W/O | 74.9 | 96.9 | 88.4 | 74.8 | 71.8 | 88.3 | 75.0 | 37.4 | 29.9 | 69.2 |
| | | SVD | 81.4 | **97.5** | 92.0 | 78.1 | 76.8 | 91.8 | 80.8 | 38.8 | 30.6 | 69.2 |
| | | LRS | **84.5** | 96.8 | **92.6** | **83.2** | **79.8** | **92.4** | **82.6** | **42.8** | **33.7** | **70.7** |
| | TGR | W/O | 87.5 | 99.7 | 95.8 | 89.1 | 83.7 | 95.1 | 82.5 | 49.1 | 39.0 | 74.1 |
| | | SVD | 87.8 | **100.0** | **95.9** | 89.2 | 84.1 | 94.7 | 83.8 | 51.5 | 40.2 | 74.5 |
| | | LRS | **88.5** | 99.7 | 95.6 | **92.7** | **84.8** | **95.5** | **85.0** | **54.2** | **43.4** | **76.3** |
| | ATT | W/O | 89.0 | 97.9 | 95.4 | 89.3 | 85.8 | 94.5 | 87.4 | 49.3 | 38.9 | 73.3 |
| | | SVD | **90.3** | **99.0** | 95.5 | 89.5 | 86.0 | **95.2** | 87.2 | 51.1 | 39.9 | 74.3 |
| | | LRS | 89.1 | **99.0** | **95.8** | **90.7** | **87.0** | 93.9 | **88.5** | **53.4** | **42.5** | **74.8** |

