# OpenReview forum: "Low-Rank and Sparsity Are All You Need: Exploring Robust Hierarchical Latent Subspaces for Transferable Adversarial Attack"
_ICML.cc/2026/Conference — ICML 2026 regular_

### Official Review · Reviewer_Msug · 2026-03-03

**Soundness:** 2
**Presentation:** 2
**Significance:** 2
**Originality:** 2
**Overall Recommendation:** 3
**Confidence:** 4

**Summary:**

The authors propose a low-rank and sparse feature-space at-tack that explicitly decomposes intermediate representations into global semantic components and localized discriminative perturbations. Experiments across CNN and Vision Transformer architectures demonstrate that the proposed method outperforms state-of-the-art attacks in black-box transfer settings.

**Compliance With Llm Reviewing Policy:**

Affirmed.

**Final Justification:**

My concerns have been partially resolved, but the concern on novelty still remains. Thus, I will raise my score to 3.

**Key Questions For Authors:**

The questions are listed in the Weaknesses part.

**Limitations:**

yes

**Strengths And Weaknesses:**

**Strengths**
1. The methodology part is easy to understand
2. The experimental results look good
3. The experiments are extensive

**Weaknesses**
1. The low-rank and sparsity ideas are not novel. This work only combines them together to improve transferable adversarial attack. The insight of using these two techniques is not very clear.
2. As for Hierarchical Mixture of Robust Low-Rank and Sparse Experts, why this strategy? The insight is not clear. Lack of ablation study and theoretical / numerical analysis.
3. Lack of baselines. You should also consider the methods only using low-rank / sparsity.  Not just different attack methods.
4. Insufficient ablation study. This method introduces many hyperparameters, such as layer indices for experts, number of rank, sparsity level, which should be considered in the ablation study
4. The presentation of tables and figures can be improved to make them more readable.

---

> ### Author Rebuttal · Authors · 2026-03-31
>
> We thank the reviewer for the insightful comment.
>
> **Question 1: On the Insight of Combining Low-Rank and Sparse Modeling**
>
> **(1) Novelty:**
> Regarding novelty, our method differs from traditional sparse attacks, which primarily focus on perturbation design at the input level and are typically aimed at improving imperceptibility,
> and also differs from mainstream attack methods that are constrained to operate within dense model feature spaces，
> we are the first to introduce a low-rank–sparse structured feature space into adversarial attacks.
> This redefines the role of structured decomposition in adversarial optimization and provides a new perspective for understanding transferable perturbations.
>
> We further propose a fast alternating update mechanism (WALA).
> By leveraging a warm-start strategy based on the continuity of representation evolution in iterative attacks,
> WALA improves optimization stability while balancing attack success rate and efficiency.
> Unlike traditional decomposition methods that solve each iteration independently, our approach better matches the dynamic nature of adversarial optimization.
>
> **(2) Insight:**
> The key insight of combining low-rank and sparse modeling is their complementarity in capturing adversarial vulnerability:
> the low-rank component encodes model-agnostic global semantic structure, improving generalization,
> while the sparse component captures local and highly sensitive discriminative directions,
> enhancing decision boundary effects. Their combination balances global consistency and local effectiveness, improving transferability.
>
> We validate this via attention rollout visualization
> (see [https://anonymous.4open.science/r/LRS_attention_rollout-DC20/attention_rollout.png](https://anonymous.4open.science/r/LRS_attention_rollout-DC20/attention_rollout.png)).
> The results show a clear functional separation: the low-rank component captures coarse semantic structure, while the sparse component highlights key local regions.
> This indicates that LRS is not a simple additive formulation, but a more expressive and robust structured attack space.
>
> **Question 2: Why Hierarchical Experts**
>
> We clarify that the motivation behind the proposed hierarchical Mixture-of-Experts strategy stems from the heterogeneity of feature representations across different layers.
>
> Shallow layers capture local patterns, while deeper layers encode more abstract semantic information. Therefore, applying a uniform constraint across all layers is not optimal.
>
> Low-rank and sparse components are better suited to different layer characteristics. Their complementarity enables a more comprehensive modeling of structural properties in features.
>
> Based on this observation, we design a hierarchical expert strategy to match the characteristics of each layer.
>
> In addition, ablation studies on the expert strategy can be found in our response to Reviewer 8dtN (Question 2).
>
>
>
> **Question 3: Baselines**
>
> We clarify that our method is not an isolated attack, but a general structured module that can be seamlessly integrated into various iterative attack frameworks.
>
> Our main baselines already cover both classical dense perturbation methods and existing structured approaches,
> including dense attacks (e.g., MI-FGSM and its variants) and representative low-rank attacks (e.g., SVD).
> We evaluate integrations with gradient-based methods (e.g., PGN, GGS) and feature-level attacks (e.g., BFA, P2FA) in the appendix,
> further demonstrating the generality of our approach.
>
> It is worth noting that existing sparse methods are typically input-level,
> formulation-specific attacks rather than modular designs, and thus are not directly comparable in the same sense.
> Rather than serving as a replacement for these methods,
> our approach can be viewed as a unified framework that subsumes them as special cases (e.g., reducing to pure low-rank modeling when the sparse component is removed).
>
>
> **Question 4: The ablation study**
>
> Regarding expert layer index selection, we do not randomly choose layer indices.
> The detailed ablation study in Appendix Figure 6 validates the rationale behind our design.
> By evaluating different expert configurations across shallow, intermediate, and deep layers,
> we find that the best attack success rate is achieved when the layer-wise selection follows the evolution from local details to global semantics,
> demonstrating the robustness of our hierarchical expert strategy.
>
> Regarding the rank ratio and sparsity level,
> our method models their dynamic interaction rather than treating them as fixed hyperparameters.
> Detailed analysis can be found in our response to Reviewer S5WP (Question 3).
>
> **Question 5: On the Presentation of Tables and Figures**
>
> We believe the current figures already clearly illustrate the core mechanism and workflow of our method.
> We have further refined them to improve readability, including more compact layouts, clearer structure, and additional annotations.

---

> > ### Author Rebuttal · Reviewer_Msug · 2026-04-02
> >
> > Thanks for the rebuttal. My concerns have been partially resolved, but the concern on novelty still remains. Thus, I will raise my score to 3.

---

> > > ### Author Response · Authors · 2026-04-05
> > >
> > > We thank the reviewer for the constructive feedback and insightful questions regarding the novelty of our work.
> > >
> > > We would like to emphasize that while the robust PCA formulation is well established in the literatrue, its direct use is not the primary novelty of our work.
> > > Instead, our key contribution lies in **systematically rethinking adversarial attack modeling through structured latent subspace design**, which is **architecture-agnostic** (e.g., CNNs, ViTs) and **broadly applicable across different attack paradigms** (e.g., gradient-based, input transformation-based, and feature-based methods).
> > >
> > > In this regard, our work is conceptually aligned with recent advances in large language models, where classical techniques such as low-rank adaptation (e.g., LoRA) and sparsity are revisited not as standalone novelties, but as principled tools for structured and efficient representation learning, such as compression and pruning.
> > > Similarly, we do not claim novelty in low-rank or sparse modeling itself; rather, we introduce a **novel integration of these structures into adversarial attack modeling**, tailored to address black-box transferability limitations.
> > >
> > > Concretely, we propose a unified **hirarchical structured decomposition framework** that enables structured compression of multi-layer surrogate features and reconstruction of adversarial directions within a constrained latent subspace, leading to more informative and transferable perturbations than conventional dense and unstructured formulations.
> > >
> > > Building on this perspective, our method systematically contrasts with existing approaches along several key dimensions, including:
> > >
> > >
> > > **(1) Difference in Latent Subspace Modeling: Unstructured Dense vs. Structured Low-rank + Sparse**
> > >
> > > - **Existing methods:**
> > > Existing approaches perform constrained optimization in the input space or at a single-layer latent level, while operating in an unconstrained high-dimensional dense latent space without structural regularization.
> > > Such formulations tend to capture model-specific noise patterns, leading to overfitting and limited transferability.
> > >
> > > - **Our method:** To the best of our knowledge, we are the first to introduce a hierarchical structured latent subspace by jointly modeling low-rank and sparse components, leveraging their complementary roles in capturing global semantics and localized discriminative patterns.
> > > This design enables cross-layer feature compression and reconstruction within a shared latent subspace.
> > > As a result, the optimization is constrained to more informative and robust directions, and explicitly capturing a shared adversarial subspace that aligns with cross-model semantics,
> > > thereby improving black-box transferability.
> > > Empirically, CKA analysis (see the response to Weakness 1 of Reviewer qmzX) shows that our LRS consistently achieves higher cross-model similarity, indicating better alignment in shared subspaces.
> > >
> > >
> > > **(2) Difference in the Role of Sparsity: Perceptual Constraint vs. Discriminative Subspace Modeling**
> > >
> > > - **Existing methods:**
> > > Existing sparse attacks (e.g., SparseFool, Homotopy-Attack, Sparse-PGD) enforce input-level sparsity to enhance imperceptibility by minimizing the number of modified pixels.
> > >
> > > - **Our method:** We leverage sparsity in latent representation space to capture discriminative feature directions. This reinterprets sparsity as a discriminative subspace mechanism, which selectively emphasizes directions that drive prediction shifts, fundamentally changing its role in adversarial attack design.
> > >
> > > **(3) Difference in Optimization Strategy: Independent Decomposition vs. Iteration-aware Dynamic Optimization**
> > >
> > > - **Existing methods:**
> > > SVD-based attack rely on power iteration algorithm within each attack step, performing low-rank decomposition independently across each attack iteration.
> > > Such designs ignore temporal dependencies across iterations, resulting in redundant computations and high overhead.
> > >
> > > - **Our method:**
> > > We propose a Warm-started Alternating Low-rank Approximation (WALA) algorithm that exploits iteration continuity, shifting from static per-step estimation to iteration-aware dynamic optimization, thereby improving both efficiency and stability (see Fig.1(b)).
> > > Under a limited attack budget (e.g., the widely used 10-step setting), we decompose the USV learning process into two stages:
> > > first optimizing a regularized low-rank formulation over U and V via gradient descent, followed by a closed-form estimation of S.
> > > This design reduces computational cost and improves practical efficiency for real-world attack scenarios.

---

### Official Review · Reviewer_qmzX · 2026-03-09

**Soundness:** 3
**Presentation:** 3
**Significance:** 2
**Originality:** 2
**Overall Recommendation:** 4
**Confidence:** 4

**Summary:**

This paper propose to decompose the internal representation of a neural network into sparse and low-rank components and find adversarial attacks through passing only these components to later networks to compute the adversarial loss. There is a efficient algorithm proposed to find the decomposition with warm-up initialization. Then, the transferability of the attacks are tested with ImageNet across multiple network architectures.

**Compliance With Llm Reviewing Policy:**

Affirmed.

**Final Justification:**

The author rebuttal addressed my main concern about the motivation of using low-rank attacks, for which I increased my rating from 3 to 4. However, as other reviewers have pointed out, the main contribution of this paper is modifying the low-rank attack to a low-rank+sparse attack and introducing multiple associated experts; those ideas are not very novel and the evaluation boils down to whether there is sufficient ablation studies to showcase the effectiveness of those modifications. Given those points, I think rating 4 is appropriate.

**Key Questions For Authors:**

See weakness.

**Limitations:**

yes

**Strengths And Weaknesses:**

Strength:
1. Writings are clear.
2. The experiments are extensive in terms of the number of models and the number of attack baselines.

Weakness:

This work builds upon the work of Weng et al., where the latter just did the SVD to find low-rank component. While this fact alone is not a weakness. But there is an issue associated with it: The introduction does not motivate well the approach of "low-rank adversarial attack" (the work of Weng et al. and this work). Critically, the goal is to find transferable attacks. But why low-rank adversarial attack has better transferability is not discussed(lines 62 to 84 are merely a literature review, it does not explain why internal representation manipulation is needed for transferability).

---

> ### Author Rebuttal · Authors · 2026-03-31
>
> We thank the reviewer for the question.
>
> **Weakness 1: Why Low-Rank is Beneficial**
>
> From the perspective of attack mechanisms, existing transferable adversarial attacks are typically optimized in high-dimensional dense feature spaces.
> Such approaches are prone to overfitting to the specific response patterns of the surrogate model, thereby limiting cross-model generalization.
>
> In contrast, introducing low-rank constraints effectively restricts perturbations to a subspace spanned by dominant singular directions,
> guiding the optimization along principal semantic directions shared across models.
> These directions correspond to the major energy distribution in feature representations and tend to exhibit higher consistency across different architectures.
> As a result, low-rank modeling can suppress unstructured noise and model-specific components in high-dimensional spaces,
> reducing overfitting to the surrogate model and enabling perturbations to induce more consistent prediction shifts across models, thus improving transferability.
>
> While the low-rank component captures global shared structures, it may smooth out local sensitive features that are crucial for influencing decision boundaries.
> In contrast, the sparse component explicitly models these localized and discriminative directions. We provide attention rollout visualizations for both components (see [https://anonymous.4open.science/r/LRS_attention_rollout-DC20/attention_rollout.png](https://anonymous.4open.science/r/LRS_attention_rollout-DC20/attention_rollout.png)).
>
> From the visualization, we observe that the sparse component reintroduces critical local perturbations that may be overlooked by the low-rank component,
> thereby complementing it, enabling a balance between global consistency and local effectiveness, and improving transferability.
>
> To further verify that the low-rank component preserves cross-model shared semantic information,
> and that our LRS formulation maintains higher representational consistency across models,
> we conduct additional cross-model representation similarity analysis based on CKA. As shown in the table,
> we observe that: (1) both SVD and LRS exhibit lower CKA values than clean samples,
> since adversarial perturbations alter the original feature distributions and degrade cross-model feature alignment;
> (2) LRS consistently achieves higher cross-model CKA values than SVD,
> indicating that LRS better aligns feature subspaces across models and supports the modeling of more model-agnostic representations.
>
> | Transfer | Layer | Clean | SVD | LRS |
> | --- | --- | --- | --- | --- |
> | incv3->incv4 | Pool | 0.7986 | 0.4233 | 0.4813 |
> | incv3->incv4 | FC | 0.5626 | 0.1891 | 0.2086 |
> | incv3->incresv2 | Pool | 0.7612 | 0.3927 | 0.4495 |
> | incv3->incresv2 | FC | 0.5854 | 0.1873 | 0.2135 |
> | incv3->vgg16 | Pool | 0.5065 | 0.4512 | 0.4887 |
> | incv3->vgg16 | FC | 0.2944 | 0.1341 | 0.1532 |
> | incv3->vgg19 | Pool | 0.5079 | 0.4373 | 0.4767 |
> | incv3->vgg19 | FC | 0.2986 | 0.1310 | 0.1541 |
>
>
> **Weakness 2: Why Operate on Internal Representations**
>
> We further clarify why modeling intermediate features helps improve transferability.
>
> Compared to directly optimizing in the input space, imposing structural constraints in the intermediate feature space introduces a stronger inductive bias.
> Specifically, the input space is typically high-dimensional, dense, and redundant,
> where the optimization process can easily exploit model-specific fine-grained patterns or high-frequency noise.
> This often leads to overfitting to the surrogate model and consequently weakens cross-model generalization.
>
> In contrast, the intermediate feature space has undergone a series of transformations and compressions through the early layers of the network,
> thereby preserving more stable semantic representations, improving transferability.
> Recent works that incorporate feature-space considerations into attack design (e.g., BFA [1], P2FA [2]) have demonstrated this advantage.
>
>
> Imposing structural constraints such as low-rank modeling in this space essentially introduces a form of structured regularization on the perturbations,
> restricting them to a subspace spanned by dominant feature directions. These directions usually correspond to semantic structures shared across models, and are therefore more model-agnostic.
>
> From this perspective, performing structured modeling in the feature space effectively avoids redundant degrees of freedom in the input space,
> suppresses interference from model-specific noise, and guides perturbations to evolve along more stable and transferable representation directions,
> thereby significantly improving the generalization ability of cross-model attacks.
>
> [1] Improving the transferability of adversarial examples through black-box feature attacks. Neurocomputing 2024.
>
> [2] Pixel2Feature Attack (P2FA): Rethinking the Perturbed Space to Enhance Adversarial Transferability. ICML 2025.

---

> > ### Author Rebuttal · Reviewer_qmzX · 2026-03-31
> >
> > I have adjusted the score accordingly

---

> > > ### Author Response · Authors · 2026-04-05
> > >
> > > We sincerely thank the reviewer for the positive feedback and for adjusting the score. We are glad that our clarifications were helpful in addressing your concerns, and we greatly appreciate your recognition of our efforts.
> > >
> > > We will incorporate the reviewer’s suggestions and further refine the presentation of the underlying motivation and feature-space analysis in the revised manuscript.

---

### Official Review · Reviewer_8dtN · 2026-03-11

**Soundness:** 3
**Presentation:** 3
**Significance:** 3
**Originality:** 3
**Overall Recommendation:** 4
**Confidence:** 4

**Summary:**

This paper proposes a transferable adversarial attack framework that enhances black-box transferability by decomposing intermediate features into low-rank and sparse components. Building on this, the paper further introduces a hierarchical expert design across shallow, middle, and deep layers. Shallow layers emphasize sparse structures, deep layers emphasize low-rank structures, and middle layers use a joint decomposition. To reduce the computational overhead of repeated decomposition during iterative attack optimization, the paper also proposes a warm-started alternating low-rank approximation solver. Experimental results show the effectiveness of the proposed methods.

**Compliance With Llm Reviewing Policy:**

Affirmed.

**Final Justification:**

The authors' rebuttal has addressed most of my concerns. However, I did not see much why the low-rank information should correspond to cross-model shared vulnerability rather than merely to dominant surrogate-specific responses, which may exceed the scope of this work. I choose to maintain the positive score.

**Key Questions For Authors:**

1.	The idea of decomposing intermediate features before perturbation optimization is reasonable. I would like the authors to further explain what information is preserved in the low-rank subspace and why this information should correspond to cross-model shared vulnerability rather than merely to dominant surrogate-specific responses. Could the authors provide more evidence, e.g., cross-model subspace alignment, representation similarity analysis, showing that low-rank components indeed encode more model-agnostic semantic structure.
2.	The method contains many components, making it difficult to isolate the main source of improvement, and the expert assignment appears heuristic and is not fully ablated.
3.	The proposed WALA update is also heuristically motivated. However, it is unclear whether the alternating update admits any meaningful convergence guarantee, or under what conditions it can approximate the desired low-rank structure during iterative attack optimization.
4.	The experiments include a wide range of classical transfer-based attack baselines. However, many of these methods are out of date.

**Limitations:**

Yes

**Strengths And Weaknesses:**

### Strengths
1.	The paper is interesting and well-motivated. The decomposition perspective is intuitive, and the method is broadly compatible with existing attack frameworks.
2.	The method is evaluated on multiple attack baselines and tested against diverse target architectures, including both CNNs and vision transformers, as well as defended models.

### Weaknesses
1. The central mechanism behind why low-rank decomposition should improve transferability is still not fully validated.
2. Some key design choices, especially the depth-specific assignment, seem more heuristic than rigorously justified.

---

> ### Author Rebuttal · Authors · 2026-03-31
>
> We thank the reviewer for the insightful suggestion.
>
> **Question 1: Model-agnostic semantic structure**
>
> The low-rank component captures dominant directions of feature variation,
> which are more consistent across model architectures due to their association with global semantics.
> To further validate this, we provide additional cross-model CKA analysis. Detailed results are given in our response to Reviewer qmzX (Question 1).
>
>
> **Question 2: Expert Assignment**
>
> To evaluate the contribution of each component and validate the expert assignment strategy, we conduct ablation studies by systematically varying cross-layer expert configurations (see table).
>
> First, compared to the baseline without experts (“w/o expert”), all expert-based variants consistently improve performance, demonstrating the effectiveness of introducing structured decomposition into intermediate features.
>
> Second, applying a uniform structure across all layers (e.g., all-sparse “all-s” or all-low-rank “all-l”) leads to inferior performance, indicating that a single structure is insufficient to capture the complexity of feature representations.
>
> Finally, different layer-wise combinations of low-rank (l), sparse (s), and mixed (m) structures exhibit significant performance differences.
> Notably, our configuration (s-m-l, i.e., shallow sparse – middle mixed – deep low-rank) consistently achieves the best results across both CNN and ViT settings,
> suggesting that expert assignment aligns with hierarchical representation properties rather than being heuristic.
>
>
> | source | method | incv3 | vgg19 | densenet121 | cait-s | deit-b | vit-b | avg |
> | --- | --- | --- | --- | --- | --- | --- | --- | --- |
> | incv3 | w/o expert | 100.0 | 59.9 | 47.4 | 15.0 | 17.1 | 25.0 | 44.1 |
> | incv3 | all-s expert | 100.0 | 65.5 | 57.5 | 18.9 | 19.0 | 25.9 | 47.8 |
> | incv3 | all-l expert | 99.9 | 64.8 | 56.2 | 19.1 | 19.6 | 26.2 | 47.6 |
> | incv3 | l-m-s expert | 100.0 | 61.4 | 51.1 | 17.1 | 18.0 | 24.5 | 45.4 |
> | incv3 | l-s-m expert | 100.0 | 64.2 | 57.0 | 17.8 | 20.3 | 26.5 | 47.6 |
> | incv3 | s-l-m expert | 100.0 | 65.1 | 58.4 | 18.1 | 19.2 | 26.1 | 47.8 |
> | incv3 | s-m-l expert (ours) | 100.0 | 66.5 | 60.4 | 19.6 | 21.0 | 26.5 | 49.0 |
> | vit-b | w/o expert | 39.0 | 57.3 | 44.8 | 44.4 | 49.0 | 99.9 | 55.7 |
> | vit-b | all-s expert | 52.7 | 69.6 | 59.7 | 75.2 | 79.1 | 100.0 | 72.7 |
> | vit-b | all-l expert | 52.6 | 69.3 | 60.9 | 76.0 | 78.6 | 100.0 | 72.9 |
> | vit-b | l-m-s expert | 53.0 | 70.0 | 60.2 | 76.2 | 77.4 | 100.0 | 72.8 |
> | vit-b | l-s-m expert | 52.2 | 68.0 | 60.3 | 76.6 | 78.1 | 100.0 | 72.5 |
> | vit-b | s-l-m expert | 52.7 | 68.3 | 60.5 | 76.5 | 76.3 | 100.0 | 72.4 |
> | vit-b | s-m-l expert (ours) | 53.5 | 70.3 | 61.8 | 77.3 | 79.9 | 100.0 | 73.8 |
>
>
> **Question 3: Convergence of WALA Update**
>
> We further clarify that the core updates of WALA (Eqs. 8–9) are an alternating gradient optimization over the low-rank factors (U_l) and (V_l).
>
> Under standard conditions such as Lipschitz continuity and smoothness, this type of alternating optimization generally exhibits favorable theoretical properties in non-convex settings and is typically considered to converge to a stationary point of the objective.
>
> Importantly, in the context of adversarial attacks, our goal is not to obtain a strictly globally convergent solution, but rather to achieve an effective structural approximation within a limited iteration budget, thereby improving transferability and efficiency. As such, the optimization process inherently reflects a trade-off between optimization accuracy and attack efficiency.
>
> Moreover, the warm-start mechanism initializes each update using the decomposition from the previous iteration, leveraging the continuity of representation evolution to enhance stability and accelerate convergence, allowing the low-rank structure to be more effectively approximated within limited steps.
>
> In summary, WALA is not an empirical heuristic, but a principled adaptation of the standard alternating optimization framework tailored to the limited-step nature of adversarial attacks.
>
>
> **Question 4: Concerns on Outdated Baseline Methods**
>
> Although some baseline methods remain de facto standards for evaluating transferability (e.g., MI-FGSM, DI-FGSM, TI-FGSM), they are still widely used in recent work.
>
> In addition, we include a more recent method, GI-FGSM (2024), to reflect progress in gradient-based attack design.
>
> We also note that recent research has gradually shifted from designing new FGSM variants to improving transferability through alternative mechanisms.
> To provide a more comprehensive evaluation, we further include such methods in the Appendix (Table 3), mainly covering:
> (i) gradient modulation and sampling strategies (e.g., PGN, 2023; GGS, 2025), and
> (ii) feature-level attack methods (e.g., BFA, 2024; P2FA, 2025).
>
> Overall, the selected baselines cover both classical methods and recent representative approaches, providing a comprehensive and fair evaluation.

---

> > ### Author Rebuttal · Reviewer_8dtN · 2026-04-04
> >
> > My questions are fully resolved. Thank you to the authors for your efforts.

---

> > > ### Author Response · Authors · 2026-04-05
> > >
> > > We sincerely thank the reviewer for the encouraging feedback. We are pleased that all concerns have been fully addressed, and we truly appreciate your thoughtful evaluation and support.
> > >
> > > We will incorporate the reviewer’s comments and further strengthen the corresponding analyses and clarifications in the revised manuscript.

---

### Official Review · Reviewer_S5WP · 2026-03-15

**Soundness:** 3
**Presentation:** 3
**Significance:** 3
**Originality:** 2
**Overall Recommendation:** 3
**Confidence:** 4

**Summary:**

This Paper explicitly enforce rank and sparsity constraints, enabling precise control over global vs. local feature allocation. This is crucial in adversarial optimization, where decomposition is repeatedly applied across iterative gradient updates. which enabling the generation of diverse and highly transferable adversarial perturbations.  The paper  design a hierarchical mixture of robust experts, applying sparse, balanced, and low-rank constraints to shallow, middle, and deep layers respectively, which significantly improves cross-layer consistency and adversarial transferability.

**Compliance With Llm Reviewing Policy:**

Affirmed.

**Key Questions For Authors:**

1. Figure 4. Effect of layer configuration on black-box Attack Success Rate (ASR). Results are averaged across the five CNN models mentioned. The single-layer SVD variant achieves an ASR of 76.4, while the multi-layer SVD variant achieves 76.2. Regarding the single-layer and multi-layer SVD variants, please explain why the single-layer metric is marginally higher than the multi-layer metric?。。
2. SVD-based attack (Weng et al., 2024)  enhances transferability by preserving the dominant singular directions of intermediate activations. This technique can be integrated into a variety of different methods to boost their performance.  Can SVD be equated to low-rank and sparse decomposition, or does it offer a more robust foundational capability?

**Limitations:**

yes

**Strengths And Weaknesses:**

This paper pioneers the introduction of low-rank and sparse decomposition into the realm of adversarial attacks, uncovering the complementary interplay between global semantic structures and local discriminative patterns within deep features. This novel perspective provides a fresh framework for understanding the generation mechanisms of adversarial examples. Furthermore, by designing a Hierarchical Mixture of Robust Experts, the paper innovatively integrates the distinctive characteristics of deep features across different layers, leading to a substantial enhancement in attack transferability. Hyperparameter Sensitivity: While the paper offers an analysis of hyperparameter selection, it lacks a thorough investigation into strategies for automatically tuning these parameters to accommodate diverse tasks or datasets. This limitation may hinder the method's practical applicability in real-world scenarios. Reliance on Existing Tools: Despite proposing an innovative decomposition strategy, the core ideas of the paper are still rooted in established low-rank and sparse decomposition theories, without achieving a complete departure from the framework of traditional methodologies. Consequently, the contributions of this work are primarily oriented towards the optimization and integration of existing techniques, resulting in relatively limited originality.

---

> ### Author Rebuttal · Authors · 2026-03-31
>
> We thank the reviewer for the insightful question.
>
> **Question 1: Single-layer vs. Multi-layer SVD**
>
> We attribute this to differences in optimization behavior between single-layer and multi-layer modeling.
> Single-layer SVD focuses optimization on a specific layer,
> producing stronger local perturbations and sometimes slightly higher attack success rates,
> but it is more prone to overfitting the surrogate model.
> In contrast, multi-layer modeling introduces cross-layer constraints to capture complementary information,
> improving stability and cross-model consistency, but may disperse optimization strength, leading to comparable or slightly lower performance in a single setting.
>
>
> We further observe that the benefit of multi-layer modeling depends on the diversity of layer-wise representations. In VGG-style networks, feature evolution across layers is relatively smooth and homogeneous, leading to limited gains. In contrast, architectures such as Inception and DenseNet introduce more diverse transformations (e.g., multi-branch processing and dense connections), which increase representation diversity and make multi-layer modeling more advantageous.
>
> Overall, although the multi-layer variant does not always achieve the highest ASR in specific settings, it demonstrates more consistent performance across different architectures and settings, indicating better generalization ability and transfer robustness.
>
>
> **Question 2: Can SVD be equated to low-rank and sparse decomposition**
>
>  We would like to clarify that SVD-based methods (e.g., Weng et al., 2024) can be viewed as a special case of our proposed low-rank and sparse (LRS) framework. Specifically, we consider:
>
> $$
> \min_{L_l, S_l} \|H_l D_l - L_l - S_l\|_F^2, \quad
> \text{s.t. } \operatorname{rank}(L_l) \le r,\ \|S_l\|_0 \le k
> $$
>
> When the sparsity constraint is suppressed (i.e., $k = 0$, leading to $S_l = 0$), the formulation reduces to the standard low-rank approximation:
>
> $$
> \min_{L_l} \|H_l D_l - L_l\|_F^2, \quad
> \text{s.t. } \operatorname{rank}(L_l) \le r
> $$
>
> According to the Eckart–Young–Mirsky theorem, the optimal solution corresponds to truncated SVD. Therefore, SVD-based methods can be interpreted as a special case of our framework that models only the low-rank component, while LRS provides a more general formulation.
>
> Building upon this, our method extends SVD-based modeling in two key aspects:
>
> (1) **Structured decomposition:** LRS explicitly decomposes features into a low-rank component $L_l$ (capturing global semantic structure) and a sparse component $S_l$ (capturing localized discriminative variations). This enables joint modeling of cross-model shared structure and local perturbation patterns, whereas SVD focuses solely on the former.
>
> (2) **Dynamic optimization:** Our WALA update incorporates a warm-start strategy, leveraging the continuity of feature evolution across iterations to improve optimization stability and efficiency. In contrast, standard SVD is solved independently at each iteration without exploiting such inter-iteration continuity.
>
>
> **Weakness 1: Hyperparameter Sensitivity**
>
>  We would like to clarify that our method does not rely on static hyperparameter tuning. Instead, we model the coupling between low-rankness and sparsity as a function of network depth. Specifically, the rank ratio and sparsity level are not treated as independent fixed hyperparameters, but as depth-dependent variables that evolve across layers.
>
> In particular, our strategy can be interpreted as following a structured evolution pattern:
> $$
> \kappa(l+1) \ge \kappa(l), \quad \rho_s(l+1) \le \rho_s(l)
> $$
> where $\kappa(l)$ and $\rho_s(l)$ denote the relative strength of low-rank and sparse components at layer $l$, respectively. This reflects a monotonic transition from sparse, local patterns in shallow layers to low-rank, global structures in deeper layers.
>
> This design is well aligned with the hierarchical nature of deep representations, evolving from local to global semantics, and thus naturally generalizes across different model architectures without requiring model-specific tuning.
> Empirically, we observe consistent and stable performance across various models (see Figure 3).
>
>
> **Weakness 2: Originality**
>
> We acknowledge that this work is rooted in classical low-rank and sparse decomposition theory.
> However, our core contribution lies in rethinking and redefining the role of such structures in adversarial attacks, which has not been investigated in prior work.
> Prior works mainly focus on manipulating dense or low-rank feature maps, completely different from our hierarchical low-rank and spare feature maps.
>
> We provide a comprehensive reformulation and innovation of both the modeling of low-rank–sparse structures and the optimization process for adversarial attacks.
>
> Detailed analysis can be found in our response to Reviewer Msug (Question 1).

---

> > ### Author Rebuttal · Reviewer_S5WP · 2026-04-05
> >
> > I thank authors for the response. I have already understood the author's main points.

---

> > > ### Author Response · Authors · 2026-04-05
> > >
> > > We are glad that the reviewer finds the concerns addressed.
> > >
> > > If the clarifications and additional evidence sufficiently resolve the raised issues, we would greatly appreciate it if the reviewer could reconsider the evaluation and update the score accordingly.

---

### Decision · Program_Chairs · 2026-04-30

**Decision:**

Accept (regular)

**Comment:**

This work aims to propose a transferable adversarial attack framework which explicitly decomposes intermediate representations into global semantic components and localized discriminative perturbations.

The authors' rebuttal partially resolved the reviewers' concerns. However, the proposed method is still considered to have certain limitations in terms of novelty. Several reviewers view the proposed method as an incremental work that applies classical decomposition techniques to the adversarial attack domain without providing sufficient novel insight. This work lacks a sufficiently new learning principle or algorithmic breakthrough that differentiates this from a well-executed application of Robust PCA. Therefore, this paper is recommended for “Weak accept”.